# BFTS: Thompson Sampling with Bayesian Additive Regression Trees

**Ruizhe Deng** [1]  **Bibhas Chakraborty** [2]  **Ran Chen** [3]  **Yan Shuo Tan** [1]

## Abstract

We propose Bayesian Forest Thompson Sampling (BFTS), which performs Thompson sampling using arm-wise Bayesian Additive Regression Trees (BART) to model each action's mean reward and generate MCMC-based posterior draws for decision-making. We derive an information-theoretic Bayesian regret bound of order $\tilde{\mathcal{O}}(K\sqrt{T})$ for ideal posterior sampling under a correctly specified Bayesian design. Empirically, BFTS achieves competitive regret on nonlinear synthetic benchmarks with near-nominal uncertainty calibration, attains the best average rank across nine OpenML contextual bandit benchmarks, and yields higher estimated policy values than linear, neural, and tree-ensemble baselines in a Drink Less micro-randomized trial case study. Across OpenML benchmarks, BFTS is robust to hyperparameter choices.

## 1. Introduction

Contextual bandits are central to mobile health (mHealth) interventions, where an agent must repeatedly select treatments based on a user's context to maximize health outcomes (Langford & Zhang, 2007; Klasnja et al., 2015; Deliu et al., 2024). A dominant approach is *Thompson Sampling* (TS) (Thompson, 1933; Chapelle & Li, 2011; Russo et al., 2018), which balances exploration and exploitation by sampling actions from a Bayesian posterior over reward functions. The performance of TS in contextual bandits is fundamentally limited by the quality of this underlying reward model.

**The modeling gap.** In mHealth applications, the standard practice is to employ *linear* reward models due to their simplicity and interpretability (Klasnja et al., 2015; Lei et al., 2022). However, human behavior is rarely linear; these models suffer from high bias and fail to capture complex interactions or threshold effects inherent in health data.

To address nonlinearity, recent works have proposed *Kernel* (Chowdhury & Gopalan, 2017) or *Neural* (Zhang et al., 2021) TS. While expressive, these approaches present significant hurdles for real-time deployment: they are computationally expensive and notoriously sensitive to hyperparameters. In an online setting where data arrives sequentially, standard tuning techniques (like cross-validation) are infeasible, making these models brittle in practice.

**The tree ensemble paradox.** For tabular data—which characterizes most mHealth contexts—*tree ensembles* (e.g., Random Forests, Gradient Boosting) are state-of-the-art for offline prediction (Grinsztajn et al., 2022). They handle nonlinearity, interactions, sparsity, and heterogeneous variable scales or types out of the box, but lack the Bayesian generative structure required by Thompson Sampling. Adapting them for TS typically requires heuristics to approximate uncertainty (Nilsson et al., 2024), which breaks the theoretical link between the posterior and the regret. Consequently, rigorous regret guarantees for tree-based bandits remain difficult to obtain.

**Our contribution: Bayesian Forest Thompson Sampling (BFTS).** We propose BFTS to close this gap. We replace heuristic tree ensembles with *Bayesian Additive Regression Trees (BART)* (Chipman et al., 2010; Hill et al., 2020)—a fully probabilistic sum-of-trees model. BFTS retains the predictive power of trees while providing a principled, exact Bayesian posterior for Thompson Sampling. In summary, our contributions are:

(i) **Algorithm:** We introduce BFTS, the first contextual bandit method leveraging BART's exact posterior for Thompson sampling, combining tree robustness with principled uncertainty with minimal tuning.

(ii) **Theory:** We derive an information-theoretic Bayesian regret bound $\mathbb{E}[\text{Regret}_T] = \tilde{\mathcal{O}}(K\sigma\sqrt{T})$ and provide supporting frequentist analysis for a feel-good variant under sparsity assumptions.

(iii) **Empirics:** BFTS achieves low regret on synthetic

---

[1]Department of Statistics and Data Science, National University of Singapore, Singapore [2]Centre for Biomedical Data Science, Duke-NUS Medical School, Singapore [3]Department of Statistics and Data Science, Washington University in St. Louis, USA. Correspondence to: Yan Shuo Tan <yanshuo@nus.edu.sg>.

*Proceedings of the 43rd International Conference on Machine Learning*, Seoul, South Korea. PMLR 306, 2026. Copyright 2026 by the author(s).

benchmarks, best average rank across nine OpenML datasets, and strong offline policy evaluation results on the Drink Less mHealth trial, with robust performance across hyperparameter settings.

**Conflict of Interest Disclosure.** The authors declare no financial conflicts of interest related to this work.

## 2. Related Work

**Non-linear contextual bandits.** While linear Thompson Sampling (Agrawal & Goyal, 2013) is well-understood, its bias in complex environments has motivated the use of richer function approximators. *Kernelized/GP Bandits* (Krause & Ong, 2011; Chowdhury & Gopalan, 2017) offer a non-parametric Bayesian alternative, but they scale poorly with the horizon ($\mathcal{O}(T^3)$ for exact inference) or rely on approximations that dilute their theoretical guarantees (Zenati et al., 2022). *Neural Bandits* (Zhang et al., 2021) capture rich representations but face two hurdles. Theoretically, their analysis relies on the "lazy training" (NTK) regime requiring unrealistic network widths. In practice, performance depends on hyperparameters (e.g., learning rate, architecture) that are difficult to tune online, where cross-validation is inapplicable.

**Tree-based bandits.** Despite the dominance of tree ensembles in tabular supervised learning (Grinsztajn et al., 2022), their adoption in bandits has been slowed. Féraud et al. (2016) only uses decision stumps, while practical tree-ensemble bandit implementations, (Elmachtoub et al., 2017; Nilsson et al., 2024) often rely on heuristic uncertainty quantification and lack regret bounds. BFTS addresses this by replacing the heuristic quantification with a valid MCMC sampler targeting the exact BART posterior. A complementary line of work analyzes tree-based bandit regret through a tree neural tangent kernel (TNTK) on *soft* trees (Iwazaki & Suzumura, 2024), whereas our analysis targets the exact, hard-tree BART posterior via direct MCMC.

**The theoretical foundations of BART.** Chipman et al. (2010) introduced BART as a sum-of-trees prior paired with a likelihood. A rich theoretical literature has since emerged, establishing the frequentist optimality of the BART posterior. Seminal works have proved posterior contraction rates for BART over $\alpha$-Hölder smooth function classes (Linero, 2018; Ročková & van der Pas, 2020; Ročková & Saha, 2019), confirming that the Bayesian ensemble adapts to the underlying smoothness and sparsity of the regression function. More recently, Jeong & Ročková (2023) extended these results to spatially inhomogeneous functions, showing that BART can adapt to varying local smoothness.

## 3. Methodology

### 3.1. Problem Setup

We consider the standard contextual bandit setting over $T$ rounds. At each round $t \in [T]$ (where $[T] \stackrel{\text{def}}{=} \{1, \ldots, T\}$), the learner observes a context vector $X_t \in \mathcal{X} \subseteq [0,1]^p$, selects an action $A_t \in \mathcal{A}$ (where $|\mathcal{A}| = K$), and receives a scalar reward:

$$R_t = f_0(X_t, A_t) + \varepsilon_t,$$

where $f_0 : \mathcal{X} \times \mathcal{A} \to \mathbb{R}$ is the unknown mean reward function and $\varepsilon_t$ is independent mean-zero noise. We denote the history as $\mathcal{H}_t = \{(X_s, A_s, R_s)\}_{s=1}^t$. The goal is to minimize cumulative regret against the (oracle) optimal policy that knows the underlying reward function and always takes the best action $A^*(x) \in \arg\max_{a \in \mathcal{A}} f_0(x, a)$ for any given context $x$.

We adopt a *Thompson Sampling* (TS) approach. TS maintains a Bayesian posterior $\Pi(\cdot \mid \mathcal{H}_{t-1})$ over a model class $\{f_{\boldsymbol{\theta}} : \boldsymbol{\theta} \in \boldsymbol{\Theta}\}$. In each round, it samples a parameter $\widehat{\boldsymbol{\theta}}_t \sim \Pi(\cdot \mid \mathcal{H}_{t-1})$ and acts greedily with respect to the sampled model: $A_t \in \arg\max_{a \in \mathcal{A}} f_{\widehat{\boldsymbol{\theta}}_t}(X_t, a)$.

### 3.2. Bayesian Forest Specification

We model the reward function using *independent arm-wise BART priors*. Specifically, we posit that $f_0(x, a) = g_a(x)$ for distinct functions $g_a : \mathcal{X} \to \mathbb{R}$, each modeled by an independent BART ensemble with $m$ trees:

$$f_{\boldsymbol{\theta}}(x, a) = g_{\theta_a}(x) = \sum_{j=1}^m g_{\mathcal{T}_j^{(a)}, M_j^{(a)}}(x), \quad \boldsymbol{\theta} = ((\theta_a))_{a \in \mathcal{A}}.$$

Here, $\theta_a = \{(\mathcal{T}_j^{(a)}, M_j^{(a)})\}_{j=1}^m$ denotes the forest structure and leaf parameters for arm $a$.

**Prior and likelihood.** We adopt a Gaussian likelihood $R_t \sim \mathcal{N}(f_{\boldsymbol{\theta}}(X_t, A_t), \sigma^2)$ with a conjugate inverse-gamma prior on $\sigma^2$. For the prior parameters, we adopt the default BART settings for the leaf value variance $\sigma_\mu^2 = (0.5/(\kappa\sqrt{m}))^2$ where $\kappa = 2$. As recommended by the theoretical analysis of Ročková & van der Pas (2020), for tree structures we use a depth-exponential splitting prior. Specifically, given a node $\Omega \subset \mathcal{X}$ at depth $d(\Omega)$, the probability that node splits is given by $p_{\text{split}}(\Omega) = \alpha_{\text{qd}}^{d(\Omega)}$ with $\alpha_{\text{qd}} = 0.45$; this ensures a stronger decay rate than the original BART prior (Chipman et al., 2010), which uses $p_{\text{split}}(\Omega) = \alpha_o(1 + d(\Omega))^{-\beta_o}$. We adopt the quick-decay form for three reasons: (i) it admits a clean control of the tree-structure entropy and the expected leaf count, which simplifies the structure term in Lemma 2; (ii) the frequentist analysis in Appendix E relies on BART contraction results that are stated under quick-decay; and (iii) it empirically

outperforms the original Chipman et al. prior in our sensitivity sweep (Figure 8). The split thresholds are selected uniformly at random from a grid, while the split axes are either selected uniformly at random or from a Dirichlet distribution (Linero, 2018). Detailed prior specifications and hyperparameters are provided in Appendix B.

### 3.3. BFTS Algorithm and Inference

We now define *Bayesian Forest Thompson Sampling* (BFTS). Conceptually, BFTS performs standard Thompson Sampling using the BART posterior defined in Section 3.2.

**Ideal vs. practical execution.** For theoretical clarity, it is useful to distinguish between the ideal exploration scheme and its computational realization:

- **Ideal BFTS:** An idealized algorithm that draws samples *exactly* from the posterior $\Pi(\boldsymbol{\theta} \mid \mathcal{H}_t)$ at every round. Our regret analysis in Section 4 establishes bounds for this idealized process, assuming perfect inference.

- **Practical BFTS (Algorithm 1):** Since exact sampling from the BART posterior is analytically intractable, our deployed algorithm approximates Ideal BFTS using Markov Chain Monte Carlo (MCMC).

**Arm modeling and action encoding** Contextual bandit methods in experiments need to specify how the discrete action $a \in \mathcal{A}$ enters the reward model (Zhang et al., 2021). We refer to three choices: (i) *separate-arm* modeling, where one fits $K$ independent functions $g_a(x)$ so that $f(x,a) = g_a(x)$; (ii) a single joint model with a `one-hot` arm indicator, using $\tilde{x} = (\mathbf{e}_a, x) \in \mathbb{R}^{K+p}$; and (iii) `multi`/positional encoding, mapping $x \in \mathbb{R}^p$ to $\tilde{x}_a \in \mathbb{R}^{Kp}$ by placing $x$ in the $a$-th block and zero-padding elsewhere.

BFTS adopts the separate-arm specification. This choice preserves the arm-wise posterior factorization needed by our regret analysis (see Section 4) and avoids action-encoding-induced dimension inflation (from $p$ to $p + K$ or $Kp$) that can weaken nonparametric rates. Empirically, separate-arm performs best overall in sensitivity analysis (see Section 8).

**Posterior computation via MCMC.** To implement Practical BFTS, we run independent MCMC chains for each arm in parallel using a Metropolis-within-Gibbs sampler. The sampler iteratively updates each tree $\mathcal{T}_j^{(a)}$ in the ensemble by proposing a structural modification (e.g., grow, prune), which is accepted or rejected via a standard Metropolis-Hastings step. Conditional on the tree structure, the leaf parameters $M_j^{(a)}$ are sampled directly from their conjugate Gaussian full conditionals. Finally, after a complete pass over all $m$ trees, the noise variance $\sigma^2$ is drawn from its

---

**Algorithm 1** Bayesian Forest Thompson Sampling (BFTS)

1: **Input:** Prior $\Pi$, Horizon $T$, Refresh schedule $r(\cdot)$, initial round-robin length $\tau$ (per arm), Burn-in $n_{\text{burn}}$, Samples $n_{\text{post}}$
2: Initialize data $\mathcal{H}_0 = \emptyset$
3: Initialize stored posterior samples $\mathcal{S} = \emptyset$
4: **for** $t = 1$ **to** $T$ **do**
5:     Observe context $X_t$
6:     **if** $t \le \tau K$ **then**
7:         {Phase 1: Initialization (Round-Robin)}
8:         Select $A_t$ sequentially from actions $\{1, \ldots, K\}$
9:     **else**
10:        {Phase 2: Thompson Sampling}
11:        Draw $\widehat{\boldsymbol{\theta}}$ from $\mathcal{S}$ without replacement {Reshuffle $\mathcal{S}$ if it is exhausted}
12:        Compute greedy action $A_t = \arg\max_{a \in \mathcal{A}} f_{\widehat{\boldsymbol{\theta}}}(X_t, a)$
13:     **end if**
14:     Observe reward $R_t$
15:     Update history $\mathcal{H}_t \leftarrow \mathcal{H}_{t-1} \cup \{(X_t, A_t, R_t)\}$
16:     **if** $r(t) > r(t-1)$ **or** $t = \tau K$ **then**
17:        {Phase 3: Posterior Refresh (MCMC)}
18:        Run MCMC for $n_{\text{burn}} + n_{\text{post}}$ steps using data $\mathcal{H}_t$
19:        Discard first $n_{\text{burn}}$ samples
20:        Update pool $\mathcal{S} \leftarrow \{\boldsymbol{\theta}^{(s)}\}_{s=1}^{n_{\text{post}}}$
21:     **end if**
22: **end for**

---

Inverse-Gamma full conditional. Detailed formulas are provided in Appendix B.3.

**Batched updates and mixing.** Running a full MCMC chain at every round $t$ is computationally prohibitive. We adopt a *batched* schedule defined by a refresh function $r(t)$. The posterior is updated only when $r(t) > r(t-1)$. At each refresh, we use a cold-start MCMC and re-initialize the sampler from the prior to prevent the chains from becoming trapped in local modes. Between refresh points, the posterior is held fixed. To perform Thompson Sampling in these intermediate rounds, we draw samples at random from the posterior sample pool generated in the last refresh. This batching perspective is related to adaptive batched Thompson Sampling, which can retain near-optimal regret guarantees with only a logarithmic number of batch updates (Kalkanlı & Özgür, 2021).

## 4. Regret Analysis

We now analyze the theoretical properties of Ideal BFTS (as defined in Section 3). Our primary contribution is a bound on the *Bayesian regret*, quantifying the efficiency of the Bayesian forest prior in the bandit setting.

The Bayesian regret is defined under a *Bayesian design*: the true reward functions are drawn from the prior, i.e., for each arm $a \in \mathcal{A}$, the function $f_0(\cdot, a)$ is a realization of a BART prior $\Pi_{BART}$. The expected Bayesian cumulative regret is $\mathbb{E}[\text{Regret}_T] = \mathbb{E}[\sum_{t=1}^{T}(f_0(X_t, A_t^*) - f_0(X_t, A_t))]$, where the expectation is taken over the prior, contexts, and noise. We adopt standard regularity conditions: (A1) Gaussian noise with fixed known variance $\sigma^2$; (A2) exogenous i.i.d. contexts independent of $\boldsymbol{\theta}^*$; and (A3) tree splits are represented as non-redundant refinements on a fixed finite grid of size $N_{\max}$ (Chipman et al., 2010); see Appendix D for the finite-structure technical details.

**Theorem 1** (Bayesian regret of BFTS). *Let $\Pi = \Pi_{BART}^{\otimes K}$ be the product BART prior on $f_0$ with $m$ trees per arm. Under assumptions (A1)–(A3), the expected Bayesian regret of Ideal BFTS satisfies:*

$$\mathbb{E}[\text{Regret}_T] \leq K\sqrt{C_{\text{rp}}\, T \log(eKT)\, m\Psi_T},$$

*where $\Psi_T = C_{\text{str}} \log(pN_{max}) + \frac{1}{2} C_{\text{leaf}} \log\big(1 + \frac{T}{4K\kappa^2\sigma^2}\big)$ is the information complexity term, and $C_{\text{rp}}, C_{\text{str}}, C_{\text{leaf}}$ are finite constants independent of $T$ and $K$. In particular, for fixed BART model hyperparameters and split grid, the regret is $\widetilde{\mathcal{O}}(K\sqrt{T})$.*

*Proof.* The proof combines the information ratio bound of Russo & Van Roy (2016) with a novel bound on the mutual information for BART ensembles. Standard results bound Bayesian regret by $\mathbb{E}[\text{Regret}_T] \leq \sqrt{T\bar{\Gamma}_T I(\boldsymbol{\theta}^*; \mathcal{H}_T)}$. Appendix D controls $\bar{\Gamma}_T$ using a history-dependent predictive proxy and bounds $I(\boldsymbol{\theta}^*; \mathcal{H}_T)$ via Lemma 2. $\square$

**Lemma 2** (Information gain of Bayesian forests). *Under the assumptions of Theorem 1, the mutual information between the true Bayesian forest parameters $\boldsymbol{\theta}^*$ and the history $\mathcal{H}_T$ satisfies $I(\boldsymbol{\theta}^*; \mathcal{H}_T) \leq Km\Psi_T$.*

**Proof sketch of Lemma 2.** The detailed proof is in Appendix D.3.2. The argument proceeds in four steps.

*Step 1: Factorization.* Although $A_t$ depends on past rewards, Bayes' rule conditions on the realized history: given $(\mathcal{H}_{t-1}, X_t)$, the algorithm's randomized decision rule is a fixed function of $(\mathcal{H}_{t-1}, X_t)$ and fresh internal randomness, hence $p(A_t \mid \mathcal{H}_{t-1}, X_t, \boldsymbol{\theta}^*) = p(A_t \mid \mathcal{H}_{t-1}, X_t)$. With exogenous contexts, sequential factorization gives

$$p(\mathcal{H}_T \mid \boldsymbol{\theta}^*) = \prod_{t=1}^{T} p(X_t \mid \mathcal{H}_{t-1})\, p(A_t \mid \mathcal{H}_{t-1}, X_t)$$
$$\times p(R_t \mid X_t, A_t, \theta_{A_t}^*)$$
$$\propto \prod_{t=1}^{T} p(R_t \mid X_t, A_t, \theta_{A_t}^*),$$

so conditional on the realized design $(X_{1:T}, A_{1:T})$ the reward likelihood factorizes by arm. Combined with the product prior $\Pi = \Pi_{BART}^{\otimes K}$, this yields the exact posterior product form $\Pi(\boldsymbol{\theta}^* \mid \mathcal{H}_T) = \bigotimes_{a=1}^{K} \Pi_a(\theta_a^* \mid \mathcal{H}_T)$; see Lemma 9 in Appendix D.3.2. Therefore, the mutual information factors into a sum: $I(\boldsymbol{\theta}^*; \mathcal{H}_T) = \sum_{a=1}^{K} I(\theta_a^*; \mathcal{H}_T)$.

*Step 2: Structure-leaf decomposition.* For a single arm, we use the chain rule to decompose the information into the tree structure $\mathcal{T}^{(a)} = \{\mathcal{T}_j^{(a)}\}_{j=1}^{m}$ and leaf parameters $M^{(a)} = \{M_j^{(a)}\}_{j=1}^{m}$, i.e. $I(\theta_a; \mathcal{H}_T) = I(\mathcal{T}^{(a)}; \mathcal{H}_T) + I(M^{(a)}; \mathcal{H}_T \mid \mathcal{T}^{(a)})$.

*Step 3: Bounding tree structure entropy.* The tree structure term $I(\mathcal{T}^{(a)}; \mathcal{H}_T)$ is bounded by the entropy of the prior. The depth-exponential splitting prior ensures this entropy is logarithmic in the grid size $N_{\max}$, i.e. $I(\mathcal{T}^{(a)}; \mathcal{H}_T) \leq C_{\text{str}} m \log(pN_{\max})$

*Step 4: Bounding leaf parameter capacity.* The leaf term $I(M^{(a)}; \mathcal{H}_T \mid \mathcal{T}^{(a)})$ reduces to the capacity of a Gaussian linear model (the leaves are just constant regressors given the structure). This yields the standard $\log T$ determinant bound, i.e. $I(M^{(a)}; \mathcal{H}_T \mid \mathcal{T}^{(a)}) \leq \frac{1}{2} C_{\text{leaf}} m \log\big(1 + \frac{T}{4K\kappa^2\sigma^2}\big)$.

**Connection to frequentist minimax rates.** Theorem 1 is a Bayesian-design guarantee for Ideal BFTS. Under misspecification, however, standard Thompson sampling can under-explore and is difficult to analyze in nonparametric classes. For this reason, Appendix E studies a *feel-good variant* (Zhang, 2022) of BFTS that uses the same arm-wise BART posterior as BFTS, but applies a lightweight "feel-good" resampling step to encourage optimism (Algorithm 1v). This modification is a vanishing perturbation: setting $\lambda = 0$ recovers standard BFTS, and the theoretically optimal $\lambda^\star \asymp \epsilon_T \to 0$ (where $\epsilon_T$ is a valid BART contraction rate). Under $\alpha$-Hölder smoothness with intrinsic dimension $d$, the resulting frequentist regret is minimax-optimal up to logs (Rigollet & Zeevi, 2010), i.e. $\widetilde{\mathcal{O}}\big(K\, T^{(\alpha+d)/(2\alpha+d)}\big)$.

We emphasize that this does *not* constitute a frequentist regret bound for BFTS itself. Rather, it certifies that the same BART prior underlying BFTS supports minimax-rate learning once paired with the vanishing perturbation of Thompson Sampling. This "feel-good" modification has recently attracted attention in the literature, both theoretically (Li & Gu, 2025) and empirically (Anand & Liaw, 2025).

*Remark 1* (Why use arm-wise (separate) models?). The factorization in Lemma 2 is specific to our *arm-wise* modeling choice: each arm $a$ has its own BART parameter $\theta_a$ and likelihood depends on $\boldsymbol{\theta}^*$ only through the played arm. If instead one encodes the discrete action into the covariate vector (see Section 3.3) and fits a single BART model jointly across arms, then the posterior no longer factorizes

by arm, and the mutual-information decomposition used in Lemma 2 does not apply. The separate design is also required in our frequentist analysis.

The reliance on $K$ may seem sub-optimal. However, in a Bayesian setting, this only reflects the structural complexity of the product prior across arms. From a frequentist nonparametric perspective, encoding the action into the covariates increases the effective input dimension (e.g., from $d$ to $d + K$ under `one-hot`, or to $dK$ under `multi` if one ignores its block-sparsity), which worsens standard Hölder-type rates; thus the arm-wise bound can be more favorable when $K$ grows slowly with $T$, e.g. $K = o(T^{\alpha/(2\alpha+d)})$.

# 5. Numerical Experiments

Across synthetic and real-data benchmarks, we demonstrate that BFTS delivers significant gains when reward functions exhibit nonlinearity, sparsity, or heterogeneity. Diagnostics further confirm that our Bayesian approach yields calibrated uncertainty estimates, translating into efficient exploration.

## 5.1. Experimental Setup

We benchmark BFTS against standard contextual bandit baselines using *cumulative regret* over a horizon of $T = 10,000$ steps. Results reported in this section are across $R = 12$ independent replications, with all methods evaluated on identical interaction streams.

**Model specification.** We use a BART prior with an ensemble size of $m = 100$ trees per arm. This is reduced from the default choice of $m = 200$ (Chipman et al., 2010) to balance computational cost with ensemble diversity in the online setting, following Murray (2021). We use $n_{\text{post}} = 500$ posterior samples after $n_{\text{burn}} = 500$ burn-in steps, without thinning (supported by Table 7). For further efficiency, we employ a *batched posterior refresh schedule*: defining a refresh index $r(t) \stackrel{\text{def}}{=} \lceil 8 \log t \rceil$, we re-run MCMC at round $t$ only if $r(t) > r(t-1)$, consistent with Nilsson et al. (2024).

**Compute & parallelization.** NeuralTS (Zhang et al., 2021) runs on an NVIDIA A40 GPU (along with one full CPU socket, 36 cores). All other methods run on CPU (4 logical cores). BFTS utilizes 4 parallel MCMC chains. RFTS and XGBoostTS (i.e., TETS-RF and TETS-XGBoost in (Nilsson et al., 2024)) are parallelized using OpenMP with `nthread=4`.

**Context encoding.** We apply method-specific arm representations. For LinUCB (Li et al., 2010), LinTS, and NeuralTS, we use positional (`multi`) encoding, mapping $\mathbf{x} \in \mathbb{R}^d$ to $\mathbb{R}^{Kd}$ by placing $\mathbf{x}$ in the $k$-th block for arm $k$ and zero-padding elsewhere, suggested by Zhang et al. (2021). For XGBoostTS and RFTS, suggested by Nilsson et al. (2024), we use a one-hot arm feature by concatenating

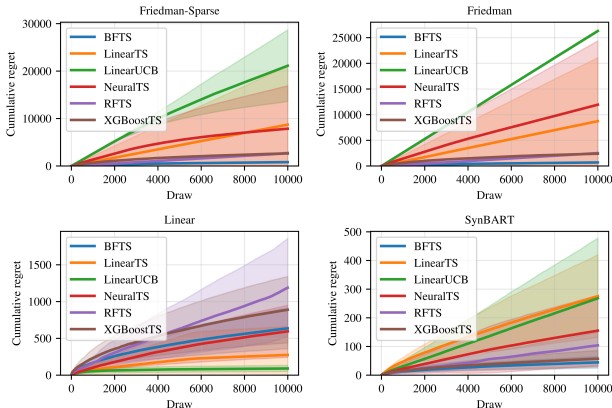

*Figure 1.* **Cumulative regret trajectories on synthetic benchmarks.** We report mean $\pm$ SD over $R = 12$ replications up to $T = 10,000$. Full synthetic benchmark regret curves are provided in Appendix (Figure 4).

$\mathbf{e}_k \in \{0, 1\}^K$ to the context, yielding $\tilde{\mathbf{x}}_k = (\mathbf{e}_k, \mathbf{x})$. BFTS uses separate-arm BART models.

## 5.2. Synthetic Study

We first evaluate performance across three synthetic Data-Generating Processes (DGPs): (i) linear rewards, (ii) nonlinear Friedman-type rewards with varying sparsity, correlation structures, underlying function, and heteroscedastic noise, and (iii) a SYNBART DGP in which each arm's reward function is sampled from the same BART prior family (and hyperparameters) used by BFTS, hence the Bayesian design of BFTS is correctly specified. Section A.1 provides complete definitions and parameter settings.

Figure 1 summarizes cumulative regret trajectories on representative synthetic scenarios; full trajectories across all synthetic scenarios are deferred to Appendix (Figure 4). The results align with theoretical expectations: across the eight synthetic scenarios, BFTS achieves the best average rank (1.63), outperforming the runner-up (NeuralTS, 3.00); see Table 2. In the Friedman scenarios, the gains are substantial and persist throughout the horizon. Conversely, in the Linear setting, linear baselines outperform BFTS, as they benefit from correct specification. In the nonlinear scenarios the comparison reverses for the same reason: the linear reward model cannot capture the underlying function.

**Uncertainty quantification.** BFTS demonstrates robust uncertainty quantification for the reward function values, which we examine, per round, by drawing random covariate vectors and calculating the proportion whose function values lie within their nominal 95% credible intervals. For example, on the Friedman Sparse Disjoint task, the mean credible interval coverage at $t = 10,000$ is 0.944 across replications. In contrast, XGBoostTS exhibits unstable cov-

*Table 1.* **Final cumulative regret on OpenML benchmarks.** We report mean $\pm$ SD over $R = 12$ replications; lower is better. Full results are shown in Table 3.

| Method | Adult | MagicTelescope | Mushroom | Shuttle |
|---|---|---|---|---|
| BFTS | **1538.7** $\pm$ **41.2** | **1488.2** $\pm$ **36.8** | **53.9** $\pm$ **7.6** | **106.9** $\pm$ **8.4** |
| XGBoostTS | 1557.0 $\pm$ 37.3 | 1588.5 $\pm$ 24.4 | 77.4 $\pm$ 10.0 | 170.4 $\pm$ 76.3 |
| RFTS | 1625.3 $\pm$ 41.8 | 1595.3 $\pm$ 50.3 | 69.1 $\pm$ 19.6 | 175.6 $\pm$ 52.9 |
| LinTS | 1756.4 $\pm$ 100.9 | 2086.6 $\pm$ 31.7 | 500.6 $\pm$ 135.1 | 699.2 $\pm$ 138.3 |
| LinUCB | 1771.7 $\pm$ 22.4 | 2111.8 $\pm$ 30.5 | 414.8 $\pm$ 49.0 | 766.9 $\pm$ 35.4 |
| NeuralTS | 3979.7 $\pm$ 1684.3 | 2005.2 $\pm$ 64.0 | 789.4 $\pm$ 1468.7 | 395.2 $\pm$ 281.3 |
| NeuralTS[†] | 2092.5 $\pm$ 48.0 | 2037.4 $\pm$ 61.3 | 115.0 $\pm$ 35.8 | 232.0 $\pm$ 149.5 |

[†] reported by Zhang et al. (2021).

erage, while other baselines significantly under-cover. In the linear setting, the coverage is also near nominal, though the interval width is wider than the linear baselines. Figure 2 visualizes the coverage–width trajectories: at each evaluation round $t$, we plot an agent's empirical coverage (y-axis) against its mean $95\%$ credible-interval length (x-axis) on a fixed probe set, and connect the points over $t$. In the model-matched SYNBART scenario, the empirical coverage of BFTS is slightly below nominal (Figure 5). Because BFTS draws from the *practical* posterior produced by finite batched MCMC rather than the exact posterior, and prior work on BART has documented that approximate posterior intervals can exhibit mild undercoverage when chains mix slowly (Ronen et al., 2022; Tan et al., 2024), we interpret this gap as evidence of practical posterior approximation error rather than a failure of the Bayesian model.

### 5.3. OpenML Benchmark

We transform standard OpenML (Vanschoren et al., 2014) classification tasks into contextual bandit problems following the protocol of Riquelme et al. (2018); Zhang et al. (2021); Nilsson et al. (2024): each class label becomes an arm, with binary reward 1 for the correct class and 0 otherwise. This introduces model misspecification, as the true Gaussian likelihood assumption of BFTS does not hold. We assess this misspecification empirically in Section A.4, where a logistic-BART variant of BFTS (Murray, 2021; Dumitrascu et al., 2018) neither improves nor degrades regret in our setting and runs about $1.6\times$ slower. We make use of the 6 datasets examined by Zhang et al. (2021) (and introduce 3 of our own), strictly follow their data preprocessing process, and present both the regret values calculated in their original paper and in our own reproduction of their results.

Table 1 reports performance on four representative datasets. We calculate cumulative regret up to $T = 10,000$ (except for Mushroom, which only has 8124 rows). BFTS achieves the lowest mean final regret across all tasks. On *Adult*, BFTS and XGBoostTS are numerically close, but on *MagicTelescope*, *Mushroom*, and *Shuttle*, BFTS shows larger improvements over baselines. Full results for five additional datasets are in Section A.2. While BFTS excels on tabular data, we note it is outperformed by NeuralTS on *MNIST*. This is expected: BFTS is designed for tabular heterogeneity, whereas NeuralTS leverages inductive biases suited for high-dimensional image data.

## 6. Case Study: The "Drink Less" mHealth Trial

To validate BFTS in a realistic intervention setting, we apply it to data from *Drink Less*, a behavior-change app designed for adults at risk of hazardous drinking.

### 6.1. Study Overview

The data comes from a micro-randomized trial (MRT) involving $n = 349$ participants over a 30-day period (Bell et al., 2020). The goal was to optimize the specific wording of push notifications to maximize user engagement. At 8:00 p.m. daily (local time), participants were randomized with static probabilities $(0.4, 0.3, 0.3)$ to one of three actions: (i) No message, (ii) standard message, and (iii) a varying message chosen from a message bank. The reward $R_{i,d} \in \{0, 1\}$ is defined as *proximal engagement*: whether the user $i$ opened the app within one hour (8–9 p.m.) following the decision point on that day $d$.

The MRT produces a panel dataset $\{(X_{i,d}, A_{i,d}, R_{i,d})\}$. To benchmark online bandit algorithms on this logged data, we construct a sequential-simulation testbed by unfolding the panel into a sequence of $T = n \times 30 = 10470$ decision points, while preserving within-participant temporal order $d$, to enable sequential simulation of adaptive algorithms that update after observing outcomes.

This environment presents a difficult contextual bandit challenge: the signal-to-noise ratio is low (engagement is sparse), and the reward function likely depends on complex interactions between static user traits (e.g., `AUDIT_score`, `age`) and dynamic states (e.g., `days_since_download`). The context vector $X_t$ includes these covariates along with one-hot user identifiers to capture unobserved heterogeneity. Further processing details are in Section A.8.

### 6.2. Off-Policy Evaluation (OPE)

Since we cannot deploy BFTS live in the past trial, we implement replay-style sequential simulation, where at each

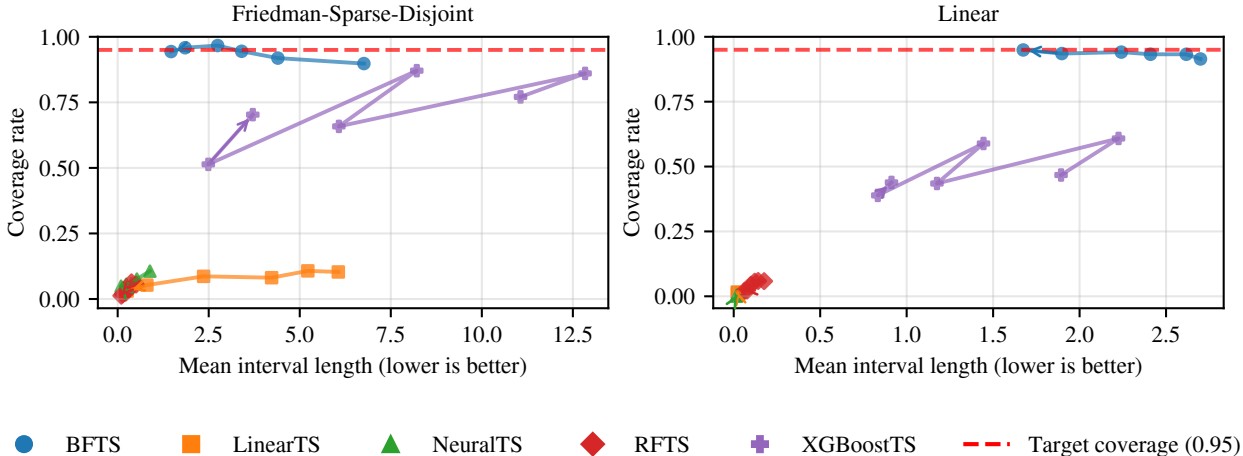

Figure 2. **Coverage vs. interval length on synthetic benchmarks.** We plot CI frontiers for representative nonlinear (Friedman) and linear scenarios. Each marker corresponds to an evaluation round $t \in \{200, 500, 1000, 2000, 5000, 10000\}$ and lines connect rounds in temporal order (arrows pointing to later rounds). The red dashed line indicates the nominal 0.95 coverage target (below: overconfident; above: conservative). Full uncertainty diagnostics are shown in Appendix (Figure 5).

step $t$, the algorithm is updated only if its chosen action matches the logged action $A_t$ in the historical dataset (Li et al., 2011; Liu et al., 2025).

For a target policy $\pi$, the (one-step) policy value is $V(\pi) = \mathbb{E}_{X \sim \mathcal{D}} \left[ \sum_{a \in \mathcal{A}} \pi(a \mid X) \mu(X, a) \right]$, where $\mu(X, a) = \mathbb{E}[R \mid X, a]$ is the conditional mean reward. We assess the quality of the learned policies using *self-normalized importance sampling (SNIPS)* (Swaminathan & Joachims, 2015). In the Appendix (Figure 12b), we also estimate the policy value with a doubly robust (DR) estimator (Dudik et al., 2011) to demonstrate that the estimated values are relatively insensitive to the choice of estimator. To ensure validity, we verify that importance weights remain bounded (see diagnostics in Section A.8).

**Results.** Figure 3 illustrates the estimated policy value over the learning horizon. BFTS achieves the highest estimated policy value consistently along the horizon, especially in the early stages.

- **Early Horizon** ($t = 1000$): BFTS improves the engagement probability by $+\mathbf{27.0}\%$ relative to the original random policy (0.0235 absolute gain) and by $+\mathbf{5.5}\%$ relative to the best baseline (LinUCB).

- **Final Horizon** ($t = 10{,}000$): BFTS outperforms the original policy by $\mathbf{38.2}\%$ (0.0345 absolute gain) and by a marginal $\mathbf{1.5}\%$ relative to the best baseline (LinTS tuned).

The consistent gap over linear baselines suggests that BFTS successfully captures non-linear interaction effects in user behavior that linear models miss. Full bootstrap distribu-

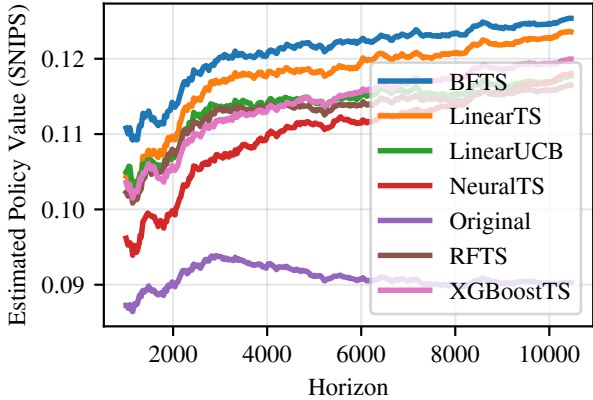

Figure 3. **Estimated policy value on the Drink Less mHealth trial data.** Policy value is estimated via SNIPS and is plotted as a function of horizon. BFTS achieves the highest estimated reward consistently along the horizon.

tions of the final policy gaps are reported in Appendix (Figure 11).

### 6.3. Policy Interpretability

Beyond performance, BFTS offers intrinsic interpretability. By analyzing the posterior split probabilities of the BART ensemble, we can identify which features drive decision-making. We find that the BFTS policy concentrates the vast majority of its splitting mass on three features: days_since_download (0.192), AUDIT_score (0.138), and age (0.128). This aligns with domain intuition: engagement naturally decays over time (days_since_download), and baseline risk

(AUDIT_score) moderates receptivity to intervention. Unlike black-box neural policies, BFTS provides this "variable importance" profile automatically, a critical feature for clinical safety and validation.

## 7. MCMC and Computational Considerations

BART posteriors are known to be multimodal and difficult to sample (Ronen et al., 2022; Tan et al., 2024). Our diagnostics (Table 6) confirm imperfect parameter-space mixing: at $t = 10,000$, median $\widehat{R}$ ranges from 1.19 to 1.90 across synthetic scenarios. Metropolis–Hastings acceptance rates are acceptable, between 0.10 and 0.36. That said, for bandit decision-making, exact parameter convergence is secondary to decision-level stability and posterior calibration. We measure the latter via *Policy $\Delta$-TV*, which tracks the total variation distance of the induced optimal-arm probability $\pi_t(\cdot \mid x)$ between MCMC snapshots (definition in Section A.6). By $t = 10,000$, Policy $\Delta$-TV is low (0.03–0.08; Table 6), indicating that the induced decision rule stabilizes even if specific tree parameters mix slowly. We emphasize that Policy $\Delta$-TV is a *decision-stability diagnostic*, not a formal regret bridge from Practical to Ideal BFTS. A rigorous bridge would decompose per-round excess regret as $\Delta_t(x) \cdot \mathrm{TV}(\widetilde{\pi}_t(\cdot \mid x), \pi_t^{\mathrm{ideal}}(\cdot \mid x))$, where $\Delta_t(x)$ is the instantaneous arm gap; closing this decomposition for nonparametric BART posteriors with batched MCMC remains open. Related approximation analyses in the parametric setting include Li et al. (2024) for tabular reinforcement learning and Li et al. (2025) for linear contextual bandits, which rely on closed-form perturbations of parametric models that do not directly apply to our nonparametric MCMC sampler. Furthermore, posterior calibration remains good: Expected Calibration Error (ECE) evolution and interval-coverage diagnostics are reported in Appendix (Figure 5). Consistent with this, our sensitivity sweeps show small returns from further increasing the MCMC budget (more chains/trees/burn-in; Section 8). Together, these results suggest that despite elevated $\widehat{R}$, the posterior approximation is already adequately reliable for posterior sampling in BFTS.

While slower than optimized linear methods and XG-BoostTS, BFTS is comparable in speed to RFTS and NeuralTS (GPU) (detailed timing results are in Section A.5), and it remains viable for behavioral interventions. At $T = 10,000$, cumulative runtime is $\approx 33$ minutes on Covertype and $\approx 44$ minutes on MNIST (Python implementation). This fits comfortably within the latency requirements of mHealth server-side logic. In addition, BFTS's re-fitting time grows slowly with more data, which, combined with the refresh schedule, yields sublinear cumulative time.

## 8. Sensitivity Analysis

We analyze the sensitivity of BFTS design choices on the four core OpenML datasets (*Adult*, *MagicTelescope*, *Mushroom*, and *Shuttle*), aggregated via normalized regret (final regret divided by the best final regret within each dataset; lower is better); full plots are in Section A.3.

**Encoding choice.** Using a single model with a one-hot arm feature substantially degrades performance (average normalized regret $3.15\times$) compared to our default separate-arm approach, while multi is close to separate (mean 1.051). This empirically supports our methodological argument in Section 3.3.

**Refresh schedule.** Across the refresh schedules we tested (logarithmic, square-root, and hybrid variants), performance differences are modest (mean normalized regret varies by only about 1.01–1.07; Appendix Figure 8), supporting our default logarithmic refresh as a robust operating point.

**Returns to increased MCMC budget.** Increasing chain count does not improve regret (e.g., 1.000 for 1 chain vs. 1.071 for 4 chains), and increasing the number of trees or burn-in iterations yields diminishing returns with only mild non-monotonicity (trees: 1.151 at $m = 50$, 1.058 at $m = 100$, 1.068 at $m = 150$; burn-in: 1.112 at 250 iterations, 1.086 at 500, 1.079 at 750). Detailed one-factor bars and interaction heatmaps are in Section A.3. Furthermore, swapping cold- for warm-start at nskip $\in \{250, 500\}$ yields no measurable change in regret (Table 7), so the default cold-start with nskip $= 500$ is robust along both axes.

**Prior hyperparameters.** Enabling the "quick-decay" prior of Ročková & Saha (2019) yields a modest improvement (mean 1.113 vs. 1.227 without), and a Dirichlet-sparse split prior also helps slightly (mean 1.100 vs. 1.154 without). The split-grid cap and tree-structure priors have moderate effects: varying max_bins yields mean normalized regret in the range 1.10–1.17, and overly small depth priors can be measurably harmful (e.g., $\alpha_{\mathrm{qd}} = 0.1$ yields 1.160 vs. about 1.08–1.10 for $\alpha_{\mathrm{qd}} \in \{0.3, 0.45\}$). Moderate leaf shrinkage tends to perform best (mean 1.07–1.17 across $\kappa \in \{1.5, 2.0, 2.5\}$); see Appendix (Figure 8) for one-factor summaries. Finally, we report interaction heatmaps for key pairs (trees $\times$ burn-in, $\kappa \times \alpha_{\mathrm{qd}}$, and Dirichlet prior $\times$ split grid) in Appendix (Figure 9) to confirm that the main effects above persist under joint perturbations. Overall, aside from the encoding choice, *BFTS is robust to reasonable hyperparameter and compute-budget variations*, which motivates our default implementation settings as a stable operating point for tabular bandit tasks.

## 9. Conclusion and Discussion

We presented *Bayesian Forest Thompson Sampling* (BFTS), a method that bridges the gap between the predictive power of tree ensembles and the principled exploration of Thompson Sampling. By leveraging the BART posterior, BFTS successfully adapts to the nonlinear, sparse, and heterogeneous reward landscapes typical of mobile health applications. We have validated the efficacy empirically on tabular benchmarks and the *Drink Less* micro-randomized trial and theoretically via a Bayesian regret analysis under ideal posterior sampling, as well as a complementary frequentist minimax bound for a "feel-good" variant.

Our work opens several avenues for investigation where theory meets practice:

**Computational fidelity.** We approximate the ideal posterior via batched MCMC. While our diagnostics show this is sufficient for effective decision-making, it introduces a theoretical gap. Future work could rigorously analyze how MCMC mixing rates interact with regret accumulation. Related work on approximate Thompson Sampling includes (Phan et al., 2019; Mazumdar et al., 2020; Osband et al., 2023), which highlight both the potential error and algorithmic strategies for reducing per-round inference cost. More broadly, approximate posterior sampling has also been implemented using sequential Monte Carlo methods (Bijl et al., 2017).

**Scalable inference.** While feasible for mHealth, standard MCMC is slow for high-frequency applications. Integrating advances in methodology or software for accelerated BART inference (Lakshminarayanan et al., 2015; He & Hahn, 2023; Herren et al., 2025; Petrillo, 2025) is an important direction.

**Information sharing and scaling in $K$.** Our current approach models arms independently, and the regret bound carries an explicit leading factor of $K$ that reflects this product-prior factorization (Lemma 2). The arm-wise design targets moderate-$K$ tabular settings typical of mHealth interventions: *Drink Less* has $K = 3$ and *DIAMANTE* (Aguilera et al., 2024) has $K = 9$. Consistent with this scoping, BFTS is competitive but does not dominate on the larger-$K$ tasks (Covertype and MNIST, both $K = 10$) where linear and neural baselines remain strong. Extending BFTS to a joint model with arm features, or to continuous action spaces along the lines of Boyne et al. (2025), requires both a different posterior factorization argument and additional optimization machinery to maximize a sampled tree ensemble over the action variable; we view this as important future work.

**Non-stationarity.** mHealth rewards often drift (e.g., user habituation). The Bayesian framework of BFTS is naturally extensible to this non-stationary setting. Such regimes, together with longer horizons, reduced forced exploration, and smaller inter-arm reward gaps, would also more clearly expose the value of posterior-based exploration relative to a greedy variant of BFTS.

**Extrapolation behavior.** Standard BART predicts a constant leaf value, with essentially unchanged posterior uncertainty, outside the convex hull of observed covariates. In online settings where the context distribution shifts or covers the domain only gradually, this can produce overconfident predictions in unexplored regions. Wang et al. (2024) mitigate this by grafting local Gaussian processes into BART leaves, at the cost of complicating the standard Bayesian interpretation and the theoretical guarantees we rely on. We did not observe a clear failure of this kind in our experiments, but flag it as a known limitation of standard BART in online deployment and as a natural direction for future work.

In summary, BFTS demonstrates that fully Bayesian nonparametrics are not just a theoretical curiosity but a practical alternative for contextual bandits on tabular data.

# Acknowledgements

RC gratefully acknowledges support from NSF–DMS 2515896. BC gratefully acknowledges support from MOE Tier 2 grant MOE-T2EP20122-0013. YT gratefully acknowledges support from NUS Start-up Grant A-8000448-00-00 and MOE AcRF Tier 1 Grant A-8002498-00-00. Additionally, BC and YT gratefully acknowledge joint support from MOE AcRF Tier 1 Grant A-8004458-00-00.

# Impact Statement

This work develops a contextual bandit method aimed at improving decision-making on tabular data, motivated in part by personalized mobile health interventions. If deployed responsibly, better-calibrated exploration can reduce trial-and-error costs and potentially improve user outcomes by adapting treatment decisions to heterogeneous populations.

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

# A. Experimental Details

All experiments are conducted using the same global seed 42 unless otherwise specified. Per-replication random seeds are spawned from this global seed and its specific indices.

## A.1. Synthetic Simulation

### A.1.1. SCENARIOS AND PARAMETERS.

All synthetic experiments use horizon $T = 10{,}000$ and $R = 12$ independent replications. The scenarios include Linear ($P = 10, K = 3$), Friedman variants (Friedman/Friedman2/Friedman3, $P = 5, K = 2$), sparse Friedman variants ($P = 20$ or 50, $K = 2$, with different correlation structures), and SYNBART ($P = 4, K = 3$) where arm reward functions are sampled from a BART prior.

### A.1.2. DATA-GENERATING PROCESSES (DGPS).

For all synthetic scenarios, the bandit interaction proceeds for $t = 1, \ldots, T$ with context $X_t \in \mathbb{R}^P$ and arms $a \in \{1, \ldots, K\}$. Given $X_t$, each arm has expected reward $\mu_a(X_t)$; if arm $a$ were pulled at time $t$, the realized reward is

$$Y_{t,a} = \mu_a(X_t) + \varepsilon_{t,a},$$

with independent Gaussian noise. Unless stated otherwise, $\varepsilon_{t,a} \sim \mathcal{N}(0, \sigma^2)$ for all arms.

**Linear.** Contexts are i.i.d. $X_t \sim \mathrm{Unif}([0,1]^P)$. For each arm $a$, coefficients $\beta_a \in \mathbb{R}^P$ are drawn with $\beta_{a,j} \overset{\text{iid}}{\sim} \mathcal{N}(0,1)$ for $j \leq d$ and $\beta_{a,j} = 0$ for $j > d$ (default $d = P$). The mean reward is

$$\mu_a(x) = \beta_a^\top x.$$

**Friedman (nonlinear, $K = 2$).** Contexts are i.i.d. $X_t \sim \mathrm{Unif}([0,1]^P)$. For FRIEDMAN we use the standard Friedman1 function

$$f(x_1, \ldots, x_5) = 10 \sin(\pi x_1 x_2) + 20(x_3 - 0.5)^2 + 10x_4 + 5x_5.$$

For FRIEDMAN2 and FRIEDMAN3, we rescale the first four coordinates:

$$x_1' = 100x_1, \quad x_2' = 40\pi + 520\pi x_2, \quad x_3' = x_3, \quad x_4' = 1 + 10x_4,$$

and define

$$f_2(x) = \frac{1}{125}\sqrt{(x_1')^2 + \left(x_2'x_3' - \frac{1}{x_2'x_4'}\right)^2}, \qquad f_3(x) = \frac{1}{0.1}\arctan\left(\frac{x_2'x_3' - \frac{1}{x_2'x_4'}}{x_1'}\right).$$

Arm 1 has mean $\mu_1(x) = f(x)$ where $f$ is the scenario-specific function ($f/f_2/f_3$). For arm 2 we consider:

- *Disjoint* (Friedman-Sparse-Disjoint): let $x^{\mathrm{rev}} = (x_P, x_{P-1}, \ldots, x_1)$ and set $\mu_2(x) = f(x^{\mathrm{rev}})$ (i.e., apply the same scenario-specific function to the reversed feature order).

- *Shared* (Friedman, Friedman2, Friedman3, Friedman-Sparse, Friedman-Heteroscedastic): $\mu_2(x) = f(x_1, \ldots, x_5) + 5\sin(\pi x_1 x_2)$.

The sparse variants differ only in $P$ (here, $P = 20$).

**Friedman-Heteroscedastic.** Same means as the shared Friedman variant above, but with arm-specific noise variances: for each arm $a$, sample $\sigma_a^2 = 10^U$ with $U \sim \mathrm{Unif}(-1, 1)$ once per replication, and draw $\varepsilon_{t,a} \sim \mathcal{N}(0, \sigma_a^2)$.

**SynBART.** Contexts are i.i.d. $X_t \sim \mathrm{Unif}([0,1]^P)$. For each arm $a$, we sample a mean-reward function $\mu_a(\cdot)$ from the same BART prior family (and hyperparameters) used by BFTS (Section B):

$$\mu_a(x) = \sum_{j=1}^m h\big(x; T_{a,j}, M_{a,j}\big),$$

with $m = 100$ trees, the depth-geometric splitting prior (P2-a) with $\alpha_{\mathrm{qd}} = 0.45$, Gaussian leaf prior (P3) with $\kappa = 2$, Dirichlet-sparse split-variable probabilities $s \sim \mathrm{Dir}(\zeta/p^\xi, \ldots, \zeta/p^\xi)$ with $(\zeta, \xi) = (1, 1)$, and split-grid cap $N_{\max} = 100$. Noise variance is fixed to $\sigma^2 = 0.01$.

*Table 2.* Simulation benchmarks: final cumulative regret at $T = 10{,}000$ (mean $\pm$ SD over $R = 12$ replications). Lower is better.

| Method | Avg. rank | Friedman (Sparse and Disjoint) | Friedman (Sparse) | Friedman (Heteroscedastic) | Friedman | Friedman2 | Friedman3 | Linear | SynBART |
|---|---|---|---|---|---|---|---|---|---|
| BFTS | 1.63 | **660.1** **± 51.8** | **814.2** **± 86.6** | **715.5** **± 135.5** | **687.8** **± 95.2** | *377.4* *± 84.1* | *592.3* *± 65.0* | 635.7 ± 277.0 | **44.5** **± 20.1** |
| LinTS | 4.38 | 8221.9 ± 3365.1 | 8721.9 ± 12334.8 | 19677.9 ± 11361.6 | 8734.7 ± 12353.1 | 605.6 ± 243.1 | 10892.6 ± 12888.5 | *274.8* *± 387.5* | 275.6 ± 143.9 |
| LinUCB | 5.00 | 10219.1 ± 5462.7 | 21133.0 ± 7556.2 | 26265.6 ± 137.1 | 26305.1 ± 200.9 | 777.7 ± 161.7 | 24048.4 ± 7252.0 | **90.6** **± 42.0** | 268.2 ± 209.5 |
| NeuralTS | 3.00 | *1444.7* *± 172.9* | 7858.9 ± 9031.1 | 16299.0 ± 11811.6 | 11942.3 ± 12426.1 | **134.6** **± 109.2** | **422.7** **± 655.1** | 595.3 ± 354.1 | 155.3 ± 122.6 |
| RFTS | 3.88 | 7625.3 ± 255.0 | 2737.0 ± 200.9 | 2601.3 ± 170.8 | 2512.9 ± 128.3 | 1084.3 ± 81.3 | 953.4 ± 134.5 | 1188.7 ± 666.2 | 104.0 ± 45.8 |
| XGBoostTS | 3.13 | 5010.1 ± 138.3 | *2634.2* *± 86.6* | *2417.8* *± 135.0* | *2397.0* *± 104.0* | 793.9 ± 47.6 | 1749.0 ± 68.3 | 888.9 ± 452.0 | *57.3* *± 28.6* |

**Posterior uncertainty diagnostics.** To complement regret comparisons, we summarize uncertainty calibration via CI frontiers and ECE evolution trends. To make these diagnostics comparable across time and across replications, we evaluate them on a fixed probe set $\mathcal{X}_{\text{probe}} = \{x^{(j)}\}_{j=1}^{J}$ with $J = 40$ contexts per scenario (fixed across replications via a scenario-seeded RNG). For synthetic scenarios, $x^{(j)}$ are sampled from the scenario's context generator; for OpenML scenarios, we use a deterministic subsample of $J = 40$ rows from the dataset. For each agent, bandit round $t$, probe point $x^{(j)}$, and arm $a$, we form a pointwise $95\%$ credible interval from posterior samples via the empirical quantiles $\left[q_{0.025}(x^{(j)}, a, t), q_{0.975}(x^{(j)}, a, t)\right]$. We then compute (i) for synthetic scenarios (where $f_0$ is known), a coverage indicator $\mathbf{1}\{f_0(x^{(j)}, a) \in [q_{0.025}, q_{0.975}]\}$, and (ii) an interval length $q_{0.975} - q_{0.025}$. Finally, at each $t$ we aggregate these quantities by an unweighted average over replications $s$, probes $j$, and arms $a$, yielding an empirical coverage rate and mean interval length for that agent at round $t$. The CI frontier plots the resulting trajectory of points (mean length$_t$, coverage$_t$) as $t$ increases.

**Feature inclusion diagnostics.** We report feature-inclusion frontiers for representative synthetic scenarios in Figure 6. The feature-inclusion frontier plots the cumulative splitting mass as features are ranked by posterior inclusion. A steep early rise indicates that a small subset of covariates drives a substantial fraction of the policy's decision rules for those sparse scenarios, indicating that BFTS is successfully capturing the sparse nature of the reward function.

### A.2. OpenML Benchmark

We evaluate on nine OpenML classification datasets: Adult, Covertype, EEGEyeState, GasDrift, MNIST, MagicTelescope, Mushroom, PageBlocks, and Shuttle. Each dataset induces a contextual bandit with contexts as feature vectors and arms as class labels; rewards are $0/1$ depending on whether the chosen arm matches the true class. We provide cumulative regret curves for all datasets in Figure 7. Baseline methods are set to the default values recommended in the original papers. For **LinTS** and **NeuralTS**, we use a two-stage grid tuning procedure: we screen the hyperparameter grid on runs with a different seed (seed 0) and then fix, for each dataset, the best setting (by mean final cumulative regret) for the full runs.

- **LinTS** grid: $\nu \in \{1.0, 0.1, 0.01\}$, $\lambda = 1$. (Zhang et al., 2021)

- **NeuralTS** grid: $\lambda \in \{1, 0.1, 0.01, 0.001\}$ and $\nu \in \{0.1, 0.01, 0.001, 0.0001, 0.00001\}$. (Zhang et al., 2021)

- **XGBoostTS** and **RFTS**: Fixed $\nu = 1$. (Nilsson et al., 2024)

*Remark* 2. Our evaluation selects hyperparameters for NeuralTS and LinTS via separate low-budget pilot runs (best in 4 runs for OPE, in 1 run for other datasets), rather than selecting the best *post hoc* configuration on the full experiment. While this may not capture the absolute peak performance of these methods, it represents a realistic and competitive tuning strategy for online learning, where extensive cross-validation is often computationally prohibitive or data-inefficient. In fact, providing any separate tuning budget already favors these baselines compared to a strictly online setting. Moreover, on tabular contextual-bandit benchmarks, Nilsson et al. (2024) similarly report that tree-ensemble bandits (e.g., XGBoost/RF-based TS variants) are among the strongest baselines and often outperform NeuralTS. We also report the NeuralTS numbers for the 4 core datasets from the original paper (**NeuralTS**[†] in Table 1). These results are not competitive with the tree-ensemble bandits or with our own results.

*Table 3.* OpenML benchmark: final cumulative regret at $T = 10,000$ (mean $\pm$ SD over $R = 12$ replications). Lower is better.

| Method | Avg. rank | Adult | MagicTelescope | Mushroom | Shuttle | Covertype | EEGEyeState | GasDrift | MNIST | PageBlocks |
|---|---|---|---|---|---|---|---|---|---|---|
| Context Dim ($P \times K$) | – | $15 \times 2$ | $12 \times 2$ | $23 \times 2$ | $10 \times 7$ | $9 \times 10$ | $15 \times 2$ | $129 \times 6$ | $785 \times 10$ | $11 \times 5$ |
| BFTS | 1.89 | **1538.7** $\pm$ **41.2** | **1488.2** $\pm$ **36.8** | **53.9** $\pm$ **7.6** | **106.9** $\pm$ **8.4** | 3734.3 $\pm$ 161.2 | *2273.5* $\pm$ *39.9* | *818.1* $\pm$ *75.5* | 4385.3 $\pm$ 490.6 | **266.3** $\pm$ **16.4** |
| LinTS | 3.89 | 1756.4 $\pm$ 100.9 | 2086.6 $\pm$ 31.7 | 500.6 $\pm$ 135.1 | 699.2 $\pm$ 138.3 | **3480.0** $\pm$ **79.9** | 3623.5 $\pm$ 54.7 | **481.2** $\pm$ **15.8** | 6757.5 $\pm$ 60.6 | 363.7 $\pm$ 21.6 |
| LinUCB | 4.78 | 1771.7 $\pm$ 22.4 | 2111.8 $\pm$ 30.5 | 414.8 $\pm$ 49.0 | 766.9 $\pm$ 35.4 | 3709.9 $\pm$ 46.8 | 3615.3 $\pm$ 31.1 | 922.6 $\pm$ 20.8 | 7391.7 $\pm$ 22.1 | 372.8 $\pm$ 15.6 |
| NeuralTS | 5.00 | 3979.7 $\pm$ 1684.3 | 2005.2 $\pm$ 64.0 | 789.4 $\pm$ 1468.7 | 395.2 $\pm$ 281.3 | 4695.5 $\pm$ 1009.3 | 4962.5 $\pm$ 36.8 | 7584.8 $\pm$ 245.0 | **1700.7** $\pm$ **81.2** | 421.9 $\pm$ 30.5 |
| RFTS | 3.11 | 1625.3 $\pm$ 41.8 | 1595.3 $\pm$ 50.3 | *69.1* $\pm$ *19.6* | 175.6 $\pm$ 52.9 | 3582.4 $\pm$ 66.4 | 2635.2 $\pm$ 86.9 | 2284.5 $\pm$ 443.7 | 5745.2 $\pm$ 399.9 | *292.8* $\pm$ *22.4* |
| XGBoostTS | 2.33 | *1557.0* $\pm$ *37.3* | *1588.5* $\pm$ *24.4* | 77.4 $\pm$ 10.0 | *170.4* $\pm$ *76.3* | *3541.3* $\pm$ *60.2* | **2132.3** $\pm$ **38.5** | 1240.8 $\pm$ 38.1 | *4080.5* $\pm$ *110.4* | 296.3 $\pm$ 14.0 |

*Table 4.* OpenML benchmark: pairwise win–tie–lose matrix across 9 datasets (cell format: W–T–L). Significant differences are determined via a Mann–Whitney test with Bonferroni correction ($\alpha = 0.1$).

| Agent | BFTS | LinTS | LinUCB | NeuralTS | RFTS | XGBoostTS | Total |
|---|---|---|---|---|---|---|---|
| BFTS | — | 7–0–2 | 8–1–0 | 8–0–1 | 6–2–1 | 5–2–2 | 34–5–6 |
| LinTS | 2–0–7 | — | 4–5–0 | 5–2–2 | 2–0–7 | 1–1–7 | 14–8–23 |
| LinUCB | 0–1–8 | 0–5–4 | — | 6–0–3 | 1–0–8 | 1–0–8 | 8–6–31 |
| NeuralTS | 1–0–8 | 2–2–5 | 3–0–6 | — | 1–2–6 | 1–2–6 | 8–6–31 |
| RFTS | 1–2–6 | 7–0–2 | 8–0–1 | 6–2–1 | — | 0–5–4 | 22–9–14 |
| XGBoostTS | 2–2–5 | 7–1–1 | 8–0–1 | 6–2–1 | 4–5–0 | — | 27–10–8 |

In Table 3, best and second-best methods (per dataset) are highlighted in bold and italics, respectively.

## A.3. Sensitivity Analysis

We report additional sensitivity plots for MCMC settings, tree priors, and encoding choices (all aggregated across the four core OpenML datasets used in the sensitivity experiment) using normalized regret (aggregated across datasets; lower is better). The sensitivity sweep uses a total of 16 runs (4 core OpenML datasets with 4 independent replications per dataset). We also include interaction heatmaps (Figure 9) to visualize joint effects. In this sweep, increasing chain count does not improve mean normalized regret (1.000 for 1 chain vs. 1.071 for 4 chains), and increasing MCMC budget yields modest returns (trees: 1.151 at $m = 50$ vs. 1.058 at $m = 100$ and 1.068 at $m = 150$; burn-in: 1.112 at `nskip`= 250 vs. 1.086 at `nskip`= 500 and 1.079 at `nskip`= 750). Although 1 chain is slightly better in this particular sweep, we use 4 chains as a default to reduce Monte Carlo variability and to support more stable posterior diagnostics; overall regret differences are modest. These results motivate our default configuration as a stable operating point.

## A.4. Likelihood Choice on Binary Rewards

The OpenML benchmark protocol introduces binary $\{0, 1\}$ rewards, so the Gaussian working likelihood used by BFTS is technically misspecified. To assess whether a probabilistically correct alternative would substantively change conclusions, we implemented a logistic-BART variant of BFTS that targets the same arm-wise BART posterior but under a logistic likelihood with the Pólya–Gamma augmentation MCMC of Murray (2021).

Table 5 compares the two variants on four binary-reward OpenML datasets. Neither dominates: the Gaussian variant wins on *MagicTelescope* and *Mushroom*, the logistic variant wins on *EEGEyeState*, and the two are within one standard deviation on *Adult*. The logistic variant runs roughly $1.6\times$ slower due to the latent-variable augmentation. We therefore use the Gaussian working likelihood as a robust default in the main experiments, consistent with the broader contextual bandit literature (Riquelme et al., 2018; Zhang et al., 2021; Nilsson et al., 2024); the logistic variant remains a viable option when calibrated binary probabilities are needed.

*Table 5.* Likelihood ablation on binary-reward OpenML datasets. Final regret at $T = 10{,}000$ (mean $\pm$ s.d. over 12 replications). Bold marks the lower-regret variant per row.

| Dataset | BFTS-Gaussian | BFTS-Logistic |
|---|---|---|
| Adult | $1551 \pm 40$ | $1529 \pm 29$ |
| MagicTelescope | $\mathbf{1487 \pm 31}$ | $1533 \pm 27$ |
| Mushroom | $\mathbf{53 \pm 6}$ | $94 \pm 6$ |
| EEGEyeState | $2244 \pm 43$ | $\mathbf{2183 \pm 25}$ |

## A.5. Runtime and Refresh Cost

In BFTS, per-round action selection is inexpensive given a posterior draw, while the main cost comes from periodic posterior refreshes that rerun MCMC from scratch according to the logarithmic schedule in Section 3.2. We summarize computational cost via cumulative wall-clock time curves in Figure 10.

## A.6. MCMC Mixing

We summarize standard MCMC diagnostics for BART posterior sampling. We report $\widehat{R}$ and Metropolis–Hastings acceptance rates as standard mixing diagnostics (Table 6). While $\widehat{R}$ values are elevated across scenarios, indicating imperfect chain convergence in the high-dimensional tree space, overall acceptance rates remain stable in a typical range (at $t = 10{,}000$, acceptance ranges from $0.10$ to $0.36$ across synthetic scenarios; Table 6). Despite this, the induced decision policies are robust: by $t = 10{,}000$, the policy $\Delta$-TV is small. Predictive uncertainty diagnostics (interval-coverage frontiers and ECE evolution) are reported in Figure 5.

**Policy $\Delta$-TV (decision-level stability).** We quantify how much the *induced decision rule* changes over time via a total-variation (TV) distance computed on a fixed set of probe contexts. We use the same probe set $\mathcal{X}_{\text{probe}}$ (with $J = 40$ per scenario) introduced above for the uncertainty diagnostics in Figure 5. At a set of diagnostic snapshot times $\mathcal{T}_{\text{snap}} \subset \{1, \ldots, T\}$ (stored as draw_idx), the agent produces an estimated optimal-arm probability vector on each probe,

$$\pi_t(a \mid x^{(j)}) \approx \Pr\left(a \in \arg\max_{a' \in \mathcal{A}} f_\theta(x^{(j)}, a') \,\big|\, \mathcal{D}_t\right), \qquad a \in \mathcal{A},$$

approximated by normalizing Monte Carlo "votes" over posterior draws at snapshot $t$. Let $t^- < t$ denote the previous snapshot time. For each probe $x^{(j)}$, define the snapshot-to-snapshot policy change as

$$\text{TV}_j(t) \stackrel{\text{def}}{=} \frac{1}{2} \sum_{a \in \mathcal{A}} \left| \pi_t(a \mid x^{(j)}) - \pi_{t^-}(a \mid x^{(j)}) \right|.$$

We then aggregate across probes to summarize decision-level stability at snapshot $t$ (e.g., median or mean over $j \leq J$); Table 6 reports the mean Policy $\Delta$-TV at $t \in \{500, 2000, 10000\}$.

**Cold-start vs. warm-start.** The refresh schedule of Section 3.2 reruns each MCMC chain from scratch at every refresh with a fixed burn-in. To check whether the cold-start default is wasteful and whether the chosen burn-in is sufficient, we ran a $2 \times 2$ factorial of (cold vs. warm initialization) $\times$ (nskip $\in \{250, 500\}$) on the four core OpenML datasets (Table 7). All four variants produce statistically indistinguishable regret: pairwise differences sit within one standard deviation on every dataset. Combined with the nskip sensitivity sweep in Figure 8, this is direct evidence that cold-start with nskip $= 500$ is a robust operating point: halving the burn-in does not degrade regret, and warm-starting the chain from the previous posterior draw confers no measurable additional benefit at this budget.

## A.7. Implementation Details

Unless otherwise specified in sensitivity studies, all experiments use the following default implementation settings. The BART prior and MCMC transition details follow Sections B and B.3.

- **Posterior update schedule**: We use an adaptive logarithmic refresh schedule where the posterior is re-sampled from scratch whenever $\lceil 8 \log t \rceil$ increases.

*Table 6.* MCMC and decision-level diagnostics for BFTS across synthetic scenarios. We report $\widehat{R}$ (median and mean at $t = 10{,}000$), the decision-level policy change between consecutive diagnostic snapshots (Policy $\Delta$-TV; mean at $t \in \{500, 2000, 10000\}$), and Metropolis–Hastings acceptance rates (overall; mean at $t \in \{500, 2000, 10000\}$).

| | $\widehat{R}$ ($t = 10$k) | | Policy $\Delta$-TV | | | Acceptance (overall) | | |
| --- | --- | --- | --- | --- | --- | --- | --- | --- |
| Scenario | Med. | Mean | $t = 500$ | $t = 2$k | $t = 10$k | $t = 500$ | $t = 2$k | $t = 10$k |
| Linear | 1.19 | 1.25 | 0.11 | 0.08 | 0.05 | 0.74 | 0.54 | 0.36 |
| SynBART | 1.31 | 1.40 | 0.08 | 0.04 | 0.03 | 0.61 | 0.36 | 0.20 |
| Friedman | 1.80 | 1.88 | 0.13 | 0.08 | 0.05 | 0.23 | 0.18 | 0.13 |
| Friedman2 | 1.41 | 1.54 | 0.09 | 0.04 | 0.03 | 0.46 | 0.38 | 0.27 |
| Friedman3 | 1.62 | 1.70 | 0.09 | 0.06 | 0.04 | 0.31 | 0.25 | 0.16 |
| Friedman-Sparse | 1.89 | 1.92 | 0.13 | 0.10 | 0.08 | 0.29 | 0.19 | 0.14 |
| Fried.-Heterosced. | 1.90 | 1.94 | 0.11 | 0.09 | 0.06 | 0.20 | 0.14 | 0.11 |
| Fried.-Sparse-Disj. | 1.86 | 1.90 | 0.14 | 0.06 | 0.03 | 0.26 | 0.19 | 0.10 |

*Table 7.* Cold-start vs. warm-start MCMC initialization with two burn-in budgets. Final regret at $T = 10{,}000$ (mean $\pm$ s.d. over 8 replications).

| Variant | Adult | MagicTelescope | Mushroom | Shuttle |
| --- | --- | --- | --- | --- |
| Cold-250 | $1559 \pm 38$ | $1485 \pm 30$ | $61 \pm 13$ | $120 \pm 13$ |
| Cold-500 | $1561 \pm 40$ | $1472 \pm 26$ | $58 \pm 10$ | $128 \pm 24$ |
| Warm-250 | $1556 \pm 31$ | $1468 \pm 38$ | $63 \pm 38$ | $116 \pm 18$ |
| Warm-500 | $1560 \pm 34$ | $1483 \pm 28$ | $63 \pm 24$ | $106 \pm 16$ |

- **MCMC settings**: For each refresh, we run 4 independent chains (parallelized via Ray actors). Each chain consists of 500 burn-in iterations (`nskip`) and at most 500 post-burn-in iterations (`ndpost`). Less post-burn-in iterations will actually be used if we do not need that much posterior draws. We do not use warm-starts; each refresh starts the MCMC from the prior.

- **Exploration**: The first 5 rounds for each arm are conducted using round-robin exploration ($\tau = 5$, for a total of $5K$ rounds) before switching to TS.

At each refresh, we rerun multi-chain BART MCMC from scratch and collect only post-burn-in draws (burn-in iterations are not saved). All post-burn-in draws from the 4 chains are pooled into a single draw set $\mathcal{S}_t = \{(\boldsymbol{\theta}^{(j)}, (\sigma^2)^{(j)})\}_{j=1}^{N_t}$, which is used until the next refresh. For standard BFTS, we randomly shuffle the indices $\{1, \dots, N_t\}$ to form a queue and consume indices sequentially; once exhausted, we reshuffle and repeat.

### A.8. DrinkLess Case Study

We also include diagnostics to assess OPE reliability in the DrinkLess case study. We use a user-level cluster bootstrap with $B = 30$ replicates and evaluate online-learning agents offline via sequential simulation with replay updates.

Heavy-tailed importance weights can lead to high-variance OPE estimates and unreliable policy comparisons. Here the weights remain moderate (shown on a log scale), providing evidence against severe variance blow-up due to positivity violations. Under the static behavior propensities $(0.4, 0.3, 0.3)$, weights are bounded by $1/\min_a \pi_0(a) = 3.\overline{3}$; empirically, for BFTS we have $\bar{w}_{\max} = 3.33$, $\bar{w}_{0.95} = 2.98$, and $\bar{w}_{0.99} = 3.33$ over $n = 10{,}470$ samples, with mean effective sample size ESS $\approx 4{,}944$ (median $4{,}972$, minimum $4{,}246$) across bootstrap replicates. Under offline replay, BFTS matches the logged action at a rate $0.306 \pm 0.005$ (mean $\pm$ SD across bootstrap replicates).

To compare methods to BFTS, we report the SNIPS value gap $\Delta = \widehat{V}_{\text{SNIPS}}(\pi) - \widehat{V}_{\text{SNIPS}}(\pi_{\text{BFTS}})$, where negative $\Delta$ indicates worse performance than BFTS. Figure 11a reports the SNIPS gap distributions, and Figure 11b reports the DR analogue; Figures 11c and 11d report the corresponding AUC-mean gap distributions. For DR, on each bootstrap replicate we fit a 2-fold cross-fitted per-arm ridge outcome model $\hat{q}(x, a)$, then estimate policy value as $\widehat{V}_{\text{DR}}(\pi_e) = \frac{1}{n} \sum_{i=1}^{n} \left[ \sum_a \pi_e(a \mid x_i)\hat{q}(x_i, a) + w_i(r_i - \hat{q}(x_i, a_i)) \right]$, with $w_i = \pi_e(a_i \mid x_i)/\pi_b(a_i \mid x_i)$. Uncertainty is quantified via user-level cluster

bootstrapping ($B = 30$ replicates).

## A.9. Reproducibility

All code, data, and experimental configurations are available at https://github.com/drizmiz/Bayesian-Forest-Thompson-Sampling-Supp. The implementation builds on a custom Python package (https://github.com/yanshuotan/bart-playground/tree/bandit-dev) that implements BART with bandit-specific extensions.

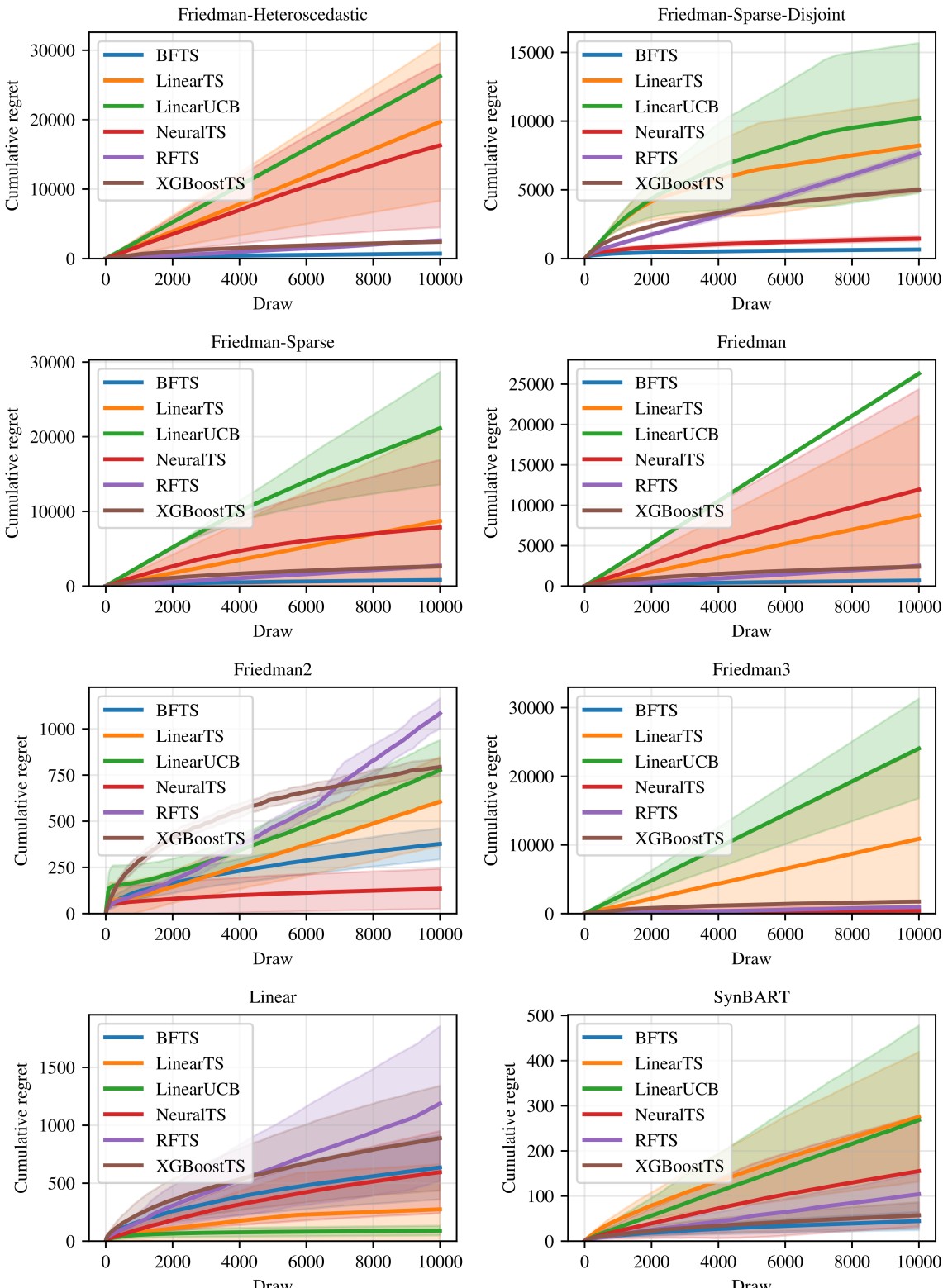

*Figure 4.* Simulation benchmarks: cumulative regret trajectories across all synthetic scenarios (mean $\pm$ SD over $R = 12$ replications) up to $T = 10,000$.

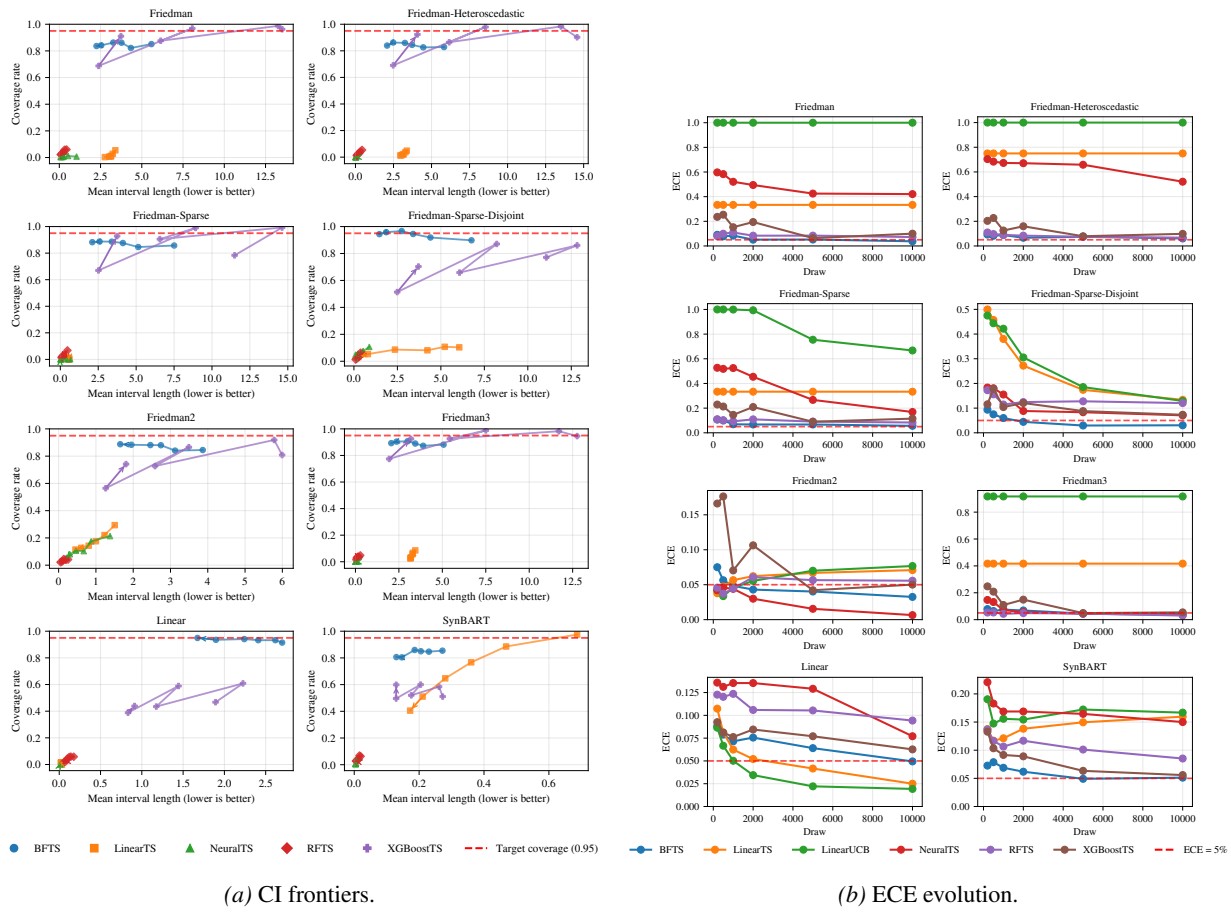

*(a)* CI frontiers.

*(b)* ECE evolution.

*Figure 5.* Simulation: posterior uncertainty diagnostics (coverage and calibration).

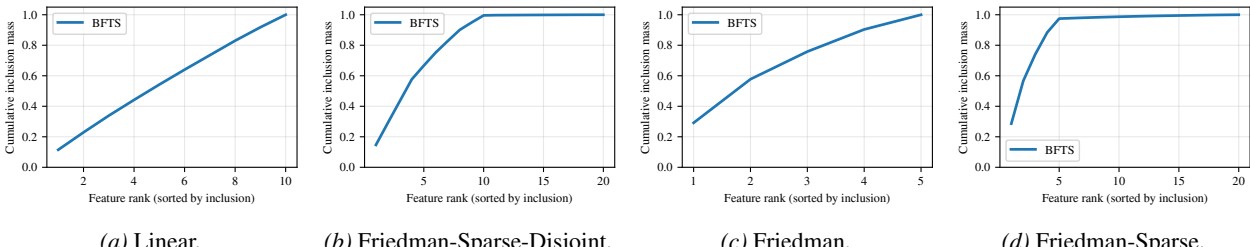

*(a)* Linear.  *(b)* Friedman-Sparse-Disjoint.  *(c)* Friedman.  *(d)* Friedman-Sparse.

*Figure 6.* Simulation: feature-inclusion frontiers for representative synthetic scenarios.

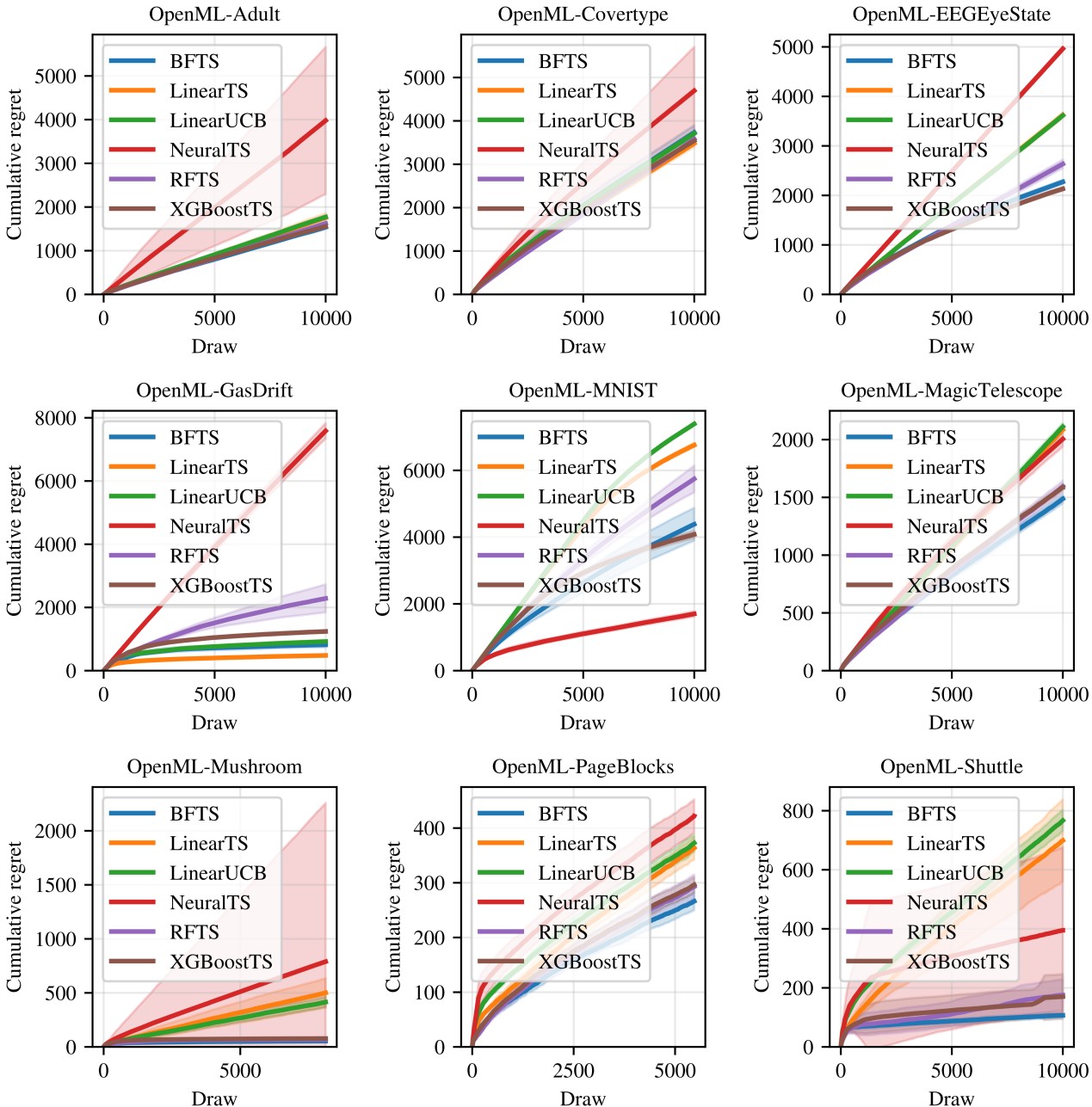

*Figure 7.* OpenML benchmark: cumulative regret trajectories on all nine datasets (mean $\pm$ SD over $R = 12$ replications).

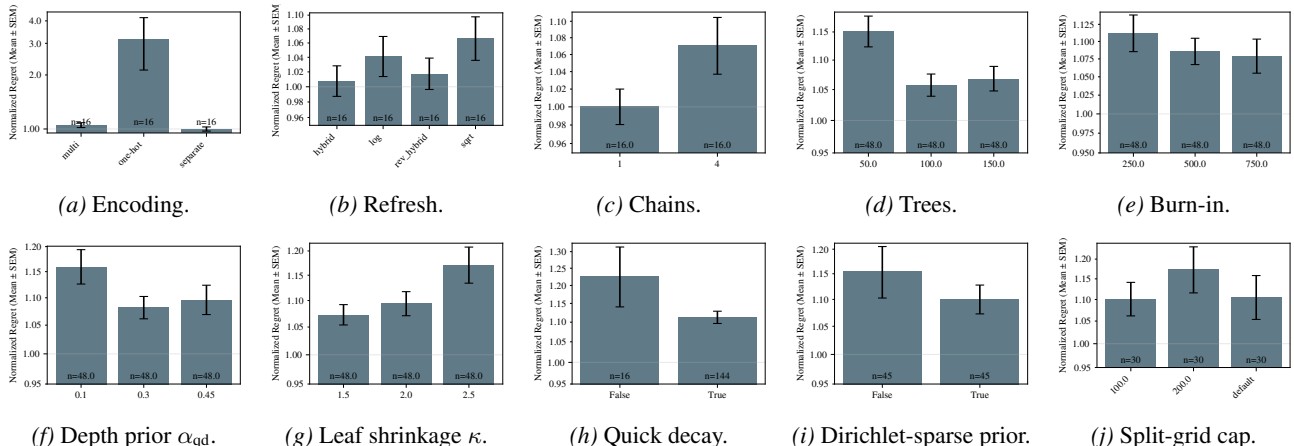

*(a)* Encoding.     *(b)* Refresh.     *(c)* Chains.     *(d)* Trees.     *(e)* Burn-in.

*(f)* Depth prior $\alpha_{\mathrm{qd}}$.     *(g)* Leaf shrinkage $\kappa$.     *(h)* Quick decay.     *(i)* Dirichlet-sparse prior.     *(j)* Split-grid cap.

*Figure 8.* Sensitivity analysis on the four core OpenML datasets (aggregated via normalized regret; lower is better).

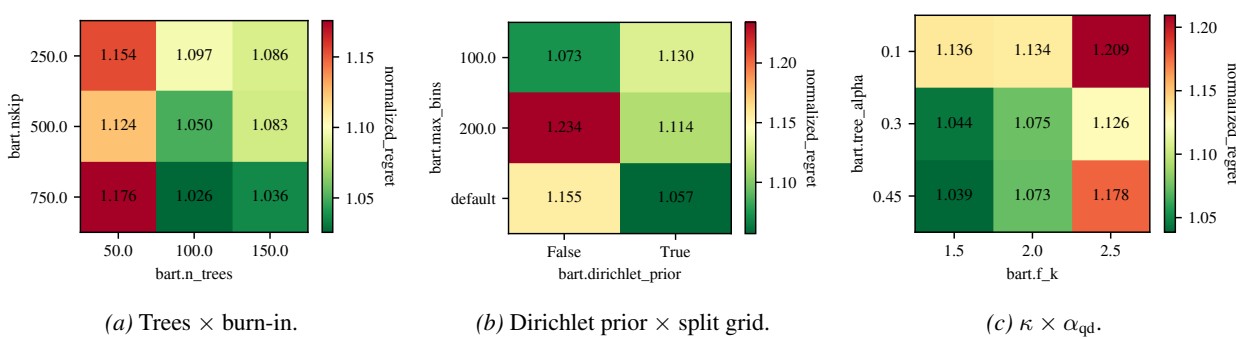

*(a)* Trees $\times$ burn-in.     *(b)* Dirichlet prior $\times$ split grid.     *(c)* $\kappa \times \alpha_{\mathrm{qd}}$.

*Figure 9.* Additional interaction heatmaps for sensitivity settings (aggregated via normalized regret; lower is better).

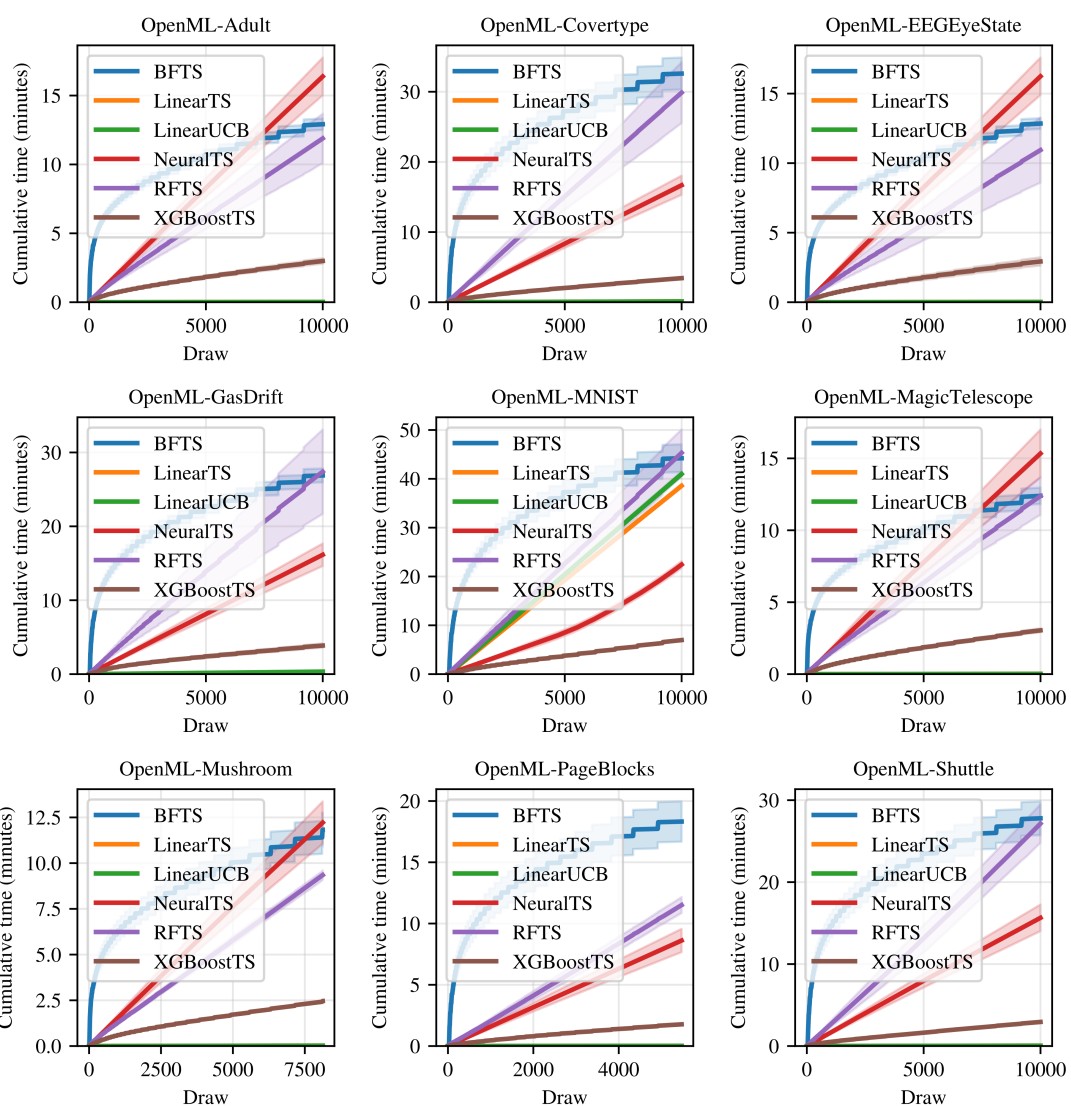

*Figure 10.* OpenML benchmark: cumulative wall-clock time trajectories (mean $\pm$ SD over $R = 12$ replications).

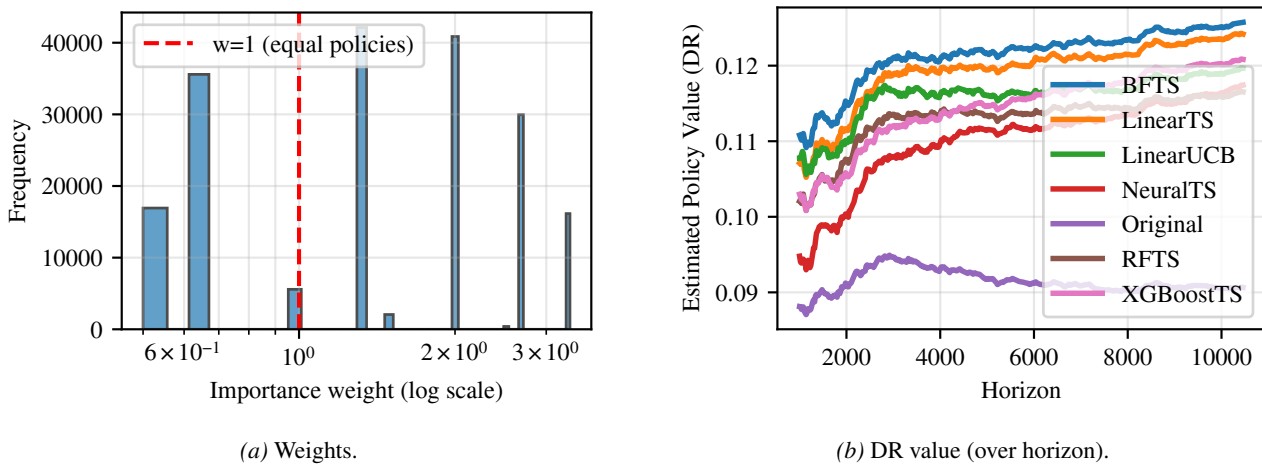

*(a)* SNIPS.

*(b)* DR.

*(c)* SNIPS (AUC mean).

*(d)* DR (AUC mean).

*Figure 11.* DrinkLess: bootstrap distribution of final policy value gaps relative to BFTS. Here AUC is the area under the policy-value curve (equivalently, the sum of policy-value gains along the horizon), and *AUC mean* is the average policy value gain.

*(a)* Weights.

*(b)* DR value (over horizon).

*Figure 12.* DrinkLess: OPE diagnostics. Left: importance-weight distribution for BFTS. Right: DR policy-value estimate over the horizon.

# B. Concrete BART Specification

## B.1. BART Model

We approximate the unknown regression function by a *sum of regression trees*. For observations $\{(x_i, Y_i)\}_{i=1}^n$, let $\{(x_i, Y_i)\}_{i=1}^n \subset [0,1]^p \times \mathbb{R}$ be i.i.d. observations from

$$Y_i = g(x_i) + \epsilon_i, \qquad \epsilon_i \stackrel{\text{iid}}{\sim} \mathcal{N}(0, \sigma^2),$$

where the ensemble function $g$ is a sum of $m$ tree contributions,

$$g(x) = \sum_{j=1}^m h(x; T_j, M_j),$$

and for tree $j$ with partition $T_j = \{\Omega_{j\ell}\}_{\ell=1}^{L_j}$ and leaf parameters $M_j = (M_{j1}, \ldots, M_{jL_j})$,

$$h(x; T_j, M_j) = \sum_{\ell=1}^{L_j} M_{j\ell} \, \mathbb{I}\{x \in \Omega_{j\ell}\}.$$

The priors and the MCMC algorithm used for posterior inference are detailed below. For the tree-splitting prior, we consider two alternatives: the original BART prior of Chipman et al. (2010) and the depth-geometric prior of Ročková & van der Pas (2020). In the empirical results of the main paper, we use the latter as a theory-guided default, but allow other choices in the sensitivity analysis; for proofs, we analyze the depth-geometric prior.

## B.2. Priors

**Error variance prior.** For our empirical implementation in the main paper, we follow the standard BART specification and place an inverse-$\chi^2$ prior on $\sigma^2$:

$$\sigma^2 \sim \nu \lambda_\sigma / \chi_\nu^2, \quad \text{equivalently,} \ \sigma^2 \sim \text{Inv-Gamma}(\nu/2, \nu\lambda_\sigma/2),$$

where $\chi_\nu^2$ denotes a chi-square distribution with $\nu$ degrees of freedom and $\lambda_\sigma$ is chosen so that $\Pr(\sigma < \hat{\sigma}) = q$ for a reference scale $\hat{\sigma}$ (e.g., residual s.d. from a linear model). Default from Chipman et al. (2010): $(\nu, q) = (3, 0.90)$.

**Hierarchical prior.** For a *fixed* ensemble size $m$, BART assumes independent priors across trees

$$\pi_{\text{BART}} = \prod_{j=1}^m \pi(T_j) \, \pi(M_j \mid T_j). \tag{P1}$$

Here, $\pi(T_j)$ is the prior on tree partitions, and $\pi(M_j \mid T_j)$ is the Gaussian leaf prior with variance $\sigma_\mu^2$, where $T_j$ is the partition of the $j$-th tree and $M_j = (M_{j1}, \ldots, M_{jL_j})$ is the vector of step heights.

**Prior on partitions $\pi(T_j)$.** Each tree is grown through a Galton–Watson branching process.

1. **Root.** Start with the single cell $[0,1]^p$.

2. **Splitting rule.** We do not impose a minimum leaf size (i.e., $n_{\min} = 1$). If a node can be split (i.e., after a split, the node's children are both non-empty), it is split with probability given by the following two rules.

   *Theory-guided rule (for proofs and default in empirics; Ročková & van der Pas (2020)).* A node $\Omega$ at depth $d(\Omega)$ is split with probability

   $$p_{\text{split}}(\Omega) = \alpha_{\text{qd}}^{d(\Omega)}, \qquad 0 < \alpha_{\text{qd}} < \tfrac{1}{2}, \tag{P2-a}$$

   which yields **q**uicker **d**ecay than that of the **o**riginal BART prior. We set $\alpha_{\text{qd}} = 0.45$ by default when this prior is used.

   *Original BART (Chipman et al., 2010).* We consider

   $$p_{\text{split}}(\Omega) = \alpha_{\text{o}} \left(1 + d(\Omega)\right)^{-\beta_{\text{o}}}, \qquad \alpha_{\text{o}} \in (0,1), \ \beta_{\text{o}} \geq 0, \tag{P2-b}$$

   with defaults $\alpha_{\text{o}} = 0.95$ and $\beta_{\text{o}} = 2$.

3. **Axis and cut-point.** Conditional on a split, the splitting coordinate $j$ is chosen from a probability vector $s = (s_1, \ldots, s_p)$. In our implementations, we consider two specifications:

- *Uniform splits (Chipman–George–McCulloch (Chipman et al., 2010)).*
  Set $s_j = 1/p$ for all $j$, so that each coordinate is equally likely a priori.
- *Dirichlet-sparse splits* (Linero, 2018; Jeong & Ročková, 2023).
  Draw $s \sim \text{Dir}(\zeta/p^\xi, \ldots, \zeta/p^\xi)$ with $\zeta > 0$ and $\xi \geq 1$, which encourages sparsity by concentrating mass on a small subset of coordinates. This prior enjoys strong guarantees in high dimensions and underpins our FGTS analysis in Section E. In our experiments, we use a Dirichlet split prior with $(\zeta, \xi) = (1, 1)$ as a practical default, while the theoretical results of Jeong & Ročková (2023) assume $\xi > 1$.

Given the splitting coordinate $j$, the cut-point $c$ is drawn uniformly from a per-feature empirical-quantile grid constructed from the observed covariate values $\{x_{ij}\}_{i=1}^n$. Concretely, we first choose $N_{\max}$ evenly spaced quantile levels $q \in [0, 1)$, compute the corresponding empirical quantiles of the column $(x_{1j}, \ldots, x_{nj})$, and then remove duplicates. As a result, the number of candidate thresholds along coordinate $j$ is at most $N_{\max}$. This empirical grid is the computational choice used in BART implementation for numerical stability (e.g., *stochtree*).

**Prior on step heights** $\pi(M_j \mid T_j)$. Let $h(x; T_j, M_j)$ denote the $j$-th tree's piecewise-constant contribution. Following Chipman et al. (2010), for each leaf $\ell$ of $T_j$ we place

$$M_{j\ell} \overset{\text{iid}}{\sim} \mathcal{N}\big(0, \sigma_\mu^2\big), \qquad \sigma_\mu = \frac{0.5}{\kappa \sqrt{m}}, \tag{P3}$$

where $\kappa > 0$ controls the overall shrinkage (default $\kappa = 2$) under the convention that $Y \in [-0.5, 0.5]$. This ensures each tree is a weak learner and the sum-of-trees remains suitably regular.

### B.3. Posterior Computation

Because the joint posterior over tree structures and leaf parameters is analytically intractable, BART uses a backfitting Gibbs sampler with Metropolis–Hastings (MH) updates for tree structures. One iteration consists of:

1. **Update trees (cycle $j = 1, \ldots, m$).** Form partial residuals

$$R_j = Y - \sum_{k \neq j} h(X; T_k, M_k).$$

Propose a local edit to $T_j$ and accept/reject via MH using the (marginal) Gaussian likelihood of $R_j$.

Standard proposals include: *GROW* (split a terminal node), *PRUNE* (collapse a pair of terminal siblings), *CHANGE* (modify a splitting rule), and *SWAP* (swap splitting rules between parent/child). After a structure move is accepted, draw leaf parameters $M_j$ from their conjugate Normal full conditional.

2. **Update error variance $\sigma^2$.** With the current sum-of-trees predictions $\hat{g}(X) = \sum_{j=1}^m h(X; T_j, M_j)$, sample $\sigma^2$ from its inverse-$\chi^2$ full conditional based on the residuals.

Repeat for $n_{\text{post}}$ iterations; discard the first $n_{\text{burn}}$ iterations as burn-in.

**MCMC details (structure MH + conjugate leaf updates).** For numerical stability, we follow Chipman et al. (2010) and apply the standard BART response rescaling internally within each arm-wise refresh before running the Gaussian BART sampler. Below we summarize the resulting MCMC updates for a single arm; see Chipman et al. (1998; 2010) for background on Bayesian tree MCMC.

*Backfitting residuals.* When updating tree $j$, treat the contributions of all other trees as fixed and define the partial residual vector $R_j = Y - \sum_{k \neq j} h(X; T_k, M_k)$, so that conditionally $R_j$ is modeled by a single regression tree with Gaussian noise.

*Tree structure MH step.* Let $T_j$ denote the current tree structure for tree $j$ and let $T_j'$ be a proposal generated by a local move (GROW/PRUNE/CHANGE/SWAP). We accept $T_j'$ with probability

$$\alpha(T_j \to T_j') = \min \left\{ 1, \ \frac{\pi(T_j')}{\pi(T_j)} \frac{p(R_j \mid T_j', \sigma^2, \sigma_\mu^2)}{p(R_j \mid T_j, \sigma^2, \sigma_\mu^2)} \frac{q(T_j \mid T_j')}{q(T_j' \mid T_j)} \right\},$$

where $\pi(\cdot)$ is the tree prior, $q(\cdot \mid \cdot)$ is the proposal distribution over tree structures, and $p(R_j \mid T_j, \sigma^2, \sigma_\mu^2)$ denotes the Gaussian *marginal* likelihood obtained by integrating out leaf values under the Normal prior $M_{j\ell} \sim N(0, \sigma_\mu^2)$. The four local moves are: GROW (split a terminal node), PRUNE (collapse a pair of terminal siblings), CHANGE (modify a split rule at an internal node), and SWAP (swap split rules between a parent and child). Proposals that create empty child nodes are rejected.

*Dirichlet-sparse split proportions (Gibbs update).* Under the Dirichlet-sparse splitting prior, the split-probability vector $\boldsymbol{s} = (s_1, \ldots, s_p)$ is assigned the prior $\boldsymbol{s} \sim \mathrm{Dir}(\zeta/p^\xi, \ldots, \zeta/p^\xi)$. Conditioning on the current forest structure $\{T_j\}_{j=1}^m$, let $c_u$ be the total number of internal nodes (across $j = 1, \ldots, m$) that split on coordinate $u \in \{1, \ldots, p\}$. Then the full conditional is conjugate:

$$\boldsymbol{s} \mid \{T_j\}_{j=1}^m \sim \mathrm{Dir}\big(\zeta/p^\xi + c_1, \ldots, \zeta/p^\xi + c_p\big).$$

We treat $\boldsymbol{s}$ as part of the MCMC state. Thus in the tree-structure MH step we evaluate the tree prior ratio conditional on the current $\boldsymbol{s}$ (i.e., $\pi(T_j' \mid \boldsymbol{s})/\pi(T_j \mid \boldsymbol{s})$).

*Leaf-value conjugate update.* Conditional on an accepted structure $T_j$, leaf values are updated from their conjugate Normal full conditional. Let $\Omega_{j\ell}$ be a leaf of $T_j$ and define $I_{j\ell} \stackrel{\text{def}}{=} \{i : x_i \in \Omega_{j\ell}\}$, $n_{j\ell} \stackrel{\text{def}}{=} |I_{j\ell}|$, and $S_{j\ell} \stackrel{\text{def}}{=} \sum_{i \in I_{j\ell}}(R_j)_i$. Then

$$M_{j\ell} \mid (T_j, R_j, \sigma^2) \sim N\left(\frac{S_{j\ell}}{n_{j\ell} + \sigma^2/\sigma_\mu^2}, \ \frac{\sigma^2}{n_{j\ell} + \sigma^2/\sigma_\mu^2}\right),$$

independently over leaves $\ell$ given $(T_j, R_j, \sigma^2)$.

*Error variance update.* With the current sum-of-trees fit $\hat{g}(X) = \sum_{j=1}^m h(X; T_j, M_j)$ and residual sum of squares $\mathrm{SSE} = \sum_{i=1}^n (Y_i - \hat{g}(x_i))^2$, the inverse-$\chi^2$ prior $\sigma^2 \sim \nu\lambda_\sigma/\chi_\nu^2$ implies the conjugate posterior

$$\sigma^2 \mid (Y, X, (T_j, M_j)_{j=1}^m) \sim \mathrm{Inv\text{-}Gamma}\left(\frac{\nu + n}{2}, \ \frac{\nu\lambda_\sigma + \mathrm{SSE}}{2}\right).$$

### B.4. Implementation Details and Defaults

**Preprocessing.**    As recommended by Chipman et al. (2010), we standardize the response so that $Y \in [-0.5, 0.5]$.

**Defaults.**    Unless otherwise stated, we use the following study-wide defaults:

$$\kappa = 2, \quad \alpha_{\mathrm{qd}} = 0.45 \text{ for (P2-a)}, \quad (\alpha_{\mathrm{o}}, \beta_{\mathrm{o}}) = (0.95, 2) \text{ for (P2-b)},$$
$$N_{\max} = 100, \quad (\nu, q) = (3, 0.90).$$

Proposal types are chosen with probabilities: GROW: 0.25, PRUNE: 0.25, CHANGE: 0.4, and SWAP: 0.1.

These defaults are chosen to be either consistent with the common practice, or theory-guided, or deemed not important during the sensitivity study. We do not tune these hyperparameters for specific datasets.

## C. Notation

For quick reference, we summarize the main symbols used throughout the paper.

| | |
|---|---|
| $\mathcal{X}$ | Context space, a subset of $\mathbb{R}^p$. |
| $X_t$ | Context observed at round $t \in \{1, \dots, T\}$. |
| $\mathcal{A}$ | Finite action set (arms), with cardinality $|\mathcal{A}| = K$. |
| $A_t$ | Action selected at round $t$. |
| $R_t$ | Reward at round $t$, modeled as $R_t = f_0(X_t, A_t) + \varepsilon_t$. |
| $f_0$ | True mean reward function $f_0 : \mathcal{X} \times \mathcal{A} \to \mathbb{R}$. |
| $\varepsilon_t$ | Mean-zero noise. |
| $\sigma_0^2$ | Data-generating noise variance proxy. |
| $\boldsymbol{\theta}$ | Model parameters $((\theta_a))_{a \in \mathcal{A}}$, with $\theta_a \in \Theta$. |
| $\mathcal{H}_t$ | History up to round $t$: $\mathcal{H}_t = \{(X_s, A_s, R_s)\}_{s=1}^t$. |
| $A^*(f, x)$ | An optimal arm for function $f$: $A^*(f, x) \in \arg\max_{a \in \mathcal{A}} f(x, a)$. |
| $f^*(f, x)$ | Optimal value for function $f$: $f^*(f, x) = \max_{a \in \mathcal{A}} f(x, a)$. |
| $A_t^*,\ f_0^*(x)$ | Shorthand: $A_t^* = A^*(f_0, X_t)$ and $f_0^*(x) = f^*(f_0, x)$. |
| $f_{\boldsymbol{\theta}}$ | Model for the mean reward, parameterized by $\boldsymbol{\theta}$. |
| $g_{\theta_a}$ | Arm-wise model: $f_{\boldsymbol{\theta}}(x, a) = g_{\theta_a}(x)$ with $(\theta_a)_{a \in \mathcal{A}}$ collecting the BART parameters across arms. |
| $\pi_{\text{BART}}$ | BART prior (see Section B), defined on $\Theta$. |
| $\Pi,\ \Pi(\cdot \mid \mathcal{H}_{t-1})$ | Prior on $\Theta$ and the posterior/pseudo-posterior given $\mathcal{H}_{t-1}$. For $K$ arms (Bayesian regret analysis), we use the product prior $\Pi = \pi_{\text{BART}}^{\otimes K}$. |
| $L_{\text{TS}}$ | Gaussian negative log-likelihood contribution used by TS (up to additive constants): $\frac{(f_{\boldsymbol{\theta}}(x,a) - r)^2}{2\sigma^2}$. When $\sigma^2$ is inferred, the additional $\frac{1}{2} \log \sigma^2$ term per observation is handled separately in the posterior via a factor of $(\sigma^2)^{-t/2}$. |
| $f_{\boldsymbol{\theta}}^*(x)$ | Model-implied optimal value: $\max_{a' \in \mathcal{A}} f_{\boldsymbol{\theta}}(x, a')$. |
| $\text{Regret}_t$ | Instantaneous regret at round $t$: $f_0^*(x_t) - f_0(x_t, A_t)$. |
| $\mathbb{E}[\cdot],\ \mathbb{E}_{\text{policy}}[\cdot]$ | Expectations: $\mathbb{E}[\cdot]$ averages over all randomness (e.g., contexts, prior, algorithm, noise) unless conditioned; $\mathbb{E}_{\text{policy}}[\cdot]$ is the conditional expectation given realized contexts $\{x_t\}_{t=1}^T$, taken over the TS algorithm's randomness, i.e. $\mathbb{E}_{\text{policy}}[\cdot] = \mathbb{E}[\cdot \mid \{x_t\}_{t=1}^T]$. |
| $\mathbb{I}(\cdot)$ | Indicator function: $\mathbb{I}(E) = 1$ if event $E$ holds, else 0. |

In Section D and Section E, we also use these symbols for the parameters in the BART prior.

For simplicity, we write $\sigma_0^2$ (homoscedastic noise); heteroscedastic/arm-wise noise extensions can be handled analogously.

Additional symbols referenced in the main paper's theoretical analysis and in Sections D and E:

- Lifted information ratio $\Gamma_t$ and conditional mutual information $I_t(\boldsymbol{\theta}^*; R_t \mid A_t)$; see formal definitions in Section D.

- Tree-related symbols: $\mathcal{T}$ (tree structures), $M$ (leaf parameters), $L$ (number of leaves); see Section B.

- Split grid cap $N_{\max}$ used in split selection; see Section B.

# D. Proof of the Bayesian Regret Bound for TS with Arm-Wise BART Prior

## D.1. Main Result

Under the arm-wise BART prior and the assumptions in the main paper, Thompson sampling satisfies

$$\mathbb{E}[\text{Regret}_T] \leq K \sqrt{C_{\text{rp}} \, T \log(eKT) \left( m C_{\text{str}} \log(p \, N_{\max}) \; + \; \tfrac{1}{2} C_{\text{leaf}} m \log\left(1 + \tfrac{1}{4\kappa^2 \sigma^2} \tfrac{T}{K}\right) \right)}$$

for a finite constant $C_{\text{rp}} > 0$ depending only on the fixed BART prior and split-grid constants, not on $K$ or $T$. In particular, when $N_{\max}$ is fixed or grows at most polynomially in $T$ (and $m$ is treated as a constant), the logarithmic factors can be absorbed into polylogarithmic terms, yielding

$$\mathbb{E}[\text{Regret}_T] = \widetilde{\mathcal{O}}\big(K\sqrt{T}\big).$$

## D.2. Proof of the Main Regret Bound

*Proof.* **Step 1 (Lifted information ratio and regret decomposition). Lifted information ratio.** Let $E_t[\cdot] = E[\cdot|\mathcal{H}_{t-1}, X_t]$ denote expectation conditioned on the history up to round $t-1$ and the realized context $X_t$. Let $\Pi_t(\cdot) = \Pi(\cdot|\mathcal{H}_{t-1})$ be the posterior distribution before observing $R_t$. The expected instantaneous regret at round $t$ is $E[r_t] = E[f_0(X_t, A_t^*) - f_0(X_t, A_t)]$. Using the tower property:

$$E[r_t] = E[E_t[f_0(X_t, A_t^*) - f_0(X_t, A_t)]]$$

The inner conditional expectation $E_t[r_t] = E_t[f_0(X_t, A_t^*) - f_0(X_t, A_t)]$ is the expected regret given the past and the current context.

Following Neu et al. (2022), we define the **lifted information ratio** for round $t$, given $\mathcal{H}_{t-1}$ and $X_t$, as:

$$\Gamma_t(X_t, \mathcal{H}_{t-1}) = \frac{(E_t[f_0(X_t, A_t^*) - f_0(X_t, A_t)])^2}{\bar{I}_t(\boldsymbol{\theta}^*; R_t)}.$$

Here, $\bar{I}_t(\boldsymbol{\theta}^*; R_t)$ is the *action-averaged* conditional information gain

$$\bar{I}_t(\boldsymbol{\theta}^*; R_t) \stackrel{\text{def}}{=} \mathbb{E}_t\left[D_{\mathrm{KL}}\left(P_{R_t|\boldsymbol{\theta}^*, \mathcal{H}_{t-1}, X_t, A_t} \| P_{R_t|\mathcal{H}_{t-1}, X_t, A_t}\right)\right],$$

where $D_{\mathrm{KL}}$ stands for the K-L divergence $D_{\mathrm{KL}}(P\|Q) = \int \log\left(\frac{\mathrm{d}P}{\mathrm{d}Q}\right)\mathrm{d}P$. Equivalently, letting $p_{t,a} = \Pr(A_t = a \mid \mathcal{H}_{t-1}, X_t)$,

$$\bar{I}_t(\boldsymbol{\theta}^*; R_t) = \sum_{a=1}^{K} p_{t,a} I(\boldsymbol{\theta}^*; R_t \mid \mathcal{H}_{t-1}, X_t, A_t = a).$$

**Regret decomposition.** Following the proof of Theorem 1 in Neu et al. (2022), note that

$$E_t[r_t] = \sqrt{\Gamma_t(X_t, \mathcal{H}_{t-1}) \cdot \bar{I}_t(\boldsymbol{\theta}^*; R_t)}.$$

Applying Cauchy-Schwarz to the expectation:

$$E[r_t] = E[E_t[r_t]] = E[\sqrt{\Gamma_t \cdot \bar{I}_t(\boldsymbol{\theta}^*; R_t)}] \leq \sqrt{E[\Gamma_t] \cdot E[\bar{I}_t(\boldsymbol{\theta}^*; R_t)]}.$$

Summing over $t$:

$$E[\overline{\mathrm{Regret}}_T] = \sum_{t=1}^{T} E[r_t] \leq \sum_{t=1}^{T} \sqrt{E[\Gamma_t]E[\bar{I}_t(\boldsymbol{\theta}^*; R_t)]}.$$

Let $\bar{\Gamma}_T = \frac{1}{T}\sum_{t=1}^{T} E[\Gamma_t]$. By Cauchy-Schwarz again on the sum:

$$E[\overline{\mathrm{Regret}}_T] \leq \sqrt{\sum_{t=1}^{T} E[\Gamma_t]\sum_{t=1}^{T} E[\bar{I}_t(\boldsymbol{\theta}^*; R_t)]} = \sqrt{T\bar{\Gamma}_T\left(\sum_{t=1}^{T} E[\bar{I}_t(\boldsymbol{\theta}^*; R_t)]\right)}$$

Gouverneur et al. (2023) uses the standard upper bound

$$\mathbb{E}\left[\sum_{t=1}^{T} \bar{I}_t(\boldsymbol{\theta}^*; R_t)\right] \leq I(\boldsymbol{\theta}^*; \mathcal{H}_T),$$

which follows from the mutual information chain rule and nonnegativity.

Combining the regret decomposition above with $\mathbb{E}[\sum_{t=1}^{T} \bar{I}_t(\boldsymbol{\theta}^*; R_t)] \leq I(\boldsymbol{\theta}^*; \mathcal{H}_T)$, we obtain

$$\mathbb{E}[\overline{\mathrm{Regret}}_T] \leq \sqrt{T\bar{\Gamma}_T \cdot I(\boldsymbol{\theta}^*; \mathcal{H}_T)}.$$

In Step 2 we bound $I(\boldsymbol{\theta}^*; \mathcal{H}_T)$ under the arm-wise BART prior.

**Step 2 (Bounding the mutual information).**

**Lemma 3** (Information gain under arm-wise BART prior). *Suppose the prior $\Pi$ on $\boldsymbol{\Theta} = (\Theta)^K$ factorizes across arms as*

$$\Pi = \bigotimes_{a \in \mathcal{A}} \pi_{BART},$$

*where each arm-wise prior $\pi_{BART}$ is a BART prior with $m$ trees, depth-decay splitting parameter $\alpha_{qd} \in (0, 1/2)$, split-grid cap $N_{max}$, and Gaussian leaf prior $N(0, \sigma_\mu^2)$ with $\sigma_\mu = 0.5/(\kappa\sqrt{m})$ for some $\kappa > 0$. Under the correctly specified Bayesian design, we draw $\theta_a^* \sim \pi_{BART}$ independently across arms and define $f_0(x, a) = g_{\theta_a^*}(x)$. Let rewards satisfy*

$$R_t = f_0(X_t, A_t) + \varepsilon_t, \qquad t = 1, \ldots, T,$$

*with $\{\varepsilon_t\}$ conditionally Gaussian given $(\boldsymbol{\theta}^*, \mathcal{H}_{t-1}, X_t, A_t)$, namely $\varepsilon_t \mid (\boldsymbol{\theta}^*, \mathcal{H}_{t-1}, X_t, A_t) \sim N(0, \sigma^2)$ for a fixed known constant $\sigma^2 > 0$. Then there exist constants $C_{str}, C_{leaf} > 0$, depending only on $(\alpha_{qd}, \kappa, \sigma^2)$, such that*

$$I(\boldsymbol{\theta}^*; \mathcal{H}_T) \leq Km C_{str} \log(p\, N_{max}) + \tfrac{1}{2} K C_{leaf} m \log\big(1 + \tfrac{1}{4\kappa^2\sigma^2} \tfrac{T}{K}\big)$$

*Proof.* See Section D.3.2.

**Step 3 (Bounding the lifted information ratio).** Marginalizing over BART tree structures yields a finite Gaussian mixture, not a single Gaussian predictive law with variance $\sigma^2$. We therefore control the lifted information ratio through the random predictive proxy in Lemma 8, which gives

$$\bar{\Gamma}_T \leq C_{rp} K \log(eKT).$$

**Step 4 (Combining bounds).** Plugging the bounds from Step 2 and Step 3 into the inequality from Step 1, and using Lemma 3, we obtain

$$\mathbb{E}[\text{Regret}_T] \leq K \sqrt{C_{rp}\, T \log(eKT) \Big(m C_{str} \log(p\, N_{max}) + \tfrac{1}{2} C_{leaf} m \log\big(1 + \tfrac{1}{4\kappa^2\sigma^2} \tfrac{T}{K}\big)\Big)}.$$

When $N_{max}$ is fixed or grows at most polynomially in $T$, the logarithmic factors can be absorbed into $\log(pT)$, yielding

$$\mathbb{E}[\text{Regret}_T] = \widetilde{\mathcal{O}}\big(K\sqrt{T}\big),$$

which completes the proof. $\square$

### D.3. Proofs of Auxiliary Lemmas

D.3.1. CORRECTED LIFTED INFORMATION RATIO BOUND

We replace the fixed-$\sigma^2$ sub-Gaussian claim by a random predictive proxy. Let

$$\mathcal{F}_t \stackrel{\text{def}}{=} \sigma(\mathcal{H}_{t-1}, X_t).$$

For each arm $a \in [K]$, define the counterfactual reward

$$R_{t,a} \stackrel{\text{def}}{=} f_0(X_t, a) + \varepsilon_{t,a},$$

where $\varepsilon_{t,a} \sim N(0, \sigma^2)$ are conditionally independent copies of the observation noise. The realized reward is $R_t = R_{t,A_t}$.

**Lemma 4** (Random-proxy lifted information ratio). *Suppose that for each round $t$ there exists a nonnegative $\mathcal{F}_t$-measurable random variable $v_t$ such that, for every arm $a \in [K]$,*

$$R_{t,a} - \mathbb{E}_t[R_{t,a}] \quad \text{is conditionally $v_t$-sub-Gaussian given $\mathcal{F}_t$,}$$

*that is,*

$$\log \mathbb{E}_t\big[\exp\{\lambda\big(R_{t,a} - \mathbb{E}_t[R_{t,a}]\big)\}\big] \leq \frac{\lambda^2 v_t}{2} \qquad \forall \lambda \in \mathbb{R}.$$

*Then*

$$\Gamma_t(X_t, \mathcal{H}_{t-1}) \leq 2K v_t \qquad a.s.$$

*Consequently,*

$$\bar{\Gamma}_T \leq \frac{2K}{T} \sum_{t=1}^{T} \mathbb{E}[v_t].$$

*Proof.* Let
$$p_{t,a} \stackrel{\text{def}}{=} \Pr(A_t = a \mid \mathcal{F}_t) = \Pr(A_t^* = a \mid \mathcal{F}_t),$$
where the second equality follows from posterior matching of Thompson sampling. For each arm $a$, define
$$Q_{t,a} \stackrel{\text{def}}{=} \mathcal{L}(R_{t,a} \mid \mathcal{F}_t), \qquad P_{t,a} \stackrel{\text{def}}{=} \mathcal{L}(R_{t,a} \mid \mathcal{F}_t, A_t^* = a),$$
and
$$\delta_{t,a} \stackrel{\text{def}}{=} \mathbb{E}_t[R_{t,a} \mid A_t^* = a] - \mathbb{E}_t[R_{t,a}].$$

Since $A_t$ and $A_t^*$ are conditionally i.i.d. under $\mathcal{F}_t$ with common law $p_{t,\cdot}$, the conditional expected regret is
$$g_t \stackrel{\text{def}}{=} \mathbb{E}_t[f_0(X_t, A_t^*) - f_0(X_t, A_t)] = \sum_{a=1}^{K} p_{t,a} \delta_{t,a}.$$

Fix $a$. By the variational form of KL divergence, for any $\lambda > 0$,
$$D_{\text{KL}}(P_{t,a} \| Q_{t,a}) \geq \lambda \delta_{t,a} - \log \mathbb{E}_t \left[ \exp\{\lambda(R_{t,a} - \mathbb{E}_t[R_{t,a}])\} \right].$$

Using the conditional $v_t$-sub-Gaussian assumption gives
$$D_{\text{KL}}(P_{t,a} \| Q_{t,a}) \geq \lambda \delta_{t,a} - \frac{\lambda^2 v_t}{2}.$$

Applying the same bound to $-R_{t,a}$ yields
$$|\delta_{t,a}| \leq \sqrt{2 v_t \, D_{\text{KL}}(P_{t,a} \| Q_{t,a})}.$$

Therefore,
$$g_t^2 = \left( \sum_{a=1}^{K} p_{t,a} \delta_{t,a} \right)^2 \leq K \sum_{a=1}^{K} p_{t,a}^2 \delta_{t,a}^2 \leq 2K v_t \sum_{a=1}^{K} p_{t,a}^2 \, D_{\text{KL}}(P_{t,a} \| Q_{t,a}).$$

Since all KL terms are nonnegative,
$$\sum_{a=1}^{K} p_{t,a}^2 \, D_{\text{KL}}(P_{t,a} \| Q_{t,a}) \leq I_t(A_t^*; R_t \mid A_t).$$

By data processing,
$$I_t(A_t^*; R_t \mid A_t) \leq \bar{I}_t(\boldsymbol{\theta}^*; R_t).$$

Thus
$$g_t^2 \leq 2K v_t \, \bar{I}_t(\boldsymbol{\theta}^*; R_t),$$

which proves
$$\Gamma_t(X_t, \mathcal{H}_{t-1}) = \frac{g_t^2}{\bar{I}_t(\boldsymbol{\theta}^*; R_t)} \leq 2K v_t.$$

Taking expectations and averaging over $t$ yields the second claim. $\qquad\square$

**Lemma 5** (Gaussian-BART predictive random proxy)**.** *Under the assumptions of Lemma 3, set*
$$s_0^2 \stackrel{\text{def}}{=} m \sigma_\mu^2 = \frac{1}{4\kappa^2}.$$

*Let $\mathfrak{T}$ denote the finite set of admissible arm-wise forest structures induced by the fixed finite split grid and non-redundant cell-refining splits. For $\tau \in \mathfrak{T}$, let*
$$\mu_{\tau,t,a} \stackrel{\text{def}}{=} \mathbb{E}[g_{\theta_a}(X_t) \mid \mathcal{F}_t, \tau]$$

*be the fixed-structure posterior predictive mean, and define*

$$D_{t,a} \overset{\text{def}}{=} \max_{\tau \in \mathfrak{T}} \mu_{\tau,t,a} - \min_{\tau \in \mathfrak{T}} \mu_{\tau,t,a}.$$

*Set*

$$v_t \overset{\text{def}}{=} \sigma^2 + s_0^2 + \frac{1}{4} \max_{a \in [K]} D_{t,a}^2.$$

*Then for every arm $a \in [K]$,*

$$R_{t,a} - \mathbb{E}_t[R_{t,a}] \quad \text{is conditionally } v_t\text{-sub-Gaussian given } \mathcal{F}_t.$$

*Proof.* Fix $t$ and $a$. Conditional on a fixed structure $\tau \in \mathfrak{T}$, the arm-wise BART model reduces to Bayesian linear regression on leaf parameters. Let $\Phi_{\tau,t,a}$ be the arm-$a$ leaf-incidence design matrix built from the observations in $\mathcal{H}_{t-1}$ with $A_s = a$, and let $c_{\tau,t,a}$ be the current query leaf-indicator vector for $X_t$. Write

$$\lambda \overset{\text{def}}{=} \frac{\sigma^2}{\sigma_\mu^2}.$$

Then

$$\mu_{\tau,t,a} = c_{\tau,t,a}^\top \big(\Phi_{\tau,t,a}^\top \Phi_{\tau,t,a} + \lambda I\big)^{-1} \Phi_{\tau,t,a}^\top R^{(a)},$$

and the posterior covariance of the leaf parameters is

$$\Sigma_{\tau,t,a} = \sigma^2 \big(\Phi_{\tau,t,a}^\top \Phi_{\tau,t,a} + \lambda I\big)^{-1}.$$

The latent predictive variance satisfies

$$s_{\tau,t,a}^2 \overset{\text{def}}{=} c_{\tau,t,a}^\top \Sigma_{\tau,t,a} c_{\tau,t,a} \le \sigma^2 \cdot \frac{\sigma_\mu^2}{\sigma^2} \|c_{\tau,t,a}\|_2^2 = m\sigma_\mu^2 = s_0^2,$$

because

$$\big(\Phi_{\tau,t,a}^\top \Phi_{\tau,t,a} + \lambda I\big)^{-1} \preceq \lambda^{-1} I$$

and $c_{\tau,t,a}$ has exactly $m$ entries equal to one. Thus

$$R_{t,a} \mid (\mathcal{F}_t, \tau) \sim N\big(\mu_{\tau,t,a}, \sigma^2 + s_{\tau,t,a}^2\big), \qquad \sigma^2 + s_{\tau,t,a}^2 \le \sigma^2 + s_0^2.$$

Let $\pi_{t,a}(\tau) = \Pr(\tau \mid \mathcal{F}_t)$, and define

$$\bar{\mu}_{t,a} \overset{\text{def}}{=} \sum_{\tau \in \mathfrak{T}} \pi_{t,a}(\tau) \mu_{\tau,t,a}.$$

Marginalizing over $\tau$, the predictive law is a finite Gaussian mixture. Therefore, for any $\lambda \in \mathbb{R}$,

$$\mathbb{E}_t\Big[e^{\lambda(R_{t,a} - \mathbb{E}_t[R_{t,a}])}\Big] = \sum_{\tau \in \mathfrak{T}} \pi_{t,a}(\tau) \exp\left(\lambda(\mu_{\tau,t,a} - \bar{\mu}_{t,a}) + \frac{\lambda^2(\sigma^2 + s_{\tau,t,a}^2)}{2}\right)$$

$$\le \exp\left(\frac{\lambda^2(\sigma^2 + s_0^2)}{2}\right) \sum_{\tau \in \mathfrak{T}} \pi_{t,a}(\tau) \exp(\lambda(\mu_{\tau,t,a} - \bar{\mu}_{t,a})).$$

The discrete random variable $\mu_{\tau,t,a}$ takes values in an interval of width $D_{t,a}$, so Hoeffding's lemma gives

$$\sum_{\tau \in \mathfrak{T}} \pi_{t,a}(\tau) \exp(\lambda(\mu_{\tau,t,a} - \bar{\mu}_{t,a})) \le \exp\left(\frac{\lambda^2 D_{t,a}^2}{8}\right).$$

Hence

$$\mathbb{E}_t\Big[e^{\lambda(R_{t,a} - \mathbb{E}_t[R_{t,a}])}\Big] \le \exp\left[\frac{\lambda^2}{2}\left(\sigma^2 + s_0^2 + \frac{D_{t,a}^2}{4}\right)\right] \le \exp\left(\frac{\lambda^2 v_t}{2}\right).$$

$\square$

**Lemma 6** (Uniform $\ell_1$-stability of fixed-structure predictive means). *For every admissible structure $\tau \in \mathfrak{T}$, there exists a finite constant $L_\tau > 0$, depending only on $(\tau, \kappa, \sigma^2)$, such that for every arm $a$ and round $t$,*

$$|\mu_{\tau,t,a}| \le L_\tau M_{t-1}, \qquad M_0 \overset{\text{def}}{=} 0, \quad M_{t-1} \overset{\text{def}}{=} \max_{1 \le s \le t-1} |R_s| \quad (t \ge 2).$$

*Consequently, with*

$$L_* \overset{\text{def}}{=} \max_{\tau \in \mathfrak{T}} L_\tau < \infty,$$

*we have*

$$D_{t,a} \le 2L_* M_{t-1} \qquad \text{for all } a, t.$$

*Proof.* Fix $\tau \in \mathfrak{T}$. Since the split-grid cap is finite and admissible splits are non-redundant refinements, only finitely many leaf-incidence row patterns can appear under $\tau$. Let these patterns be

$$\varphi_1, \ldots, \varphi_{S_\tau} \in \{0,1\}^{p_\tau},$$

where $p_\tau$ is the total number of leaves in the forest $\tau$. Let $U_\tau \in \mathbb{R}^{S_\tau \times p_\tau}$ be the matrix whose $r$-th row is $\varphi_r^\top$, and let

$$\mathcal{C}_\tau \overset{\text{def}}{=} \{\varphi_1, \ldots, \varphi_{S_\tau}\}$$

be the finite set of possible query leaf-incidence vectors.

Fix $a, t$. If $t = 1$, no past rewards have been observed, so $\mu_{\tau,t,a} = 0$ and the claim is immediate. Assume $t \ge 2$. Let $n_r$ be the number of arm-$a$ observations in $\mathcal{H}_{t-1}$ whose row pattern equals $\varphi_r$, and let $N \overset{\text{def}}{=} \mathrm{diag}(n_1, \ldots, n_{S_\tau})$. Let $\bar{Y}_r$ be the sum of the corresponding rewards in pattern $r$, so that $|\bar{Y}_r| \le n_r M_{t-1}$. Then

$$\mu_{\tau,t,a} = c_{\tau,t,a}^\top \left(U_\tau^\top N U_\tau + \lambda I\right)^{-1} U_\tau^\top \bar{Y}, \qquad \lambda = \frac{\sigma^2}{\sigma_\mu^2}.$$

Thus

$$|\mu_{\tau,t,a}| \le M_{t-1} \sum_{r=1}^{S_\tau} n_r \left| c_{\tau,t,a}^\top \left(U_\tau^\top N U_\tau + \lambda I\right)^{-1} \varphi_r \right|.$$

If $n \overset{\text{def}}{=} \sum_{r=1}^{S_\tau} n_r = 0$, then $\mu_{\tau,t,a} = 0$. Otherwise, set $q_r = n_r/n$, $q = (q_1, \ldots, q_{S_\tau}) \in \Delta_{S_\tau}$, and $\epsilon = \lambda/n \in (0, \lambda]$. Define

$$B_\tau(q) \overset{\text{def}}{=} U_\tau^\top \mathrm{diag}(q) U_\tau = \sum_{r=1}^{S_\tau} q_r \varphi_r \varphi_r^\top.$$

Then

$$|\mu_{\tau,t,a}| \le M_{t-1} F_\tau(q, \epsilon, c_{\tau,t,a}),$$

where

$$F_\tau(q, \epsilon, c) \overset{\text{def}}{=} \sum_{r=1}^{S_\tau} q_r \left| c^\top \left(B_\tau(q) + \epsilon I\right)^{-1} \varphi_r \right|.$$

For fixed $c \in \mathcal{C}_\tau$, $F_\tau(q, \epsilon, c)$ is continuous on $\Delta_{S_\tau} \times (0, \lambda]$. It admits a continuous extension to $\epsilon = 0$ by replacing $(B_\tau(q) + \epsilon I)^{-1}$ with the Moore–Penrose inverse $B_\tau(q)^+$: if $q_r > 0$, then $\varphi_r$ lies in the range of $B_\tau(q)$, so $(B_\tau(q) + \epsilon I)^{-1}\varphi_r \to B_\tau(q)^+\varphi_r$; if $q_r = 0$, the corresponding term is multiplied by $q_r$ and vanishes. Since $\mathcal{C}_\tau$ is finite and $\Delta_{S_\tau} \times [0, \lambda]$ is compact, define

$$L_\tau \overset{\text{def}}{=} \max_{c \in \mathcal{C}_\tau} \sup_{q \in \Delta_{S_\tau}, \, \epsilon \in [0, \lambda]} F_\tau(q, \epsilon, c) < \infty.$$

Therefore $|\mu_{\tau,t,a}| \le L_\tau M_{t-1}$. Since $\mathfrak{T}$ is finite, $L_* = \max_{\tau \in \mathfrak{T}} L_\tau < \infty$, and

$$D_{t,a} \le 2 \max_{\tau \in \mathfrak{T}} |\mu_{\tau,t,a}| \le 2L_* M_{t-1}.$$

$\square$

**Lemma 7** (Logarithmic growth of the maximal past reward). *Assume that contexts are generated independently of $\theta^*$ under the Bayesian design. Then there exists a finite constant $C_M > 0$, depending only on $(\kappa, \sigma^2)$, such that, with $M_0 \overset{\text{def}}{=} 0$, for all $t \geq 1$,*

$$\mathbb{E}[M_{t-1}^2] \leq C_M \log(eKt), \qquad M_{t-1} \overset{\text{def}}{=} \max_{1 \leq s \leq t-1} |R_s| \quad (t \geq 2).$$

*Proof.* The case $t = 1$ is immediate. Fix $t \geq 2$. For $s \geq 1$, a given arm $a$, and context $x$,

$$g_{\theta_a^*}(x) = \sum_{j=1}^{m} M_{j,\ell_j^{(a)}(x)}^{(a)},$$

where $\ell_j^{(a)}(x)$ is the leaf of tree $j$ containing $x$. Conditional on the tree structures, each selected leaf value is $N(0, \sigma_\mu^2)$, independently across $j = 1, \ldots, m$. Hence

$$g_{\theta_a^*}(x) \sim N(0, m\sigma_\mu^2) = N\left(0, \frac{1}{4\kappa^2}\right).$$

Because $X_s$ is independent of $\theta^*$, conditioning on $X_s = x$ does not change this prior law. Therefore, with $s_0^2 = 1/(4\kappa^2)$,

$$\Pr\left(\max_{a \in [K]} |f_0(X_s, a)| > u \,\middle|\, X_s\right) \leq 2K \exp\left(-\frac{u^2}{2s_0^2}\right).$$

Since $|f_0(X_s, A_s)| \leq \max_{a \in [K]} |f_0(X_s, a)|$ pathwise,

$$\Pr(|f_0(X_s, A_s)| > u) \leq 2K \exp\left(-\frac{u^2}{2s_0^2}\right).$$

The observation noise satisfies

$$\Pr(|\varepsilon_s| > u) \leq 2\exp\left(-\frac{u^2}{2\sigma^2}\right).$$

Using $R_s = f_0(X_s, A_s) + \varepsilon_s$ and $|x + y| > u \Rightarrow |x| > u/2$ or $|y| > u/2$, there is a constant $C_1 > 0$, depending only on $(\kappa, \sigma^2)$, such that

$$\Pr(|R_s| > u) \leq 2(K+1) \exp\left(-\frac{u^2}{C_1}\right).$$

Thus

$$\Pr(M_{t-1} > u) \leq 2(K+1)(t-1) \exp\left(-\frac{u^2}{C_1}\right).$$

Using the tail-integral identity for $M_{t-1}^2$ and splitting the integral at $u_0 = \sqrt{C_1 \log(eKt)}$ yields

$$\mathbb{E}[M_{t-1}^2] \leq C_M \log(eKt)$$

for a constant $C_M > 0$ depending only on $(\kappa, \sigma^2)$. $\qquad\square$

**Lemma 8** (Corrected lifted information ratio bound for Gaussian BART). *Under the assumptions of Lemma 3 and the fixed finite split-grid assumption, there exists a finite constant $C_{\mathrm{rp}} > 0$, depending only on the fixed BART prior and split-grid constants but not on $K$ or $T$, such that*

$$\bar{\Gamma}_T \leq C_{\mathrm{rp}} K \log(eKT).$$

*Proof.* By Lemma 6,

$$D_{t,a}^2 \leq 4L_*^2 M_{t-1}^2.$$

Hence the proxy in Lemma 5 satisfies

$$v_t = \sigma^2 + s_0^2 + \frac{1}{4} \max_{a \in [K]} D_{t,a}^2 \leq \sigma^2 + s_0^2 + L_*^2 M_{t-1}^2.$$

Taking expectations and applying Lemma 7,

$$\mathbb{E}[v_t] \leq \sigma^2 + s_0^2 + L_*^2 C_M \log(eKt) \leq C_v \log(eKt)$$

for a finite constant $C_v > 0$ independent of $K$ and $T$. By Lemmas 5 and 4,

$$\bar{\Gamma}_T \leq \frac{2K}{T} \sum_{t=1}^{T} \mathbb{E}[v_t] \leq \frac{2KC_v}{T} \sum_{t=1}^{T} \log(eKt).$$

Since $T^{-1} \sum_{t=1}^{T} \log(eKt) \leq C \log(eKT)$ for a universal constant $C$, the claim follows. $\qquad\square$

### D.3.2. PROOF OF LEMMA 3

*Proof.* Let $\boldsymbol{\theta}^* = ((\theta_a^*))_{a \in \mathcal{A}} \in (\Theta)^K$ collect all arm-wise BART parameters. We assume the noise variance $\sigma^2$ is fixed and not part of the parameter vector. We bound $I(\boldsymbol{\theta}^*; \mathcal{H}_T)$.

**Arm-wise decomposition.** Let $T_a = \sum_{s=1}^{T} \mathbb{I}(A_s = a)$ so that $\sum_{a=1}^{K} T_a = T$. For each arm $a$, decompose its BART parameter as $\theta_a^* = (\mathcal{T}^{(a)}, M^{(a)})$, where $\mathcal{T}^{(a)} = \{\mathcal{T}_j^{(a)}\}_{j=1}^{m}$ are tree structures and $M^{(a)} = (M_j^{(a)})_{j=1}^{m}$ are leaf values.

Because the prior factorizes across arms and the likelihood is arm-wise, we can decompose the information gain. To do so under adaptive data collection (where $A_t$ depends on past rewards), we use the following posterior factorization lemma.

**Lemma 9** (Posterior factorization under adaptive sampling). *Assume an arm-wise factorized prior $\Pi(\boldsymbol{\theta}^*) = \prod_{a=1}^{K} \Pi_a(\theta_a^*)$. Assume rewards satisfy an arm-wise likelihood: conditional on $(X_t, A_t)$, $R_t$ depends on $\boldsymbol{\theta}^*$ only through $\theta_{A_t}^*$, and rewards are conditionally independent over time given $(X_t, A_t, \boldsymbol{\theta}^*)$, i.e. for all $t$,*

$$p(R_t \mid \mathcal{H}_{t-1}, X_t, A_t, \boldsymbol{\theta}^*) = p(R_t \mid X_t, A_t, \theta_{A_t}^*).$$

*Assume additionally that the algorithm chooses $A_t$ using only $(\mathcal{H}_{t-1}, X_t)$ and internal randomness independent of $\boldsymbol{\theta}^*$, so that*

$$p(A_t \mid \mathcal{H}_{t-1}, X_t, \boldsymbol{\theta}^*) = p(A_t \mid \mathcal{H}_{t-1}, X_t).$$

*For any arm $a$, let $\mathcal{I}_a(t-1) \overset{def}{=} \{s < t : A_s = a\}$ denote the (random) pull index set up to time $t-1$. Then the marginal posterior for $\theta_a^*$ satisfies the design-conditional factorization*

$$\Pi_a(\theta_a^* \in \cdot \mid \mathcal{H}_{t-1}) = \Pi_a(\theta_a^* \in \cdot \mid X_{1:t-1}, A_{1:t-1}, (R_s)_{s \in \mathcal{I}_a(t-1)}) = \Pi_a(\theta_a^* \in \cdot \mid \mathcal{I}_a(t-1), (X_s, R_s)_{s \in \mathcal{I}_a(t-1)}).$$

Using Lemma 9, the likelihood factors by arm given the realized design $(X_{1:T}, A_{1:T})$, hence the posterior also factorizes across arms:

$$\Pi(\boldsymbol{\theta}^* \mid \mathcal{H}_T) = \bigotimes_{a=1}^{K} \Pi_a(\theta_a^* \mid \mathcal{H}_T).$$

Therefore, by the KL representation of mutual information and additivity of KL for product measures,

$$I(\boldsymbol{\theta}^*; \mathcal{H}_T) = \mathbb{E}[D_{\mathrm{KL}}(\Pi(\cdot \mid \mathcal{H}_T) \,\|\, \Pi)] = \sum_{a=1}^{K} \mathbb{E}[D_{\mathrm{KL}}(\Pi_a(\cdot \mid \mathcal{H}_T) \,\|\, \Pi_a)] = \sum_{a=1}^{K} I(\theta_a^*; \mathcal{H}_T).$$

It remains to bound each arm-wise mutual information term. For each arm $a$, by the chain rule,

$$I(\theta_a^*; \mathcal{H}_T) = I(\mathcal{T}^{(a)}; \mathcal{H}_T) + I(M^{(a)}; \mathcal{H}_T \mid \mathcal{T}^{(a)}),$$

so we bound the tree-structure and leaf-value contributions separately.

**Bounding Tree Structure Information.**

Fix an arm $a$. Since $\mathcal{T}^{(a)}$ is discrete, we have $I(\mathcal{T}^{(a)}; \mathcal{H}_T) \leq H(\mathcal{T}^{(a)})$. Therefore, it suffices to derive the Shannon entropy $H(\mathcal{T}^{(a)})$ of the tree structure prior. We suppose that the total number of leaves for arm $a$ is $L^{(a)}$. The trees in the ensemble are independent, so

$$H(\mathcal{T}^{(a)}) = \sum_{j=1}^{m} H(\mathcal{T}_j^{(a)}) = \sum_{j=1}^{m} \left( H(L_j^{(a)}) + H(\mathcal{T}_j^{(a)} \mid L_j^{(a)}) \right).$$

Using the argument from Ročková & Saha (2019, Corollary 5.2), which establishes a super-exponential tail bound $\mathbb{P}(L_j > l) \lesssim e^{-C_K l \log l}$, it is straightforward to show that $\mathbb{P}(L_j > l) \leq C\rho^l$ for some constant $C$ and $\rho \in (0,1)$ depending on $\alpha_{\mathrm{qd}}$. The entropy

$$H(L_j) = \sum_{l \geq 1} \mathbb{P}(L_j = l) \log \frac{1}{\mathbb{P}(L_j = l)}.$$

To bound the entropy of $L_j$, we use a standard tail-indicator (binary-entropy) bound. Define $B_l \stackrel{\text{def}}{=} \mathbf{1}\{L_j > l\}$ for $l \geq 1$. Then $L_j = 1 + \sum_{l \geq 1} B_l$ and the map $L_j \leftrightarrow (B_l)_{l \geq 1}$ is bijective, hence $H(L_j) = H((B_l)_{l \geq 1})$. By the chain rule and subadditivity,

$$H(L_j) = \sum_{l \geq 1} H(B_l \mid B_1, \ldots, B_{l-1}) \leq \sum_{l \geq 1} H(B_l) = \sum_{l \geq 1} h(\mathbb{P}(L_j > l)),$$

where $h(p) = p \log \frac{1}{p} + (1-p) \log \frac{1}{1-p}$ is the binary entropy. Using $h(p) \leq p \log \frac{e}{p} = p \log \frac{1}{p} + p$, we obtain the valid (looser) tail-only bound

$$H(L_j) \leq \sum_{l \geq 1} \mathbb{P}(L_j > l) \log \frac{e}{\mathbb{P}(L_j > l)} \leq \sum_{l \geq 1} C\rho^l \left( l \log \frac{1}{\rho} + \log \frac{e}{C} \right) < \infty.$$

The set of all trees with $l$ leaves is finite. Thus,

$$H(\mathcal{T}_j \mid L_j) = \mathbb{E}_{l \sim L_j}[H(\mathcal{T}_j \mid L_j = l)] \leq \mathbb{E}_{l \sim L_j}[\log|\{\tau : \mathrm{leaves}(\tau) = l\}|].$$

The number of distinct full binary tree shapes with $l$ leaves is the $(l-1)$th Catalan number

$$\mathrm{Cat}_{l-1} = \frac{1}{l} \binom{2(l-1)}{l-1} = \mathcal{O}(4^{l-1}/(l-1)^{3/2}) \quad \text{as } l \to \infty.$$

At each of the $l-1$ internal nodes, one chooses a splitting variable among $p$ and a cut-point among (at most) $N_{\max}$ observed values, yielding at most $(pN_{\max})^{l-1}$ choices. Combining together,

$$\log|\{\tau : \mathrm{leaves}(\tau) = L_j\}| \leq \log(4^{l-1}/(l-1)^{3/2}) + \log C_1 + \log((pN_{\max})^{l-1})$$
$$= (l-1)\log(4pN_{\max}) - \frac{3}{2}\log(l-1) + \log C_1.$$

*Remark* 3. Using the Dirichlet prior for the split probabilities is no different from using (uniformly) random probabilities in terms of the entropy bound for the prior itself.

Taking expectation,

$$\mathbb{E}_{l \sim L_j^{(a)}} \left[ \log|\{\tau : \mathrm{leaves}(\tau) = L_j^{(a)}\}| \right] \leq C_2 \log(4pN_{\max}) + C_3.$$

Finally, summing over arms,

$$\sum_{a=1}^{K} I(\mathcal{T}^{(a)}; \mathcal{H}_T) \leq \sum_{a=1}^{K} H(\mathcal{T}^{(a)}) \leq K \, m \big( C_2 \log(4pN_{\max}) + C_4 \big).$$

Here, $C_1, C_2, C_3$ and $C_4$ are constants that only depend on the splitting probability parameters.

**Bounding Leaf Information.** Fix an arm $a$ and let $\mathcal{I}_a \stackrel{\text{def}}{=} \{t \leq T : A_t = a\}$, $X^{(a)} \stackrel{\text{def}}{=} (X_t)_{t \in \mathcal{I}_a}$, and $R^{(a)} \stackrel{\text{def}}{=} (R_t)_{t \in \mathcal{I}_a}$. By Lemma 9, conditional on $\mathcal{T}^{(a)}$ the posterior for $M^{(a)}$ depends on $\mathcal{H}_T$ only through $(\mathcal{I}_a, X^{(a)}, R^{(a)})$. Therefore, by the KL representation of conditional mutual information,

$$I(M^{(a)}; \mathcal{H}_T \mid \mathcal{T}^{(a)}) = \mathbb{E}\Big[ \mathrm{D}_{\mathrm{KL}}\Big( \Pi(M^{(a)} \mid \mathcal{I}_a, X^{(a)}, R^{(a)}, \mathcal{T}^{(a)}) \,\big\|\, \Pi(M^{(a)} \mid \mathcal{T}^{(a)}) \Big) \Big]$$
$$= \mathbb{E}\Big[ \mathbb{E}\Big[ \mathrm{D}_{\mathrm{KL}}\Big( \Pi(M^{(a)} \mid D_a, R^{(a)}, \mathcal{T}^{(a)}) \,\big\|\, \Pi(M^{(a)} \mid \mathcal{T}^{(a)}) \Big) \,\big|\, D_a, \mathcal{T}^{(a)} \Big] \Big],$$

where the realized design for arm $a$ is $D_a \overset{\text{def}}{=} (\mathcal{I}_a, X^{(a)})$. Conditioning on a realization $\tau$ of $\mathcal{T}^{(a)}$ and on the realized arm-$a$ design, the model reduces to a Gaussian linear model, so the inner expected KL information gain equals $\frac{1}{2} \log \det(I + (\sigma_\mu^2/\sigma^2)X_\tau^\top X_\tau)$, whose proof is given below.

Under fixed tree structures $\tau$, BART places independent priors $M_j^{(a)} \sim N(0, \sigma_\mu^2)$ on each tree $j$'s leaf value vector for arm $a$. $M^{(a)}$, being the concatenation of all $M_j^{(a)}$'s, is a length $L^{(a)}$ vector. Let $X_\tau^{(a)}$ be the $T_a \times L^{(a)}$ matrix of leaf indicators of all $(X_s : A_s = a)$ pairs. Let the reward vector $R^{(a)}$ collect the $T_a$ rewards with $A_s = a$. For any fixed $X_\tau^{(a)}$,

$$R^{(a)} \mid M^{(a)} \sim N(X_\tau^{(a)} M^{(a)}, \sigma^2 I_{T_a}), \quad M^{(a)} \sim N(0, \sigma_\mu^2 I_{L^{(a)}}).$$

By standard Gaussian-Gaussian conjugacy,

$$M^{(a)} \mid R^{(a)} \sim N\Big( \big( (X_\tau^{(a)})^\top X_\tau^{(a)} + (\sigma^2/\sigma_\mu^2) I_{L^{(a)}} \big)^{-1} (X_\tau^{(a)})^\top R^{(a)},$$

$$\sigma^2 \big( (X_\tau^{(a)})^\top X_\tau^{(a)} + (\sigma^2/\sigma_\mu^2) I_{L^{(a)}} \big)^{-1} \Big).$$

Indeed, this posterior is conditioned on $\tau$ and the realized arm-$a$ observations $(X^{(a)}, R^{(a)})$. Using the following fact from Duchi (2007):

$$D_{\text{KL}}(P\|Q) = \frac{1}{2}\Big( (\mu_2 - \mu_1)^\top \Sigma_2^{-1} (\mu_2 - \mu_1) + \text{tr}(\Sigma_2^{-1}\Sigma_1)$$

$$- \log \frac{\det \Sigma_1}{\det \Sigma_2} - L \Big), \text{ if } P \sim N(\mu_1, \Sigma_1), Q \sim N(\mu_2, \Sigma_2),$$

the K-L divergence between the two multivariate normal distributions $M^{(a)} \mid \tau, X^{(a)}, R^{(a)}$ and $M^{(a)} \mid \tau$ is thus

$$D_{\text{KL}}(\Pi(M^{(a)} \mid \tau, X^{(a)}, R^{(a)})\|\Pi(M^{(a)} \mid \tau)) = \frac{1}{2}\Big( \mu_1^\top \Sigma_2^{-1} \mu_1 + \text{tr}(\Sigma_2^{-1}\Sigma_1) - L^{(a)} + \log \det(\Sigma_2\Sigma_1^{-1}) \Big),$$

where

$$\mu_1 = \big( (X_\tau^{(a)})^\top X_\tau^{(a)} + (\sigma^2/\sigma_\mu^2) I_{L^{(a)}} \big)^{-1} (X_\tau^{(a)})^\top R^{(a)},$$

$$\Sigma_1 = \sigma^2 \big( (X_\tau^{(a)})^\top X_\tau^{(a)} + (\sigma^2/\sigma_\mu^2) I_{L^{(a)}} \big)^{-1},$$

$$\mu_2 = \mathbf{0},$$

$$\Sigma_2 = \sigma_\mu^2 I_{L^{(a)}}.$$

Given the total law of variance

$$\Sigma_2 = \text{Var}(M^{(a)} \mid \tau) = \mathbb{E}_{R^{(a)}|X^{(a)},\tau} \text{Var}(M^{(a)} \mid \tau, X^{(a)}, R^{(a)}) + \text{Var}_{R^{(a)}|X^{(a)},\tau} \mathbb{E}(M^{(a)} \mid \tau, X^{(a)}, R^{(a)}) = \Sigma_1 + \text{Var}_{R^{(a)}|X^{(a)},\tau}[\mu_1],$$

the conditional expectation of the first three terms of the K-L divergence adds up to 0 by

$$\mathbb{E}_{R^{(a)}|X^{(a)},\tau}[\mu_1^\top \Sigma_2^{-1} \mu_1] = \text{tr}(\Sigma_2^{-1} \text{Var}_{R^{(a)}|X^{(a)},\tau}(\mu_1)) = \text{tr}(\Sigma_2^{-1}(\Sigma_2 - \Sigma_1)).$$

Further using $\det(\Sigma_2\Sigma_1^{-1}) = \det((\sigma_\mu^2/\sigma^2)(X_\tau^{(a)})^\top X_\tau^{(a)} + I_{L^{(a)}})$,

$$\mathbb{E}_{R^{(a)}|X^{(a)},\mathcal{I}_a,\tau}[D_{\text{KL}}(\Pi(M^{(a)} \mid \tau, X^{(a)}, R^{(a)})\|\Pi(M^{(a)} \mid \tau))] = \frac{1}{2} \log \det((\sigma_\mu^2/\sigma^2)(X_\tau^{(a)})^\top X_\tau^{(a)} + I_{L^{(a)}}).$$

Suppose the $r \leq \min(T_a, L^{(a)})$ eigenvalues of $(X_\tau^{(a)})^\top X_\tau^{(a)}$ are $\gamma_1, \ldots, \gamma_r$. By $\text{tr}((X_\tau^{(a)})^\top X_\tau^{(a)}) = mT_a$ and the AM–GM inequality,

$$\det((\sigma_\mu^2/\sigma^2)(X_\tau^{(a)})^\top X_\tau^{(a)} + I_{L^{(a)}}) = \prod_{i=1}^{r}(1 + (\sigma_\mu^2/\sigma^2)\gamma_i)$$

$$\leq \left( \frac{1}{r} \sum_{i=1}^{r}(1 + (\sigma_\mu^2/\sigma^2)\gamma_i) \right)^r$$

$$\leq \left( 1 + \frac{\sigma_\mu^2 T_a}{\sigma^2 r}m \right)^r \leq \left( 1 + \frac{\sigma_\mu^2 T_a}{\sigma^2}m \right)^r.$$

Finally, using $\sigma_\mu = 0.5/(\kappa\sqrt{m})$ (so that $\sigma_\mu^2 m = \frac{1}{4\kappa^2}$),

$$I(M^{(a)}; \mathcal{H}_T \mid \mathcal{T}^{(a)}) = \mathbb{E}_{D_a, \, \tau \sim \mathcal{T}^{(a)}} \left[ \frac{1}{2} \log \det\left( I_{L^{(a)}} + (\sigma_\mu^2/\sigma^2)(X_\tau^{(a)})^\top X_\tau^{(a)} \right) \right]$$

$$\leq \frac{1}{2} \min(T_a, \mathbb{E}_{\tau \sim \mathcal{T}^{(a)}}[L^{(a)}]) \log\left( 1 + \frac{1}{4\kappa^2\sigma^2} T_a \right).$$

Summing over arms and using $\sum_a T_a = T$ and concavity of $\log(1 + u)$,

$$\sum_{a=1}^{K} I(M^{(a)}; \mathcal{H}_T \mid \mathcal{T}^{(a)}) \leq \frac{1}{2} \mathbb{E}_{\tau \sim \mathcal{T}}[L] \sum_{a=1}^{K} \log\left( 1 + \frac{1}{4\kappa^2\sigma^2} T_a \right) \leq \frac{1}{2} K \, \mathbb{E}_{\tau \sim \mathcal{T}}[L] \log\left( 1 + \frac{1}{4\kappa^2\sigma^2} \frac{T}{K} \right).$$

Note that under the BART prior (Section B), the expected total number of leaves satisfies $\mathbb{E}_{\tau \sim \mathcal{T}}[L] \leq C_{\text{leaf}} m$ for a constant

$$C_{\text{leaf}} \leq \sum_{l=1}^{\infty} C\rho^l = \frac{C}{1 - \rho}$$

depending only on the splitting prior parameters. To simplify:

$$\sum_{a=1}^{K} I(M^{(a)}; \mathcal{H}_T \mid \mathcal{T}^{(a)}) \leq \frac{1}{2} K C_{\text{leaf}} m \log\left( 1 + \frac{1}{4\kappa^2\sigma^2} \frac{T}{K} \right).$$

Combining with the structure bound for $I(\mathcal{T}^{(a)}; \mathcal{H}_T)$ gives the stated lemma. $\qquad\square$

### D.3.3. PROOF OF LEMMA 9

*Proof.* Write $\boldsymbol{\theta}^* = (\theta_1^*, \ldots, \theta_K^*)$. By Bayes' rule,

$$p(\boldsymbol{\theta}^* \mid \mathcal{H}_{t-1}) \propto \Pi(\boldsymbol{\theta}^*) \, p(\mathcal{H}_{t-1} \mid \boldsymbol{\theta}^*).$$

Factorize $p(\mathcal{H}_{t-1} \mid \boldsymbol{\theta}^*)$ sequentially:

$$p(\mathcal{H}_{t-1} \mid \boldsymbol{\theta}^*) = \prod_{s=1}^{t-1} p(X_s \mid \mathcal{H}_{s-1}, \boldsymbol{\theta}^*) \, p(A_s \mid \mathcal{H}_{s-1}, X_s, \boldsymbol{\theta}^*) \, p(R_s \mid \mathcal{H}_{s-1}, X_s, A_s, \boldsymbol{\theta}^*).$$

Under exogenous contexts, $p(X_s \mid \mathcal{H}_{s-1}, \boldsymbol{\theta}^*) = p(X_s \mid \mathcal{H}_{s-1})$. By the algorithm assumption, $p(A_s \mid \mathcal{H}_{s-1}, X_s, \boldsymbol{\theta}^*) = p(A_s \mid \mathcal{H}_{s-1}, X_s)$. By the arm-wise likelihood, $p(R_s \mid \mathcal{H}_{s-1}, X_s, A_s, \boldsymbol{\theta}^*) = p(R_s \mid X_s, A_s, \theta_{A_s}^*)$. Therefore all $\boldsymbol{\theta}^*$-dependence in $p(\mathcal{H}_{t-1} \mid \boldsymbol{\theta}^*)$ is through the reward factors, and these reward factors separate by arm:

$$p(\mathcal{H}_{t-1} \mid \boldsymbol{\theta}^*) \propto \prod_{a=1}^{K} \prod_{s<t: A_s = a} p(R_s \mid X_s, A_s = a, \theta_a^*),$$

where $\propto$ hides factors that depend only on the realized design $(X_{1:t-1}, A_{1:t-1})$ (and not on $\boldsymbol{\theta}^*$). Combining with $\Pi(\boldsymbol{\theta}^*) = \prod_{a=1}^{K} \Pi_a(\theta_a^*)$ and integrating out $\theta_{-a}^*$ yields

$$p(\theta_a^* \mid \mathcal{H}_{t-1}) \propto \Pi_a(\theta_a^*) \prod_{s<t: A_s = a} p(R_s \mid X_s, A_s = a, \theta_a^*),$$

which depends on $\mathcal{H}_{t-1}$ only through the realized design $(X_{1:t-1}, A_{1:t-1})$ and the arm-$a$ subsequence $(X_s, R_s)_{s \in \mathcal{I}_a(t-1)}$, equivalently $\mathcal{I}_a(t-1)$ and $(X_s, R_s)_{s \in \mathcal{I}_a(t-1)}$. This proves the claim. $\qquad\square$

# E. A Feel-Good Variant of BFTS

**Purpose and scope.** This appendix provides supporting theory for a related "feel-good" variant (Zhang, 2022) of BFTS. The variant is identical to BFTS except for an exponential tilting/resampling step over the same BART posterior draws (so $\lambda = 0$ recovers standard BFTS, and the optimal $\lambda^\star \asymp \epsilon_T \to 0$). The frequentist regret bound in this section applies to the variant rather than standard BFTS. Empirically, the variant can be sensitive to tuning and may underperform standard BFTS; see Section E.8 for sensitivity results.

**Roadmap.** We first introduce the variant and its implementation (Algorithm 1v), then summarize the frequentist regret guarantee and give the proof, and finally report sensitivity results and the practical takeaway.

## E.1. Introduction to Feel-Good BFTS

While the Bayesian analysis yields tight bounds under the prior, it offers no guarantees when the data-generating process deviates from the BART prior. To provide robust guarantees for fixed, unknown functions $f_0$, we now turn to a *Frequentist design*, where $f_0$ is fixed and unknown and the prior used may be misspecified.

Standard TS is difficult to analyze in frequentist nonparametric settings due to insufficient exploration. To promote exploration, Zhang (2022) proposed Feel-Good Thompson Sampling (FGTS) by adding an optimistic ("feel-good") term that favors models with large maximum rewards on historical observations. Crucially, they decomposed the regret into a decoupling coefficient (capturing the algorithmic structure) and estimation errors (capturing the model loss and the feel-good exploration loss). They proved that the decoupling coefficient is bounded by the number of actions. Neu et al. (2022); Gouverneur et al. (2023) extended these ideas to the Bayesian regret setting.

We leverage their Feel-Good framework as a theoretical tool. We demonstrate that the BART prior satisfies the necessary posterior concentration properties (Condition 1) to yield adaptive regret rates. We also show that a generic $L_2(Q)$ prior-mass (contraction) condition (Condition 1) BART implies $\sum_{t=1}^{T} \mathbb{E}_{\text{policy}}[\text{Regret}_t] \leq \widetilde{\mathcal{O}}(T\epsilon_T)$; in particular this yields $\widetilde{\mathcal{O}}(KT^{(\alpha+p)/(2\alpha+p)})$ in the dense Hölder setting and $\widetilde{\mathcal{O}}(KT^{(\alpha+d)/(2\alpha+d)} \operatorname{polylog}(T, p))$ under covariate sparsity. This $L_2(Q)$ ball condition is weaker than $L_\infty$ ball conditions discussed by Agarwal & Zhang (2022).

### E.1.1. BACKGROUND

The frequentist expected cumulative regret is

$$\mathbb{E}\Big[ \sum_{t=1}^{T} \big( f_0(x_t, A^*(f_0, x_t)) - f_0(x_t, A_t) \big) \Big],$$

with expectation over the reward noise and algorithmic randomness (conditioning on the realized contexts $(x_t)$).

The Feel-Good TS (FGTS) loss (Zhang, 2022) is defined as

$$L_{\text{FGTS}}(\boldsymbol{\theta}, x, a, r) = \eta \left( f_{\boldsymbol{\theta}}(x, a) - r \right)^2 - \lambda \min \left( b, f_{\boldsymbol{\theta}}^*(x) \right),$$

where $\eta > 0$, $\lambda \geq 0$, $b > 0$, and $f_{\boldsymbol{\theta}}^*(x) = \max_{a' \in \mathcal{A}} f_{\boldsymbol{\theta}}(x, a')$. This term encourages exploration by artificially boosting the probability of sampling models with high potential maximum rewards, acting as a "soft" version of UCB's optimism.

For the frequentist analysis, we treat $f_0$ as fixed and unknown. For each arm $a$, we assume $f_0(\cdot, a)$ lies in a Hölder class $\mathcal{C}^{\alpha,R}([0,1]^p)$ with $0 < \alpha \leq 1$, and we adopt arm-wise BART priors and designs that satisfy standard contraction conditions (entropy, prior mass, and sieve bounds) under regular design assumptions (Ročková & van der Pas, 2020; Ročková, 2020; Jeong & Ročková, 2023). A formal statement of these conditions is provided in Section E.7.

Our contribution is to show that the following Condition 1 on a prior mass lower bound for an $L_2(Q)$ ball around the truth suffices for an optimal regret bound, which is the *only* nonparametric-Bayes input needed for the regret bound. Such prior concentration is a standard ingredient in posterior contraction theory (Ghosal & Van Der Vaart, 2017) and, in our setting, follows from the random-design split-net framework of Jeong & Ročková (2023) (which accommodates Dirichlet-sparse variants); see also Ročková & van der Pas (2020); Ročková (2020); Linero (2018) for related results. This is a weaker condition than $L_\infty$ ball conditions discussed by previous literature (Agarwal & Zhang, 2022) for feel-good posterior sampling.

**Condition 1** (Generic nonparametric prior mass condition). Under appropriate assumptions (see Section E.7), there exists a rate $\epsilon_n \to 0$ such that the induced prior $\Pi(g_\theta)$ over the regression function $g_\theta(\cdot)$ satisfies

$$\Pi\left(\mathcal{B}_{\epsilon_n}^{(a)}\right) \geq \exp(-c_0 n \epsilon_n^2), \tag{1}$$

for some constant $c_0 > 2$, where $\mathcal{B}_\epsilon^{(a)} \overset{\text{def}}{=} \{g_\theta : \|g_\theta - g_{0,a}\|_Q \leq \epsilon\}$ denotes an $L_2(Q)$ ball around the truth for arm $a$.

Under $d$-sparsity and a Dirichlet-sparse BART prior, Jeong & Ročková (2023) yields a prior contraction rate of the form $\epsilon_n = \sqrt{(d \log p)/n} + n^{-\alpha/(2\alpha+d)} \operatorname{polylog}(n,d)$, where the first term is a (typically lower-order) variable-selection penalty.

Specifically, the following assumption on the true mean function is needed for the condition 1 to hold.

**Assumption 1** (True mean function). For each $a \in \mathcal{A}$, $g_{0,a}$, the true mean function, is fixed, unknown, and belongs to the sparse piecewise heterogeneous anisotropic Hölder class $\Gamma_\lambda^{A_{\bar{\alpha}},d,p}(\mathfrak{X}_0)$ (optionally intersected with $\mathcal{C}([0,1]^p)$) as in Jeong & Ročková (2023, (A1)). A sufficient condition is the sparse-isotropic continuous specialization: for each arm, $g_{0,a}(x) = h_{0,a}(x_{S_{0,a}})$ with $|S_{0,a}| \leq d$ and $h_{0,a}$ being bounded is $\alpha$-Hölder on $[0,1]^d$.

We assume $h_{0,a} \in [0,1]$ from now on. This is equivalent to a bounded true mean function $f_0$.

### E.1.2. MAIN RESULT

**Theorem 10** (Nonparametric regret for FGTS with BART prior). *Suppose a random-design context process $X_t \overset{\text{iid}}{\sim} Q$ and arm-wise BART priors satisfy Condition 1 under the truncation conditions summarized in Section E.7. Suppose stochastic rewards and contexts (T1 and T3) hold. Suppose the true mean function satisfies Assumption 1. Then, for the FGTS objective with parameters $(\eta, \lambda)$ tuned as described in Section E (assuming the algorithm knows a lower bound $\alpha$ on the smoothness and an upper bound $d$ on the useful dimension, so that a valid rate $\epsilon_T$ can be specified), for sufficiently large $T$ (known to the algorithm), we have*

$$\sum_{t=1}^T \mathbb{E}_{\text{policy}}[\text{Regret}_t] \leq \widetilde{\mathcal{O}}\big(T \epsilon_T\big),$$

*with constants depending on $(\eta, c_0)$ and prior hyperparameters (where $c_0$ is the contraction constant in the prior-mass condition), and with an explicit linear factor in $K$ hidden in the soft-O notation. The optimal $\lambda^* \asymp \epsilon_T$. See Section E for the proof.*

**Corollary 11.** *In particular, in the non-sparse isotropic setting where $f_0(\cdot, a)$ may depend on all $p$ coordinates and $\epsilon_T \asymp T^{-\alpha/(2\alpha+p)} \log^{1/2} T$ (Ročková & van der Pas, 2020; Ročková, 2020), we obtain*

$$\sum_{t=1}^T \mathbb{E}_{\text{policy}}[\text{Regret}_t] \leq \widetilde{\mathcal{O}}\big(K T^{(\alpha+p)/(2\alpha+p)}\big).$$

*Under additional $d$-sparsity of the covariates (each $f_0(\cdot, a)$ depends on at most $d \ll p$ coordinates) and with a Dirichlet-sparse BART prior as in Linero (2018); Jeong & Ročková (2023), one may instead take $\epsilon_T \asymp T^{-\alpha/(2\alpha+d)} \operatorname{polylog}(T,p)$, which yields the improved sparse-regime bound*

$$\sum_{t=1}^T \mathbb{E}_{\text{policy}}[\text{Regret}_t] \leq \widetilde{\mathcal{O}}\big(K T^{(\alpha+d)/(2\alpha+d)} \operatorname{polylog}(T,p)\big).$$

*Remark* 4. The optimal $\lambda^*$ and $\tau$ changes with $T$, so the basic FGTS is not anytime. A standard doubling trick (Lattimore & Szepesvári, 2020) provides an anytime variant with at most a constant-factor overhead; see Corollary 15 for details.

**Rate interpretation and regimes.** The rate $\widetilde{\mathcal{O}}(T^{(\alpha+p)/(2\alpha+p)})$ improves with larger $\alpha$ and degrades with larger $p$. Under covariate sparsity and Dirichlet-sparse BART priors, the effective dimension $d$ replaces $p$ in the exponent, yielding $\widetilde{\mathcal{O}}(T^{(\alpha+d)/(2\alpha+d)})$ up to $\operatorname{polylog}(T,p)$ factors; this mitigates the curse of dimensionality when only a few coordinates drive rewards. Moreover, in standard nonparametric bandit settings with Hölder-smooth rewards, these rates match known minimax lower bounds up to logarithmic factors (Rigollet & Zeevi, 2010).

---

**Algorithm 1v** BFTS (FG variant).

---

**Input:** horizon $T$; initial selections $\tau$ (per arm); action set $\mathcal{A}$ with $K = |\mathcal{A}|$; prior $\Pi(\boldsymbol{\theta})$; FGTS parameters $\eta > 0, \lambda > 0$, $b > 0$; number of MCMC draws $N$.

Initialize history $\mathcal{H}_0 \leftarrow \emptyset$.

Initialize posterior $\Pi_0(\boldsymbol{\theta}) \propto \Pi(\boldsymbol{\theta})$.

Initialize feel-good scores $S_1, \ldots, S_N \leftarrow 0$.

Define a nondecreasing refresh index $r : \{0, 1, \ldots\} \rightarrow \mathbb{N}$ with $r(0) = 0$.

**for** $t = 1$ **to** $T$ **do**

    Observe context $X_t$.

    **if** $t \leq \tau K$ **then**

        Choose $A_t \leftarrow 1 + ((t - 1) \bmod K)$ (round-robin).

    **else**

        Use posterior draws $\{\boldsymbol{\theta}^{(j)}\}_{j=1}^N$ from the latest MCMC refresh targeting $\Pi_{t-1}(\boldsymbol{\theta})$.

        Sample $j^\star$ with $\Pr(j^\star = j) \propto \exp(\lambda S_j)$ {If $\lambda = 0$, this is uniform and recovers BFTS.}

        Choose $A_t \in \arg\max_{a \in \mathcal{A}} f_{\boldsymbol{\theta}^{(j^\star)}}(X_t, a)$ (break ties by minimum index).

    **end if**

    Observe reward $R_t$.

    Update history $\mathcal{H}_t \leftarrow \mathcal{H}_{t-1} \cup \{(X_t, A_t, R_t)\}$.

    **if** $r(t) > r(t - 1)$ **then**

        Run MCMC (cold-start) targeting $\Pi_t(\boldsymbol{\theta}) \propto \Pi(\boldsymbol{\theta}) \exp\big(-\eta \sum_{s=1}^t (f_{\boldsymbol{\theta}}(X_s, A_s) - R_s)^2\big)$.

        Obtain updated draws $\{\boldsymbol{\theta}^{(j)}\}_{j=1}^N \sim \Pi_t(\boldsymbol{\theta})$.

        Recompute $S_j \leftarrow \sum_{s=1}^t \min\big(b, f_{\boldsymbol{\theta}^{(j)}}^*(X_s)\big)$ for all $j = 1, \ldots, N$.

    **else**

        $\Pi_t \leftarrow \Pi_{t-1}$.

        Set $S_j \leftarrow S_j + \min\big(b, f_{\boldsymbol{\theta}^{(j)}}^*(X_t)\big)$ for all $j = 1, \ldots, N$.

    **end if**

**end for**

---

## E.2. The Algorithm

To implement the FGTS modification of the loss function, we use Sampling Importance Resampling (SIR) to sample from the posterior; and fix $\sigma^2 = \frac{1}{2\eta}$ in the BART MCMC sampler for implementing the parameter $\eta$. Algorithm 1v is the BFTS (FG variant) algorithm.

*Remark* 5. Fixing $\sigma^2 = \frac{1}{2\eta}$ in the BART prior is only for convenient sampling from the FGTS pseudo-posterior using existing BART MCMC samplers. It does not mean the true variance is fixed.

## E.3. Regret Bound for Feel-Good BFTS

Let $\boldsymbol{\theta} = ((\theta_a))_{a \in \mathcal{A}} \in \boldsymbol{\Theta} = \prod_{a \in \mathcal{A}} \Theta$. For every $a \in \mathcal{A}$, define $f_{\boldsymbol{\theta}}(x, a) = g_{\theta_a}(x)$, where each $\theta_a \in \Theta$ is independently sampled from the arm-wise BART prior $\pi_{\text{BART}}$ and the joint prior is $\Pi = \pi_{\text{BART}}^{\otimes K}$. Under appropriate conditions, we have the following expected regret bound for FGTS equipped with this prior.

$$\sum_{t=1}^T \mathbb{E}[\text{Regret}_t] \leq \mathcal{O}\Big(K\, T^{(\alpha+d)/(2\alpha+d)} \operatorname{polylog}(T, p)\Big).$$

Here $d$ is the intrinsic dimension: each arm-wise mean function depends on at most $d \leq p$ coordinates (the dense case is recovered by taking $d = p$).

## E.4. Proof of the Main Regret Bound

*Proof.* **Step 1 (Setup: random design and FGTS).** We work under a random-design assumption on contexts: $(X_t)_{t \geq 1}$ are i.i.d. draws from a distribution $Q$ supported on $[0, 1]^p$ with a bounded density. We assume the reward model $R_t = f_0(X_t, A_t) + \varepsilon_t$ with $\mathbb{E}[\varepsilon_t \mid \mathcal{H}_{t-1}, X_t, A_t] = 0$. This allows us to control the feel-good (max-over-arms) term in expectation

using an $L_2(Q)$ bound. Throughout this proof, we write $\mathbb{E}[\cdot] \overset{\text{def}}{=} \mathbb{E}_{X_{1:T} \sim Q} \mathbb{E}_{\text{policy}}[\cdot]$, where $\mathbb{E}_{\text{policy}}[\cdot] = \mathbb{E}[\cdot \mid X_{1:T}]$ is the conditional expectation given the realized context sequence. Fix a horizon $T$ and an per-arm exploration length $\tau = \tau(T)$, so that the total exploration length is $\tau K \in \{0, 1, \dots, T\}$.

To ensure validity of the prior mass condition, we analyze a BFTS (FG variant) procedure with a per-arm exploration length $\tau = \tau(T)$ such that $\tau \to \infty$ and $\tau = o(T\epsilon_T^2)$. This device ensures each arm is sampled exactly $\tau$ times for the prior-mass condition to be invoked, while contributing only a lower-order regret term $O(\tau K)$ that does not affect the final rate. Accordingly, it suffices to bound the regret over rounds $t = \tau K + 1, \dots, T$.

**Step 2 (Apply Feel-Good TS on phase II).** We will use the following theorem for FGTS (Zhang, 2022) (and can be applied to the post-exploration phase by taking the "prior" to be the posterior after exploration):

**Theorem 12** (Feel-Good TS regret bound (Zhang, 2022)). *Let $b = 1$ and $\eta \le 0.25$. If $|\mathcal{A}(X_t)| \le K$ for all $X_t$, then we have the following expected regret bound*

$$\sum_{t=1}^{T} \mathbb{E}[\text{Regret}_t] \le \frac{\lambda KT}{\eta} + 6\lambda T - \frac{1}{\lambda} \mathbb{E} \log \mathbb{E}_{\boldsymbol{\theta} \sim \Pi} \exp\left(-\sum_{s=1}^{T} \Delta L(\boldsymbol{\theta}, X_s, A_s, R_s)\right),$$

*where*

$$\Delta L(\boldsymbol{\theta}, x, a, r) = \eta\left((f_{\boldsymbol{\theta}}(x, a) - r)^2 - (f_0(x, a) - r)^2\right) - \lambda\left(\min(b, f_{\boldsymbol{\theta}}^*(x)) - f_0^*(x)\right).$$

Applying the theorem to rounds $\tau K + 1, \dots, T$ yields

$$\sum_{t=\tau K+1}^{T} \mathbb{E}[\text{Regret}_t] \le \frac{\lambda K(T - \tau K)}{\eta} + 6\lambda(T - \tau K) - \frac{1}{\lambda} \mathbb{E} \log \mathbb{E}_{\boldsymbol{\theta} \sim \Pi_{\tau K}} \exp\left(-\sum_{s=\tau K+1}^{T} \Delta L(\boldsymbol{\theta}, X_s, A_s, R_s)\right),$$

where $\Pi_{\tau K}(\cdot)$ denotes the (random) posterior after the exploration phase. Conditioning on the exploration history $S_{\tau K}$, we apply the theorem to rounds $\tau K + 1, \dots, T$ with initial distribution $\Pi_{\tau K}$, and then take expectation over $S_{\tau K}$. Thus,

$$\sum_{t=1}^{T} \mathbb{E}[\text{Regret}_t] \le \tau K + \frac{\lambda K(T - \tau K)}{\eta} + 6\lambda(T - \tau K) - \frac{1}{\lambda} Z_{T, \tau K},$$

with

$$Z_{T, \tau K} \overset{\text{def}}{=} \mathbb{E} \log \mathbb{E}_{\boldsymbol{\theta} \sim \Pi_{\tau K}} \exp\left(-\sum_{s=\tau K+1}^{T} \Delta L(\boldsymbol{\theta}, X_s, A_s, R_s)\right).$$

**Step 3 (Lower bound $Z_{T, \tau K}$).** It remains to lower bound $Z_{T, \tau K}$. For convenience, define $g_{0,a}(x) = f_0(x, a)$. Then, define

$$\mathcal{F} = \{g_\theta : \theta \in \Theta\},$$
$$\mathcal{B}_\epsilon^{(a)} = \{g_\theta : \|g_\theta - g_{0,a}\|_Q \le \epsilon\},$$
$$\Theta_\epsilon^{(a)} = \{\theta \in \Theta : g_\theta \in \mathcal{B}_\epsilon^{(a)}\},$$
$$\boldsymbol{\Theta}_\epsilon = \prod_{a \in \mathcal{A}} \Theta_\epsilon^{(a)},$$

where $\|h\|_Q = \sqrt{\mathbb{E}_{X \sim Q}[h(X)^2]}$ denotes the population $L_2(Q)$ norm.

*Remark* 6. $\mathcal{F}$ is the set of all BART regression functions. $\mathcal{B}_\epsilon^{(a)}$ is the space of the BART regression functions that are $\epsilon$-close to the ground truth of the $a$-th arm w.r.t. $\|\cdot\|_Q$. $\Theta_\epsilon^{(a)}$ is the space of the BART parameters corresponding to the functions in $\mathcal{B}_\epsilon^{(a)}$. With a slight abuse of notation, $\Pi(\cdot)$ will stand for the prior probability measure on either the model parameter space $\Theta$, or the BART parameter space $\Theta$, or the BART function space $\mathcal{F}$.

Use condition 1 for $n = T \to \infty$:

$$\epsilon_T = \sqrt{d \log p / T} + T^{-\alpha/(2\alpha+d)} \text{polylog}(T, d).$$

*Remark* 7. Here, $\epsilon_T$ is just a valid rate for the prior mass condition 1 used in the proof. It has nothing to do with the real fitting process and posterior.

**Lemma 13.** *For all sufficiently large $T$, on the event $\Theta_{\epsilon_T}$, the expected increment $\Delta L$ is small over the post-exploration rounds $s = \tau K + 1, \ldots, T$. In fact, fixing any $\boldsymbol{\theta} \in \Theta_{\epsilon_T}$,*

$$\mathbb{E}\left[\sum_{s=\tau K+1}^{T} \Delta L\left(\boldsymbol{\theta}, X_s, A_s, R_s\right)\right] \leq \eta K(T - \tau K)\epsilon_T^2 + \lambda(T - \tau K)\sqrt{K}\epsilon_T.$$

*Proof.* See Section E.6.2.

To bound the term involving $\Delta L$, we first lower bound the posterior mass of $\Theta_{\epsilon_T}$ after exploration.

**Lemma 14** (Posterior mass after exploration)**.** *Assume $b = 1$ in the FGTS loss (as in the feel-good TS theorem above), and that the model class is uniformly bounded: there exists $F_{\max} < \infty$ such that*

$$\sup_{\boldsymbol{\theta} \in \Theta} \sup_{(x,a) \in [0,1]^p \times \mathcal{A}} |f_{\boldsymbol{\theta}}(x, a)| \leq F_{\max}.$$

*A sufficient condition is (P2\*): with a fixed number $m$ of trees and leaf values supported on $[-\bar{C}_1, \bar{C}_1]$, one can take $F_{\max} = m\bar{C}_1$. Then for any measurable set $B \subset \Theta$,*

$$\Pi_{\tau K}(B) \geq \Pi(B) \exp\left(-\lambda \tau K - \eta \sum_{t=1}^{\tau K}(F_{\max} + |R_t|)^2\right).$$

*In particular, if $C_R \overset{def}{=} \sup_{t \leq \tau K} \mathbb{E}\big[(F_{\max} + |R_t|)^2\big] < \infty$ (e.g. when $f_0$ is bounded and the noise is conditionally sub-Gaussian), then*

$$\mathbb{E}[\log \Pi_{\tau K}(B)] \geq \log \Pi(B) - (\lambda + \eta C_R)\tau K.$$

*Proof.* See Section E.6.1.

Recall that $Z_{T,\tau K}$ is defined above. We can lower bound $Z_{T,\tau K}$ by restricting the expectation over $\boldsymbol{\theta}$ to the set $\Theta_{\epsilon_T}$:

$$
\begin{aligned}
Z_{T,\tau K} &\geq \mathbb{E}\left[\log \mathbb{E}_{\boldsymbol{\theta} \sim \Pi_{\tau K}}\left[\mathbb{I}(\boldsymbol{\theta} \in \Theta_{\epsilon_T})\exp\left(-\sum_{s=\tau K+1}^{T} \Delta L\left(\boldsymbol{\theta}, X_s, A_s, R_s\right)\right)\right]\right]\\
&\geq \mathbb{E}\left[\log\left(\Pi_{\tau K}(\Theta_{\epsilon_T})\mathbb{E}_{\boldsymbol{\theta} \sim \Pi_{\tau K}}\left[\exp\left(-\sum_{s=\tau K+1}^{T} \Delta L\left(\boldsymbol{\theta}, X_s, A_s, R_s\right)\right)\ \middle|\ \boldsymbol{\theta} \in \Theta_{\epsilon_T}\right]\right)\right]\\
&\geq \mathbb{E}\left[\log \Pi_{\tau K}(\Theta_{\epsilon_T})\right] - \sup_{\boldsymbol{\theta} \in \Theta_{\epsilon_T}} \mathbb{E}\left[\sum_{s=\tau K+1}^{T} \Delta L\left(\boldsymbol{\theta}, X_s, A_s, R_s\right)\right],
\end{aligned}
$$

where the last inequality used $\log \mathbb{E}[\exp(-Z)] \geq -\mathbb{E}[Z]$ and the fact that the bound holds uniformly over $\boldsymbol{\theta} \in \Theta_{\epsilon_T}$.

**Posterior mass term.** By Lemma 14, we have $\mathbb{E}[\log \Pi_{\tau K}(\Theta_{\epsilon_T})] \geq \log \Pi(\Theta_{\epsilon_T}) - (\lambda + \eta C_R)\tau K$. Moreover, since $\theta_a$ are *a priori* independently drawn from $\Theta$, we have $\log \Pi(\Theta_{\epsilon_T}) = \sum_{a \in \mathcal{A}} \log \Pi(\Theta_{\epsilon_T}^{(a)})$. From (1), $\log \Pi(\Theta_{\epsilon_T}^{(a)}) \geq -c_0 T \epsilon_T^2$ for each arm $a$. Combining these bounds and Lemma 13 to bound the supremum term over $\boldsymbol{\theta} \in \Theta_{\epsilon_T}$, we obtain for sufficiently large $T$,

$$Z_{T,\tau K} \geq -Kc_0 T\epsilon_T^2 - (\lambda + \eta C_R)\tau K - \eta KT\epsilon_T^2 - \lambda T\sqrt{K}\epsilon_T.$$

**Step 4 (Optimize $\lambda$ and choose $\tau(T)$).** Substituting this into the regret bound gives (absorbing lower-order $\tau$ terms),

$$\sum_{t=1}^{T} \mathbb{E}[\text{Regret}_t] \leq \tau K + \frac{\lambda K(T - \tau K)}{\eta} + 6\lambda(T - \tau K) - \frac{1}{\lambda} Z_{T,\tau K}$$

$$\leq \tau K + \lambda \left( \frac{K}{\eta} + 6 \right) T + \frac{1}{\lambda} \left( (c_0 + \eta)KT\epsilon_T^2 + (\lambda + \eta C_R)\tau K + \lambda T\sqrt{K}\epsilon_T \right)$$

$$= \tau K + \lambda \left( \frac{K}{\eta} + 6 \right) T + \frac{(c_0 + \eta)KT\epsilon_T^2}{\lambda} + T\sqrt{K}\epsilon_T + \frac{(\lambda + \eta C_R)\tau K}{\lambda}.$$

Let

$$A \stackrel{\text{def}}{=} \frac{K}{\eta} + 6, \qquad B \stackrel{\text{def}}{=} (c_0 + \eta)K.$$

We have

$$\sum_{t=1}^{T} \mathbb{E}[\text{Regret}_t] \leq 2\tau K + AT\lambda + \frac{BT\epsilon_T^2 + \eta C_R \tau K}{\lambda} + T\sqrt{K}\epsilon_T.$$

Optimizing the $\lambda$-dependent terms gives

$$\lambda^* = \sqrt{\frac{BT\epsilon_T^2 + \eta C_R \tau K}{AT}}.$$

Plugging this choice in yields

$$AT\lambda^* + \frac{BT\epsilon_T^2 + \eta C_R \tau K}{\lambda^*} = 2\sqrt{AT \left( BT\epsilon_T^2 + \eta C_R \tau K \right)}.$$

In particular, if $\tau(T) = o(T\epsilon_T^2)$ (so that $\eta C_R \tau K = o(BT\epsilon_T^2)$), then

$$2\sqrt{AT \left( BT\epsilon_T^2 + \eta C_R \tau K \right)} = 2T\epsilon_T \sqrt{AB} \, (1 + o(1)),$$

so $\lambda^* \asymp \epsilon_T$ and the leading regret term remains of order $KT\epsilon_T$ (for fixed $\eta$).

Thus, the total expected regret is bounded by

$$\sum_{t=1}^{T} \mathbb{E}[\text{Regret}_t] \leq 2T\epsilon_T \sqrt{\left( \frac{K}{\eta} + 6 \right)(c_0 + \eta)K} + T\sqrt{K}\epsilon_T + 2\tau K.$$

Again, since the first term is of order $KT\epsilon_T$ and dominates the second term $T\sqrt{K}\epsilon_T$, and $\tau(T) = o(T\epsilon_T^2)$, the two-stage explore-then-FGTS procedure attains the $O(KT\epsilon_T)$ regret bound.

*Remark* 8. The choice of $\tau(T)$ should also guarantee that under round-robin exploration each arm is sampled at least $\tau \to \infty$ times for condition 1 to hold.

**Step 5 (Plug in $\epsilon_T$).**

$$T\epsilon_T = \sqrt{dT \log p} + T^{(\alpha+d)/(2\alpha+d)} \text{polylog}(T, p).$$

Combining with the bound from Step 4 (whose leading term is of order $KT\epsilon_T$ for fixed $\eta$), we obtain the explicit two-term regret bound

$$\sum_{t=1}^{T} \mathbb{E}[\text{Regret}_t] \leq \mathcal{O}\left( K\left( \sqrt{dT \log p} + T^{(\alpha+d)/(2\alpha+d)} \text{polylog}(T, p) \right) \right),$$

and since $(\alpha + d)/(2\alpha + d) > 1/2$ for $\alpha > 0$, the second term dominates $\sqrt{dT \log p}$ asymptotically (up to logarithmic factors), yielding $\sum_{t=1}^{T} \mathbb{E}[\text{Regret}_t] = \mathcal{O}\left( KT^{(\alpha+d)/(2\alpha+d)} \text{polylog}(T, p) \right)$, which is the characteristic sparse-regime rate for nonparametric bandits with effective dimension $d$, up to logarithmic factors.

$\square$

### E.5. Corollary 15

**Corollary 15** (Anytime FGTS via doubling)**.** *In particular, there exists a schedule of parameters $\{\lambda_m\}_{m\geq 0}$ and a restart strategy over epochs $\{2^m\}_{m\geq 0}$ such that the resulting anytime FGTS algorithm satisfies*

$$\sum_{t=1}^{T} \mathbb{E}[\text{Regret}_t] \leq C\,K\,T^{(\alpha+d)/(2\alpha+d)}\,\text{polylog}(T,p)$$

*for all $T \geq 1$, where $C > 0$ is a constant depending only on $(\alpha, p, d, \eta)$ and BART hyperparameters. The proof follows from applying the FGTS regret bound on each epoch and a standard doubling-trick argument.*

*Proof.* The argument is a standard doubling-trick (restart) conversion (Lattimore & Szepesvári, 2020). For epoch length $H_m = 2^m$, set $\lambda_m \stackrel{\text{def}}{=} \lambda^*(H_m)$ and $\tau_m \stackrel{\text{def}}{=} \tau(H_m)$, i.e., the tuned parameters from the known-horizon bound. Let $\gamma = (\alpha + d)/(2\alpha + d) \in (0,1)$. For each $H \geq 1$, let $\mathcal{A}(H)$ denote the (known-horizon) explore-then-FGTS algorithm with parameters tuned for horizon $H$, so that

$$\mathbb{E}[\text{Regret}_{1:H}(\mathcal{A}(H))] \leq C_0\,K\,H^\gamma\,\text{polylog}(H,p)$$

for a constant $C_0 > 0$ independent of $H$.

Define epochs of lengths $H_m = 2^m$ for $m = 0, 1, 2, \ldots$. In epoch $m$, run $\mathcal{A}(H_m)$ for $H_m$ rounds and then restart (reinitialize) at the next epoch. For any horizon $T \geq 1$, let $M$ be such that $2^M \leq T < 2^{M+1}$. Since regret is nonnegative,

$$\mathbb{E}[\text{Regret}_{1:T}] \leq \sum_{m=0}^{M} \mathbb{E}[\text{Regret}_{\text{epoch } m}] \leq \sum_{m=0}^{M} C_0\,K\,2^{m\gamma}\,\text{polylog}(2^m, p).$$

Using $\text{polylog}(2^m, p) \leq \text{polylog}(T, p)$ and the geometric sum $\sum_{m=0}^{M} 2^{m\gamma} \leq \frac{2^{(M+1)\gamma}}{2^\gamma - 1} \leq \frac{2^\gamma}{2^\gamma - 1}\,T^\gamma$, we obtain

$$\mathbb{E}[\text{Regret}_{1:T}] \leq C\,K\,T^\gamma\,\text{polylog}(T, p)$$

for $C = C_0 \cdot \frac{2^\gamma}{2^\gamma - 1}$, completing the proof. $\qquad\square$

### E.6. Proofs of Auxiliary Lemmas

#### E.6.1. PROOF OF LEMMA 14

*Proof.* By definition, $\Pi_{\tau K}(d\boldsymbol{\theta}) \propto \exp\big(-\sum_{t=1}^{\tau K} L_{\text{FGTS}}(\boldsymbol{\theta}, X_t, A_t, R_t)\big)\,\Pi(d\boldsymbol{\theta})$.

For $b = 1$,

$$-L_{\text{FGTS}}(\boldsymbol{\theta}, X_t, A_t, R_t) = -\eta(f_{\boldsymbol{\theta}}(X_t, A_t) - R_t)^2 + \lambda \min\big(1, f_{\boldsymbol{\theta}}^*(X_t)\big) \leq \lambda,$$

since $(f_{\boldsymbol{\theta}} - R_t)^2 \geq 0$ and $\min(1, f_{\boldsymbol{\theta}}^*(X_t)) \leq 1$. Hence

$$\mathbb{E}_{\boldsymbol{\theta}\sim\Pi} \exp\Big(-\sum_{t=1}^{\tau K} L_{\text{FGTS}}(\boldsymbol{\theta}, X_t, A_t, R_t)\Big) \leq \exp(\lambda \tau K).$$

On the other hand, since $\lambda \min(1, f_{\boldsymbol{\theta}}^*(X_t)) \geq 0$,

$$\exp\Big(-\sum_{t=1}^{\tau K} L_{\text{FGTS}}(\boldsymbol{\theta}, X_t, A_t, R_t)\Big) \geq \exp\Big(-\eta\sum_{t=1}^{\tau K}(f_{\boldsymbol{\theta}}(X_t, A_t) - R_t)^2\Big).$$

Using $|f_{\boldsymbol{\theta}}(x,a)| \leq F_{\max}$, we have $(f_{\boldsymbol{\theta}}(X_t, A_t) - R_t)^2 \leq (F_{\max} + |R_t|)^2$, and therefore

$$\int_B \exp\Big(-\sum_{t=1}^{\tau K} L_{\text{FGTS}}(\boldsymbol{\theta}, X_t, A_t, R_t)\Big)\Pi(d\boldsymbol{\theta}) \geq \Pi(B)\exp\Big(-\eta\sum_{t=1}^{\tau K}(F_{\max} + |R_t|)^2\Big).$$

Combining the numerator and denominator bounds yields

$$\Pi_{\tau K}(B) \geq \Pi(B)\exp\Bigg(-\lambda\tau K - \eta\sum_{t=1}^{\tau K}(F_{\max} + |R_t|)^2\Bigg).$$

Taking logs and expectations gives $\mathbb{E}[\log \Pi_{\tau K}(B)] \geq \log \Pi(B) - (\lambda + \eta C_R)\tau K$. $\qquad\square$

E.6.2. PROOF OF LEMMA 13

*Proof.* Fix an per-arm exploration length $\tau$ as in the main proof and consider the post-exploration rounds $t = \tau K + 1, \ldots, T$.

Define

$$\delta_{\boldsymbol{\theta}}(x, a) = f_{\boldsymbol{\theta}}(x, a) - f_0(x, a).$$

Decomposing

$$
\eta \left( (f_{\boldsymbol{\theta}} - r)^2 - (f_0 - r)^2 \right) = \eta \left( (\delta_{\boldsymbol{\theta}} + f_0 - r)^2 - (f_0 - r)^2 \right) \\
= \eta \left( \delta_{\boldsymbol{\theta}}^2 + 2\delta_{\boldsymbol{\theta}}(f_0 - r) \right),
$$

we obtain

$$\Delta L = \eta \left( \delta_{\boldsymbol{\theta}}^2 + 2\delta_{\boldsymbol{\theta}}(f_0 - r) \right) - \lambda \big( \min(1, f_{\boldsymbol{\theta}}^*) - f_0^* \big),$$

$$
\sum_{t=\tau K+1}^{T} \Delta L\left( \boldsymbol{\theta}, X_t, A_t, R_t \right) = \eta \underbrace{\sum_{t=\tau K+1}^{T} \delta_{\boldsymbol{\theta}}(X_t, A_t)^2}_{\text{Part I}} + 2\eta \underbrace{\sum_{t=\tau K+1}^{T} \delta_{\boldsymbol{\theta}}(X_t, A_t)\varepsilon_t}_{\text{Part II}} + \lambda \underbrace{\sum_{t=\tau K+1}^{T} \left( f_0^*(X_t) - \min(1, f_{\boldsymbol{\theta}}^*(X_t)) \right)}_{\text{Part III}}.
$$

**(i) Squared error term.**

Since $\boldsymbol{\theta} \in \Theta_{\epsilon_T}$ implies $\|\delta_{\boldsymbol{\theta}}(\cdot, a)\|_Q \leq \epsilon_T$ for all $a$, we have

$$\mathbb{E}_{X \sim Q}\left[ \delta_{\boldsymbol{\theta}}(X, a)^2 \right] \leq \epsilon_T^2, \qquad \forall a \in \mathcal{A}.$$

Using $\sum_a \mathbb{P}(A_t = a \mid X_t, \mathcal{H}_{t-1}) = 1$,

$$
\mathbb{E}\left[ \sum_{t=\tau K+1}^{T} \delta_{\boldsymbol{\theta}}(X_t, A_t)^2 \right] = \sum_{t=\tau K+1}^{T} \mathbb{E}\left[ \sum_{a \in \mathcal{A}} \mathbb{P}(A_t = a \mid X_t, \mathcal{H}_{t-1})\, \delta_{\boldsymbol{\theta}}(X_t, a)^2 \right]
$$

$$
\leq \sum_{t=\tau K+1}^{T} \mathbb{E}\left[ \sum_{a \in \mathcal{A}} \delta_{\boldsymbol{\theta}}(X_t, a)^2 \right] = (T - \tau K) \sum_{a \in \mathcal{A}} \mathbb{E}_{X \sim Q}\left[ \delta_{\boldsymbol{\theta}}(X, a)^2 \right]
$$

$$
\leq K(T - \tau K)\epsilon_T^2.
$$

**(ii) Cross term.**

Assume the reward noise satisfies $\mathbb{E}[\varepsilon_t \mid \mathcal{H}_{t-1}, X_t, A_t] = 0$. Then, by the tower property,

$$
\mathbb{E}\left[ \sum_{t=\tau K+1}^{T} \delta_{\boldsymbol{\theta}}(X_t, A_t)\, \varepsilon_t \right] = \sum_{t=\tau K+1}^{T} \mathbb{E}\left[ \mathbb{E}\left[ \delta_{\boldsymbol{\theta}}(X_t, A_t)\, \varepsilon_t \mid \mathcal{H}_{t-1}, X_t, A_t \right] \right]
$$

$$
= \sum_{t=\tau K+1}^{T} \mathbb{E}\left[ \delta_{\boldsymbol{\theta}}(X_t, A_t)\, \mathbb{E}[\varepsilon_t \mid \mathcal{H}_{t-1}, X_t, A_t] \right] = 0.
$$

**(iii) Feel-good term.**

At time $t$,

$$
\Delta_t(\boldsymbol{\theta}) \overset{\text{def}}{=} f_0^*(X_t) - \min\{1, f_{\boldsymbol{\theta}}^*(X_t)\}
$$

$$
\leq \max\{0, f_0^*(X_t) - f_{\boldsymbol{\theta}}^*(X_t)\}
$$

$$
\leq \max_{a \in \mathcal{A}} \left| f_{\boldsymbol{\theta}}(X_t, a) - f_0(X_t, a) \right| = \max_{a \in \mathcal{A}} \left| \delta_{\boldsymbol{\theta}}(X_t, a) \right|.
$$

Noting $\max_a |u_a| \leq \sqrt{\sum_a u_a^2}$,

$$\Delta_t(\boldsymbol{\theta}) \leq \sqrt{\sum_{a \in \mathcal{A}} \delta_{\boldsymbol{\theta}}(X_t, a)^2}.$$

Taking expectation over the random-design contexts and using concavity of $z \mapsto \sqrt{z}$,

$$\mathbb{E}_{X \sim Q}\big[\Delta_t(\boldsymbol{\theta})\big] \leq \mathbb{E}_{X \sim Q}\left[\sqrt{\sum_{a \in \mathcal{A}} \delta_{\boldsymbol{\theta}}(X, a)^2}\right] \leq \sqrt{\sum_{a \in \mathcal{A}} \mathbb{E}_{X \sim Q}\left[\delta_{\boldsymbol{\theta}}(X, a)^2\right]} \leq \sqrt{K}\, \epsilon_T,$$

and therefore $\mathbb{E}\big[\sum_{t=\tau K+1}^{T} \Delta_t(\boldsymbol{\theta})\big] \leq (T - \tau K)\sqrt{K}\, \epsilon_T$.

**Conclusion.**

Combining these three pieces gives, for every $\boldsymbol{\theta} \in \boldsymbol{\Theta}_{\epsilon_T}$,

$$\mathbb{E}\left[\sum_{t=\tau K+1}^{T} \Delta L\left(\boldsymbol{\theta}, X_t, A_t, R_t\right)\right] \leq \eta\, K(T - \tau K)\epsilon_T^2 + 0 + \lambda\,(T - \tau K)\sqrt{K}\epsilon_T.$$

$\square$

### E.7. Verification of Condition 1

Our regret proof uses only one genuinely Bayesian-nonparametric input: the *prior small-ball* (prior mass) bound in Condition 1. We verify it by invoking the random-design split-net BART theory of Jeong & Ročková (2023) and extracting the corresponding prior concentration bound.

**Split-nets (Definitions 5–6 in Jeong & Ročková (2023)).** A *split-net* is a fixed (data-independent) set of candidate split locations

$$\mathcal{Z} = \{z_i \in [0,1]^p : i = 1, \ldots, b_n\}.$$

For each coordinate $j$, write $[\mathcal{Z}]_j = \{z_{ij} : i = 1, \ldots, b_n\}$ and let $b_j(\mathcal{Z}) = |[\mathcal{Z}]_j|$ be the number of *unique* split-candidates along $j$. A $\mathcal{Z}$-*tree partition* is a tree partition in which every split-point is chosen from $[\mathcal{Z}]_j \cap \operatorname{int}([\Omega]_j)$ when splitting a current box $\Omega$ along coordinate $j$.

Fix an arm $a$ and write $g_{0,a}(x) = f_0(x, a)$. Consider the random-design regression model

$$Y_i = g_{0,a}(X_i) + \varepsilon_i, \qquad X_i \sim Q, \qquad \varepsilon_i \sim N(0, \sigma_0^2),$$

where $\operatorname{supp}(Q) \subseteq [0,1]^p$ and $Q$ admits a bounded density (their Section 6.1).

We assume that $g_{0,a}$ and the split-net $\mathcal{Z}$ satisfy Assumptions **(A1)**, **(A2)**, **(A3\*)**, **(A4)**, **(A5)**, **(A6\*)**, and **(A7)** of Jeong & Ročková (2023). Concretely:

- **(A1) Truth class.** $g_{0,a}$ belongs to the sparse piecewise heterogeneous anisotropic Hölder class $\Gamma_\lambda^{A_{\bar{\alpha}}, d, p}(\mathfrak{X}_0)$ (optionally intersected with $\mathcal{C}([0,1]^p)$) as in Jeong & Ročková (2023, (A1)). A sufficient condition would be to assume the sparse-isotropic continuous specialization: for each arm, $g_{0,a}(x) = h_{0,a}(x_{S_{0,a}})$ with $|S_{0,a}| \leq d$ and $h_{0,a}$ $\alpha$-Hölder on $[0,1]^d$.

- **(A2) Rate regime.** The target rate $\epsilon_n$ in Jeong & Ročková (2023, eq. (7)) satisfies $\epsilon_n \ll 1$.

- **(A3\*) Bounded truth.** $\|g_{0,a}\|_\infty \leq C_0^*$ for some sufficiently large $C_0^* > 0$.

- **(A4) Noise scale.** $\sigma_0^2 \in [C_0^{-1}, C_0]$ for a sufficiently large $C_0 > 1$.

- **(A5) Split-net size control.** $\max_{1 \leq j \leq p} \log b_j(\mathcal{Z}) \lesssim \log n$.

- **(A6\*) Integrated $L_2$ approximation.** $\mathcal{Z}$ is suitably dense and regular so that there exist a $\mathcal{Z}$-tree partition $\widehat{\mathcal{T}}$ and $\hat{g}_{0,a} \in \mathcal{F}_{\widehat{\mathcal{T}}}$ with $\|g_{0,a} - \hat{g}_{0,a}\|_2 \lesssim \bar{\epsilon}_n$ (their Theorem 1 / Jeong & Ročková (2023, (A6\*))).

- **(A7) Depth control.** The approximating $\mathcal{Z}$-tree partition has maximal depth $\lesssim \log n$ (Jeong & Ročková (2023, (A7))).

We place the split-net forest prior assigned through **(P1)**, **(P2\*)**, and **(P3\*)** of Jeong & Ročková (2023):

- **(P1) Tree prior with Dirichlet sparsity.** For a fixed number of trees, each tree is independently assigned the split-net tree prior with Dirichlet-sparse splitting proportions (their eq. (3)) and the exponentially decaying split probability by depth (their Section 3.1).

- **(P2\*) Truncated step-heights (leaf values).** A prior supported on $[-\bar{C}_1, \bar{C}_1]$ is assigned to the step-heights for some $\bar{C}_1 > C_0^*$ (Jeong & Ročková (2023, (P2\*))).

- **(P3\*) Truncated noise variance.** A prior supported on $[\bar{C}_2^{-1}, \bar{C}_2]$ is assigned to $\sigma^2$ for some $\bar{C}_2 > C_0$ (Jeong & Ročková (2023, (P3\*))).

**Sufficient conditions for the split-net assumptions.** In a random design, $\mathcal{Z}$ must be deterministic (independent of the observed covariates), as emphasized in Jeong & Ročková (2023, Section 6.1). A convenient sufficient choice is the regular grid split-net of Jeong & Ročková (2023, Section 4.3.1):

$$\mathcal{Z} = \left\{ \frac{i - \frac{1}{2}}{m} : i = 1, \ldots, m \right\}^p, \qquad m = n^c, \ c \geq 1.$$

Then $b_j(\mathcal{Z}) = m$ for all $j$, so (A5) holds. Moreover, Jeong & Ročková (2023, Lem. 1) (see also Definitions 8 and 10 and Remark 9) provides mild sufficient conditions under which such regular grids are dense/regular enough to guarantee (A6\*) and (A7).

**Conclusion (prior small-ball / prior mass).** Under Assumptions (A1), (A2), (A3\*), (A4), (A5), (A6\*), (A7) and priors (P1), (P2\*), (P3\*), Jeong & Ročková (2023, Thm. 4) establishes $L_2(Q)$ posterior contraction at rate $\epsilon_n$. In particular, their proof verifies the required prior concentration: there exists $\bar{c} > 0$ such that for all sufficiently large $n$,

$$\Pi\left( \left\{ g_\theta : \|g_\theta - g_{0,a}\|_Q \leq \epsilon_n \right\} \right) \geq \exp(-\bar{c}\, n\, \epsilon_n^2).$$

Since arms are *a priori* independent under our product prior, the same bound holds for each arm $a \in \mathcal{A}$, which yields Condition 1.

### E.8. FGTS Sensitivity and Practical Takeaway

We study the tuning parameters of FGTS and report results aggregated via normalized regret (final regret divided by the best mean regret within each dataset). The exploration intensity parameter $\eta$ (`feel_good_eta`) has a strong effect on regret: aggressive values can increase normalized regret substantially (means 8.496 at 0.2 and 4.694 at 0.5), while the default setting performs well (mean 1.014). For $\lambda$ (`feel_good_lambda`), conditioning on the default $\eta$, the heatmap shows a smaller effect: mean normalized regret is 4.645 at $\lambda = 0$, 4.696 at $\lambda = 0.01$, and 4.863 at $\lambda = 0.05$.

Given the sensitivity to tuning and lack of consistent improvements, we do not recommend the feel-good variant as a default in practice; it is included here to complement the main Bayesian analysis of BFTS.

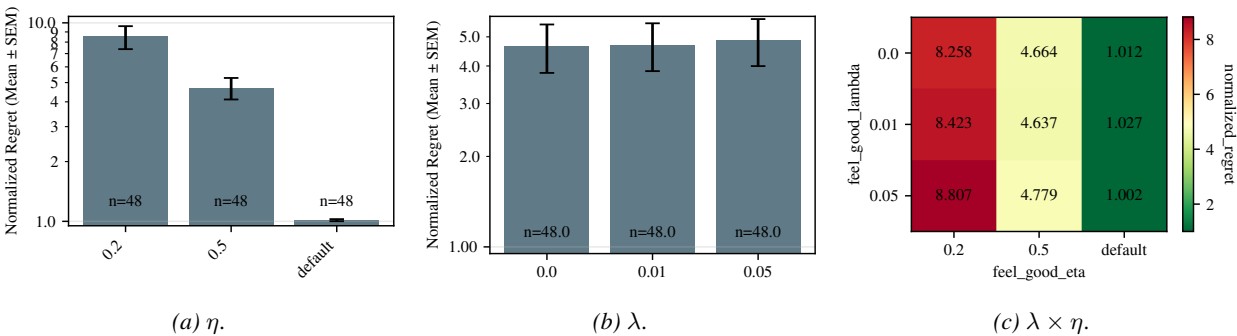

*Figure 13.* Sensitivity of FGTS to $\eta$ and $\lambda$ (aggregated via normalized regret; lower is better).

