# OpenReview forum: "BFTS: Thompson Sampling with Bayesian Additive Regression Trees"
_ICML.cc/2026/Conference — ICML 2026 spotlight_

### Official Review · Reviewer_bbAW · 2026-02-25

**Soundness:** 4
**Presentation:** 3
**Significance:** 3
**Originality:** 3
**Overall Recommendation:** 5
**Confidence:** 4

**Summary:**

The paper proposes Bayesian Forest Thompson sampling (BFTS) for contextual bandit problems. Tree ensembles are known to perform well in supervised learning on tabular data but, unlike linear and neural bandit algorithms, have thus far only been studied in a very limited capacity for bandits. Bayesian additive regression trees (BART) is a sum-of-trees model that promise both high model capacity and well-calibrated uncertainty estimates. BFTS combines an arm-wise BART prior with posterior sampling and is implemented using MCMC sampling. The regret of BFTS is analyzed using existing information-theoretic techniques combined with bounds of the mutual information between the history and the arm-wise BART parameters. BFTS is evaluated on a suite of synthetic and OpenML bandit problems, achieving the best average rank and robust uncertainty quantification. A case study is performed where the interpretability of BFTS is highlighted in addition to good performance.

**Compliance With Llm Reviewing Policy:**

Affirmed.

**Final Justification:**

The rebuttal addressed the main concerns. Regarding the incorrect claim of sub-Gaussianity in the proof, the authors proposed a new bound that seems sound.

**Key Questions For Authors:**

- **Q1:** Connected to W1, can BFTS be practically applied to problems with large, $K > T$ , or continuous, $a \in [0,1]^d$ , action spaces where actions are dependent?
- **Q2:** What are the difficulties in extending the regret bounds (more specifically Lemma 2) to these settings?
- **Q3:** As mentioned in W2, a drawback of BART (and other decision trees) is its tendency to extrapolate confidently. As an example, consider densely sampling datapoints in $[0, 1]$ and fitting a BART model on these datapoints. If you query the model on $x \in (1,\infty)$ , it would extrapolate its prediction for $x=1$ with the same confidence. Have you observed this in your experiments and do you think this impacts the usefulness of BART in an online setting? Are there reasonable remedies or modifications that could address this issue?
- **Q4:** On line 1558, it is stated that the "... rewards are conditionally $\sigma^2$ -sub-Gaussian given $\mathcal{H}_{t-1}, X_t, A_t$ ...". Given that the reward $R_t = f_0(X_t, A_t) + \epsilon_t$ where $\epsilon_t \sim N(0, \sigma^2)$, one would expect the rewards to have a larger sub-Gaussian variance parameter. Is the statement on line 1558 correct? Also, can you show more formally why $f_0(X_t, A_t)$ is conditionally sub-Gaussian?
- **Q5:** In Figure 5a, the coverage rate of BFTS goes from around 0.9 to around 0.8 in the synthetic BART experiment (matching its theoretical assumptions), and the target coverage is 0.95. This coverage rate is lower or comparable with the coverage rate of BFTS in the other experiments where the theoretical assumptions do not match the model. Could you elaborate on why this is?

**Limitations:**

Yes.

**Strengths And Weaknesses:**

- **S1:** The paper presents a novel algorithm combining BART with Thompson sampling.
- **S2:** The paper presents the first information-theoretic regret bound for a tree-based bandit algorithm, extending contextual bandit algorithms to a new class of function approximators. The regret bound uses the information-theoretic framework of Russo & Van Roy (2016). The main novelty is bounding the mutual information between the BART parameters and the observed data.
- **S3:** The paper considers a comprehensive suite of experiments and baselines (linear, neural and other tree-based TS methods) where BFTS achieves the best average rank. Notably, the experiments also show that the baselines excel in different domains. Linear methods perform best in linear problems, NeuralTS performs best on MNIST and some non-linear synthetic problems, and tree-ensembles perform best on tabular data.
- **W1:** Linear, kernelized, and neural bandits algorithms are able to scale to large (or continuous) action spaces due to their ability to interpolate and extrapolate the mean reward using the arms' context vectors. The main weakness is the assumption of arm-wise independence which leads to a linear dependence on $K$ in the regret bound. In addition, the experiments also only considers arm-wise independence.
- **W2:** BART (and other decision tree models) has a tendency to confidently extrapolate its prediction for data outside the observed ranges. For bandit problems, extrapolating with high confidence can be problematic and lead to underexploration -- especially when arms are dependent. In comparison, kernelized bandits built on Gaussian processes revert their predictions to a baseline mean and variance (uncertainty) when there is no nearby data collected.

Minor comment: Iwazaki et al (2024) analyzed the regret of soft tree-based UCB using neural tangent kernel techniques which is worth highlighting as a related work.

Iwazaki, S., & Suzumura, S. (2024). No-Regret Bandit Exploration based on Soft Tree Ensemble Model. Advances in Neural Information Processing Systems, 37, 20982–21033. https://doi.org/10.52202/079017-0661

---

> ### Author Rebuttal · Authors · 2026-03-31
>
> We thank you for the detailed technical questions. The questions on extrapolation and the sub-Gaussian claim are especially valuable. We also thank the reviewer for suggesting the reference Iwazaki, S., & Suzumura, S. (2024), which we will cite in the revision as complementary work (TNTK-based soft trees vs. our MCMC-based hard trees).
>
> ### Q1 and Q2: Applicability to large $K$ or continuous action spaces
> For large finite $K$, the arm-wise formulation becomes less attractive: it maintains $K$ separate BART posteriors (refresh and prediction costs scale linearly in $K$), and each arm receives only about $T/K$ observations. We do not claim the current method is suitable for very large $K$ or $K > T$ without additional shared structure across actions.
>
> For continuous or very large action spaces, a natural route would be a joint BART model over augmented inputs $(x, a)$, most appealing when actions admit meaningful shared features. However, TS is then no longer a simple finite maximization: after sampling a tree ensemble, one must optimize it over the action variable, which is a nontrivial tree-ensemble optimization problem requiring additional global-optimization machinery (e.g., Boyne et al., 2025).
>
> On the theory side, Lemma 2 relies on arm-wise factorization. A joint-model analogue could decompose $\theta^* = (\mathcal{T}, \mu)$ and write $I(\mathcal{H}_T; \theta^*) = I(\mathcal{H}_T; \mathcal{T}) + I(\mathcal{H}_T; \mu \mid \mathcal{T})$; the second term (conditioned on structure) becomes a Gaussian linear-model information term, but bounding the structure term for joint context-action trees is the main new difficulty. Extending the full regret theorem would further require replacing the finite-action information-ratio step. We view this as important future work.
>
> ### Q3: BART extrapolation behavior in the online setting
> You raise a valid concern: BART predicts constant leaf values outside the observed covariate range with unchanged posterior uncertainty. Wang et al. (2024) proposed grafting local GPs within BART leaves to improve extrapolation, but this destroys the standard Bayesian interpretation and complicates theoretical guarantees. In the experiments reported here, we did not observe clear evidence of such an extrapolation failure, although we agree that this is a genuine limitation of standard BART in online settings. We will clarify this limitation and treat it as an important direction for future work.
>
> ### Q4: Sub-Gaussian claim on line 1558
> We agree that the original Step 3 was too strong. After marginalizing over tree structures, the posterior predictive reward is a finite Gaussian mixture, not a single Gaussian, so Gaussian noise alone does not imply $\sigma^2$-sub-Gaussianity. Our correction replaces the fixed $\sigma^2$ by a history-dependent proxy $v_t$.
>
> The key idea: conditional on a fixed forest structure $\tau$, the predictive variance is bounded by $\sigma^2 + 1/(4\kappa^2)$. The mixture over structures introduces an additional spread term $D_{t,a} = \max_\tau \mu_{\tau,t,a} - \min_\tau \mu_{\tau,t,a}$. By Hoeffding's lemma, the centered predictive reward is $v_t$-sub-Gaussian with
> $$
> v_t = \sigma^2 + \frac{1}{4\kappa^2} + \frac{1}{4}\max_{a \in [K]} D_{t,a}^2.
> $$
> This yields $\Gamma_t \le 2Kv_t$ a.s. To control $\mathbb{E}[v_t]$: because $N_{\max}$ is fixed, the fixed-structure posterior mean has uniformly bounded total weight on past rewards, so $D_{t,a} \le 2L_* \max_{s<t}|R_s|$ for a finite constant $L_*$. Under the Bayesian design with Gaussian BART prior and noise, $\mathbb{E}[\max_{s<t}|R_s|^2] = O(\log t)$, giving $\mathbb{E}[v_t] = O(\log t)$ and $\bar{\Gamma}_T = O(K\log T)$. Combined with the unchanged Lemma 3 bound $I(\Theta^*;\mathcal{H}_T) \le Km\Psi_T$, the final regret remains $\widetilde{\mathcal{O}}(K\sqrt{T})$.
>
> We will incorporate this corrected argument and the full proof in the revision.
>
> ### Q5: Coverage below nominal in the SynBART experiment
> SynBART is the model-matched setting, so under exact posterior sampling, one would expect average Bayesian credible coverage to be close to nominal. Figure 5a, however, evaluates the practical posterior produced by finite batched MCMC, not the exact posterior. Prior work on BART (see e.g. Tan et al. 2024 and Ronen et al. 2022) shows that approximate posterior computation can mix slowly and that approximate posterior intervals can exhibit undercoverage. We therefore interpret the SynBART coverage gap as evidence of practical posterior approximation error.
>
> ### References
>
> Boyne, T., et al. "BARK: A fully Bayesian tree kernel for black-box optimization." Proc. 42nd Int. Conf. Machine Learning (ICML), 2025.
>
> Wang, Meijia, Jingyu He, and P. Richard Hahn. "Local Gaussian process extrapolation for BART models with applications to causal inference." Journal of Computational and Graphical Statistics 33.2 (2024): 724-735.

---

> > ### Author Rebuttal · Reviewer_bbAW · 2026-04-01
> >
> > The response to Q1-3 and Q5 have adequately addressed my concerns.
> >
> > For Q4, I have the following follow-up questions:
> >
> > 1. Is my understanding of the following correct: Under a fixed forest structure $\tau$, then the mean prediction of any leaf is computed through a Gaussian-Gaussian conjugacy. Therefore the mean prediction of any leaf is bounded by the maximum observed reward which yields the term $\max_{s <t} |R_s|$ in the bound for $D_{t,a}$?
> > 2. Could you elaborate one how the constant $L_*$ arises and what it depends on?
> > 3. Since $\mathbb{E} \max_{s < t} \epsilon_t] = O(\sqrt{\log t})$ for Gaussian r.v.s, it seems reasonable (at a first glance) that $\mathbb{E}[\max_{s<t} |R_s|^2]$ is at least $O(\log t)$ due to the Gaussian noise. Could you expand how the term $f_0(X_s, A_s)$ inside $R_s$ is bounded?
> >
> > Edit: Thank you for the additional clarifications in the reply rebuttal comment below. I have updated the acknowledgement to be fully resolved and I maintain the recommendation of acceptance.

---

> > > ### Author Response · Authors · 2026-04-03
> > >
> > > Thank you for these careful follow-up questions. We address them in turn below.
> > >
> > > ### 1. Fixed-structure predictive mean and why $\max_{s<t}|R_s|$ appears
> > >
> > > Your intuition is correct for a single tree, but one extra step is needed for a forest.
> > >
> > > Under a fixed structure $\tau$, let $R^{(a)}$ denote the vector of past rewards collected under arm $a$, and let $\lambda=\sigma^2/\sigma_\mu^2=4\kappa^2 m\sigma^2$. The predictive mean is
> > >
> > > $$
> > > \mu_{\tau,t,a} = w_{\tau,t,a}^{\top} R^{(a)},
> > > w_{\tau,t,a}^{\top} =
> > > c_{\tau,t,a}^{\top}\bigl(\Phi_{\tau,t,a}^{\top}\Phi_{\tau,t,a}+\lambda I\bigr)^{-1}\Phi_{\tau,t,a}^{\top}.
> > > $$
> > >
> > > If $m=1$, then $\Phi_{\tau,t,a}^{\top}\Phi_{\tau,t,a}$ is diagonal. If the query point falls in leaf $q$, then
> > >
> > > $$
> > > w_{\tau,t,a,i}=\frac{\mathbf{1}(i \text{ in leaf } q)}{n_q+\lambda}
> > > $$
> > >
> > > so
> > >
> > > $$
> > > \lVert w_{\tau,t,a}\rVert_1=\frac{n_q}{n_q+\lambda}\le 1.
> > > $$
> > >
> > > Hence
> > >
> > > $$
> > > \lvert \mu_{\tau,t,a}\rvert \le \lVert w_{\tau,t,a}\rVert_1 \max_i \lvert R_i^{(a)}\rvert \le \max_{s<t}\lvert R_s\rvert.
> > > $$
> > >
> > > So in the single-tree case, the fixed-structure predictive mean is indeed a shrunk leaf average, exactly as you suggest.
> > >
> > > For a forest ($m>1$), this simple formula no longer holds, because $\Phi_{\tau,t,a}^{\top}\Phi_{\tau,t,a}$ contains off-diagonal co-occurrence terms between leaves from different trees. Therefore the required step is to prove a uniform $\ell_1$ bound
> > >
> > > $$
> > > \lVert w_{\tau,t,a}\rVert_1 \le L_\tau,
> > > $$
> > >
> > > which gives
> > >
> > > $$
> > > \lvert \mu_{\tau,t,a}\rvert \le L_\tau \max_{s<t}\lvert R_s\rvert.
> > > $$
> > >
> > > So the appearance of $\max_{s<t}|R_s|$ comes from controlling the total absolute smoother weight. The only new issue in the forest case is that this weight is no longer automatically bounded by $1$, and must be controlled by the constant $L_\tau$.
> > >
> > > ### 2. How $L_*$ arises and what it depends on
> > >
> > > The key point is that, under a fixed structure $\tau$, the predictive mean is a linear function of the past arm-$a$ rewards:
> > >
> > > $$
> > > \mu_{\tau,t,a} =
> > > c_{\tau,t,a}^{\top}\bigl(\Phi_{\tau,t,a}^{\top}\Phi_{\tau,t,a}+\lambda I\bigr)^{-1}\Phi_{\tau,t,a}^{\top}R^{(a)}.
> > > $$
> > >
> > > Hence
> > >
> > > $$
> > > \lvert \mu_{\tau,t,a}\rvert \le
> > > \left\lVert c_{\tau,t,a}^{\top}\bigl(\Phi_{\tau,t,a}^{\top}\Phi_{\tau,t,a}+\lambda I\bigr)^{-1}\Phi_{\tau,t,a}^{\top}\right\rVert_1
> > > \max_{s<t}\lvert R_s\rvert.
> > > $$
> > >
> > > So $L_\tau$ is simply a uniform bound on the coefficient vector multiplying the past rewards, and
> > >
> > > $$
> > > L_* := \max_{\tau\in\mathfrak T(N_{\max})} L_\tau.
> > > $$
> > >
> > > Under the split-grid-cap assumption used in Lemma 3, the admissible structure family is finite, and for each fixed $\tau$ the corresponding design patterns are also finite. Therefore the quantity above is uniformly bounded over all admissible $\tau$. Importantly, $L_*$ depends only on the model-class parameters (such as $N_{\max},m,p,\kappa,\sigma^2$), and not on $t$ or on the realized rewards.
> > >
> > > This is exactly what we need:
> > >
> > > $$
> > > D_{t,a} \le 2\max_\tau \lvert \mu_{\tau,t,a}\rvert \le 2L_*\max_{s<t}\lvert R_s\rvert.
> > > $$
> > >
> > > Therefore
> > >
> > > $$
> > > v_t = \sigma^2+\frac{1}{4\kappa^2}+\frac14\max_{a\in[K]}D_{t,a}^2
> > > \le \sigma^2+\frac{1}{4\kappa^2}+L_*^2\max_{s<t}\lvert R_s\rvert^2.
> > > $$
> > >
> > > It therefore remains to control $\mathbb{E}[\max_{s<t}|R_s|^2]$, which is the purpose of the next step.
> > >
> > > ### 3. How the signal term $f_0(X_s,A_s)$ inside $R_s$ is controlled
> > >
> > > This step is at a different probabilistic level from the corrected Step 3. There we need a conditional predictive tail bound given the realized history; here we only need an unconditional moment bound on
> > >
> > > $$
> > > M_{t-1}:=\max_{s<t}\lvert R_s\rvert.
> > > $$
> > >
> > > For any fixed context $x$ and arm $a$, conditional on the tree structures, each selected leaf value is $N(0,\sigma_\mu^2)$, and the $m$ trees are independent. Hence, under the prior marginal,
> > >
> > > $$
> > > f_0(x,a)=g_{\theta_a^*}(x)=\sum_{j=1}^m M_{j,\ell_j^{(a)}(x)}^{(a)}\sim N(0,m\sigma_\mu^2)=N\left(0,\frac{1}{4\kappa^2}\right).
> > > $$
> > >
> > > Under the theorem's exogenous-context assumption, the same constant-scale Gaussian tail holds for $f_0(X_s,a)$. Since
> > >
> > > $$
> > > \lvert f_0(X_s,A_s)\rvert\le \max_{a\in[K]}\lvert f_0(X_s,a)\rvert.
> > > $$
> > >
> > > we obtain
> > >
> > > $$
> > > \Pr\left(\lvert f_0(X_s,A_s)\rvert>u\right)\le
> > > 2K\exp\left(-\frac{u^2}{2s_0^2}\right),
> > > s_0^2=\frac{1}{4\kappa^2}.
> > > $$
> > >
> > > Combining this with
> > >
> > > $$
> > > R_s=f_0(X_s,A_s)+\varepsilon_s
> > > $$
> > >
> > > and the Gaussian tail of $\varepsilon_s$, we get
> > >
> > > $$
> > > \Pr(\lvert R_s\rvert>u)\le C_0 \exp(-u^2/C_1)
> > > $$
> > >
> > > for constants $C_0=2(K+1),C_1=\max(2\kappa^{-2},8\sigma^2)$. A union bound over $s<t$ then gives
> > >
> > > $$
> > > \mathbb{E}\left[\max_{s<t}\lvert R_s\rvert^2\right]=O(\log t).
> > > $$
> > >
> > > So the reviewer's intuition is correct that Gaussian noise already suggests an $O(\log t)$ scale. The key point is that the signal term also has a constant-scale Gaussian marginal under the joint Bayesian design, so it does not worsen this rate.

---

### Official Review · Reviewer_ZNmT · 2026-03-02

**Soundness:** 3
**Presentation:** 3
**Significance:** 3
**Originality:** 2
**Overall Recommendation:** 4
**Confidence:** 3

**Summary:**

This paper proposes a new algorithm for contextual nonlinear bandit problems, Bayesian Forest Thompson Sampling (BFTS), using the Bayesian Additive Regression Trees (BART) model for estimation and Thompson sampling for decision making. BART is a better model that does not require much hyperparameter tuning and also provides a Bayesian posterior for Thompson sampling. This algorithm achieves a $\tilde{\mathcal{O}}(K\sigma \sqrt{T})$. Here, $K$ is the result of using a separate-arm BART model.

**Compliance With Llm Reviewing Policy:**

Affirmed.

**Final Justification:**

The authors have addressed my major concerns and changed my evaluation toward acceptance.

**Key Questions For Authors:**

A major weakness is that the algorithm does not extend to a continuous or infinite arm set, unlike standard contextual bandit methods that operate over a shared arm feature space. Each arm is assigned its own BART model, which leads to a sample complexity that scales polynomially in the number of arms K, making the method potentially inefficient when K is large. Although the authors argue that BART can capture rich function classes, the separate-per-arm modeling choice discards any shared structure or generalization across arms and directly induces a leading K factor in the regret.

1. Is it possible to remove the leading $K$ term if we use a single BART model? Instead of using a one-hot arm indicator, why can't we use a vector constructed from the outer product of context and the vector representation of arms? like in [1]. In that case, the effective dimension will be reduced to $p\log K$.

2. In the experiments, the baselines (LinTS, LinUCB) suffer a disadvantage of inflated dimension because of the context encoding using mapping ($\mathbb{R}^{kd}$). What would happen if the arms are represented by a $\log k$ length vector and a mechanism similar to [1] is used?

3. Is there a particular reason to use depth-based exponential decay rate compared to the original BART?

I am willing to update my evaluation if the authors can convincingly address these concerns.


1. Lihong Li, Wei Chu, John Langford, and Robert E. Schapire. 2010. A contextual-bandit approach to personalized news article recommendation.

**Limitations:**

yes

**Strengths And Weaknesses:**

I did not go through the proof line by line, but it seems all correct. Sufficient amount of experiments have been done to validate their claim.
This paper is well written and addresses a significant problem: the nonlinear reward model. However, this paper's main contribution is on the estimation part using BART, but the decision-making exactly follows the Thompson sampling.

---

> ### Author Rebuttal · Authors · 2026-03-31
>
> We thank you for carefully engaging with the theoretical aspects of our work and for your willingness to update your evaluation.
>
> ### Scope: finite discrete action spaces
> We agree that BFTS does not extend to continuous or infinite action spaces; this is by design. Our target application is mHealth, where arms are few and discrete: Drink Less has $K=3$ (Garnett et al., 2019), DIAMANTE has $K=9$ (Aguilera et al., 2024). The contribution is an original information-theoretic regret analysis for a tree-based contextual bandit, bringing BART into contextual bandits in a fully Bayesian, TS-compatible, and regret-analyzable way for finite, moderate-$K$ tabular settings. Empirically, BFTS achieves the best average rank on both synthetic and OpenML benchmarks, and the highest policy value on the Drink Less trial.
>
> We did compare joint encoding variants in our sensitivity analysis (Section 8): multi encoding achieves 1.051× normalized regret relative to separate-arm. Separate-arm's primary value is enabling the posterior factorization underlying Lemma 2 (Remark 1), not being the only functional encoding. Extending to continuous actions is also computationally intractable for BART, as finding the argmax of a single draw requires enumerating axis-aligned partition cells with no gradient signal (see Boyne et al., 2025).
>
> ### Q1: Removing the leading $K$ factor
> We appreciate this suggestion. The outer-product encoding from Li et al. (2010) is well-suited to linear bandits, where a lower-dimensional representation directly simplifies the linear reward model. For BART, which is nonparametric, changing the representation does not simplify the model in the same way: the leading $K$ is inherent to the arm-wise product prior, not an encoding artifact. Moreover, a $\log K$ binary encoding assumes arms have meaningful geometric structure. In our benchmarks, arms are nominal class labels (OpenML) or discrete message types (Drink Less); an arbitrary binary code introduces artificial similarities and is not invariant to action relabeling. Our action space has cardinality $K$ (typically 2–3), so there is no high-dimensional representation to compress.
>
> We note that Li et al. (2010) achieves $\tilde{O}(\sqrt{KdT})$, so the $K$ dependence is not removed even in the linear case. We may have misunderstood the claim that "the effective dimension will be reduced to $p\log K$"; if there is a specific result we are missing, we would be grateful for a pointer.
>
> ### Q2:  Linear baselines at a disadvantage because of inefficient encoding
> The $p \to Kp$ inflation under multi encoding is the standard protocol in Zhang et al. (2021) and Nilsson et al. (2024), so LinTS and LinUCB are not at a disadvantage relative to how they are typically evaluated. Moreover, compressing arms to $\log K$ bits while retaining a linear model cannot represent the original linear relationship, giving these baselines $O(T)$ regret from misspecification.
>
> ### Q3:  Depth-based exponential decay prior
> We use $\alpha_{\mathrm{qd}} = 0.45$ (Ročková & van der Pas, 2020) rather than the original Chipman et al. (2010) prior for three reasons: (1) it provides cleaner tree-complexity control in Lemma 2, making the structure term and expected leaf count easier to bound; (2) the frequentist analysis in Appendix E requires this quick-decay regime for existing BART contraction results; (3) empirically, it achieves better performance in our sensitivity analysis (Figure 8(h)). We will clarify this in the revision.
>
> ### References
>
> Garnett, C., Crane, D., West, R., Brown, J., & Michie, S. (2019). The development of Drink Less: an alcohol reduction smartphone app for excessive drinkers. *Translational Behavioral Medicine*, 9(2), 296–307.
>
> Aguilera, A., Arevalo Avalos, M., Xu, J., Chakraborty, B., et al. (2024). Effectiveness of a digital health intervention leveraging reinforcement learning: Results from the DIAMANTE randomized clinical trial. *Journal of Medical Internet Research*, 26(1), e60834.
>
> Boyne, T., Folch, J. P., Lee, R. M., Shafei, B., & Misener, R. (2025). BARK: A fully Bayesian tree kernel for black-box optimization. In *Proceedings of the 42nd International Conference on Machine Learning (ICML 2025)*.

---

> > ### Author Rebuttal · Reviewer_ZNmT · 2026-04-02
> >
> > I meant $\mathbb{R}^{kp} \rightarrow \mathbb{R}^{p \log k}$ when we use a $\log k$ binary encoding. However, as you mentioned, I realize this might not fully capture a non-linear model. Furthermore, since your evaluation is mainly for small set of arms, the encoding style might not be the reason for the performance degradation of LinUCB and LinTS; it is the inherent inability of the linear model.
> >
> > Thank you for the clarifications, and I shall raise the score.

---

> > > ### Author Response · Authors · 2026-04-03
> > >
> > > Thank you very much for the clarification and for the follow-up. We are glad that our rebuttal helped clarify the role of encoding in this setting. We also appreciate your recognition that, in the small-$K$ experiments, the weaker performance of LinUCB/LinTS is more likely due to linear model misspecification than to the choice of encoding.
> > >
> > > Thank you again for your careful reconsideration and for updating your evaluation. We will make this discussion clearer in the revision.

---

### Official Review · Reviewer_asBw · 2026-03-13

**Soundness:** 4
**Presentation:** 3
**Significance:** 3
**Originality:** 3
**Overall Recommendation:** 5
**Confidence:** 4

**Summary:**

Tree ensembles remain perhaps the premier method for prediction on tabular datasets, but their integration into exploration procedures like Thompson Sampling or UCB has been limited. The main issue is that it is difficult to rigorously quantify uncertainty via posterior samples or confidence bounds. This paper proposes to use Thompson Sampling together with BART (Bayesian Additive Regression Trees), which in principle admits exact posterior inference through its fully Bayesian specification. Theoretically, the authors derive an information-theoretic Bayesian regret bound of order $\tilde{O}(K\sigma \sqrt{T})$, where the information complexity term depends logarithmically on the tree structure prior (depth decay, split grid size) and the leaf parameter capacity. Empirically, they approximate posterior sampling via a batched Metropolis-within-Gibbs MCMC scheme with cold restarts on a logarithmic refresh schedule. They validate uncertainty quantification through credible interval coverage diagnostics, finding near-nominal calibration, and compare BFTS against linear, neural, and heuristic tree-ensemble baselines across eight synthetic scenarios, nine OpenML contextual bandit benchmarks, and a real mHealth micro-randomized trial (Drink Less). BFTS achieves the best average rank on both synthetic and OpenML benchmarks, with particularly strong gains in nonlinear settings, and the highest estimated policy value on the Drink Less trial, while exhibiting robust performance across hyperparameter configurations.

Overall, the manuscript's broad domain consists of contextual bandits with nonparametric reward models, motivated by mobile health applications. The authors attempt to discuss the key challenge of combining the predictive strength of tree ensembles with principled Bayesian exploration.

**Compliance With Llm Reviewing Policy:**

Affirmed.

**Key Questions For Authors:**

**Greedy BART ablation.** Have you considered evaluating a greedy variant of BFTS that always selects the arm with the highest posterior mean rather than sampling from the posterior? This would help disentangle how much of BFTS's performance comes from BART's modeling strength versus Thompson Sampling's exploration. Particularly in the settings studied here (K=2 or K=3 arms), it is plausible that implicit exploration from context variation is sufficient, and the calibrated posterior may matter less than the quality of the point predictions. This may explain why the XGBoostTS heuristic works well enough in certain problems.

**Limitations:**

Yes.

**Strengths And Weaknesses:**

# Strengths
**A. Fills a genuine gap in the literature.** To my knowledge, tree-based methods — despite being a go-to for practitioners on tabular data and a key benchmark in supervised learning — have not been rigorously combined with Thompson Sampling in this way. Prior tree-ensemble bandit methods relied on heuristic uncertainty quantification, breaking the theoretical link between posterior and regret. BFTS seems like a very sensible default in settings with tabular data, moderate dimensionality, and where compute per decision is not the bottleneck (e.g., mHealth, where decisions occur on the scale of hours or days).

***B. Transparent about what the paper does and does not do.** The authors clearly distinguish Ideal BFTS (exact posterior sampling, to which the theory applies) from Practical BFTS (batched MCMC approximation, which is what's implemented). They acknowledge where BFTS loses (linear settings, MNIST), that the frequentist bound applies to a variant they don't recommend in practice, and that MCMC mixing is imperfect. This transparency means the paper teaches the community something without leading anyone astray — a key criterion in my evaluation.

**C. Thorough and convincing empirical evaluation.** The experiments span eight synthetic scenarios, nine OpenML datasets, and a real mHealth trial, with 12 replications per setting. Beyond cumulative regret comparisons, several aspects stand out. The uncertainty calibration diagnostics (Figure 2) show that BFTS achieves near-nominal 95% credible interval coverage across settings, while baselines like XGBoostTS exhibit unstable coverage. This is important because it validates the mechanism behind low regret, not just the outcome: the posterior is faithfully representing uncertainty, which is what makes Thompson Sampling work. The sensitivity analysis is unusually comprehensive, sweeping over encoding choices, refresh schedules, number of chains, trees, burn-in iterations, and prior hyperparameters. The finding that performance is robust across these variations (normalized regret varying by only a few percent in most cases) is arguably the paper's most practically important result, since in online learning one cannot cross-validate.

**D. The theoretical analysis appears proper and sensible.** The Bayesian regret bound provides useful confirmation that the BART prior's complexity is well-controlled in the bandit setting, and the proof is cleanly structured. I have not verified the frequentist analysis in Appendix E.

# Weaknesses


**No theoretical guarantees on MCMC approximation quality.** The paper's regret bounds apply to Ideal BFTS with exact posterior sampling, but the practical algorithm relies on batched MCMC that exhibits imperfect mixing. The authors do not provide formal analysis of how approximation error in the posterior samples propagates to regret. That said, they are upfront about this gap and provide empirical evidence that decision-level stability and posterior calibration are adequate.

**The practical margin over heuristic methods is sometimes slim.** XGBoostTS provides notably poor uncertainty quantification (unstable credible interval coverage), yet it remains quite competitive on regret across many benchmarks: it is the second-best method on OpenML with an average rank of 2.33 versus BFTS's 1.89, and on datasets like Adult the two are numerically close. This raises a natural question about when the cost of principled posterior inference is justified. In settings with small action spaces and sufficient natural variation in contexts, even greedy or heuristic methods may explore adequately without a calibrated posterior. The paper would benefit from a greedy BART ablation (same model, always pick the highest posterior mean) to isolate the value of Thompson Sampling's exploration from the value of BART's modeling. More broadly, practitioners in settings where compute or latency matters may find that the simpler heuristic alternatives are sufficient.

---

> ### Author Rebuttal · Authors · 2026-03-31
>
> We thank you for the thorough and supportive review. We particularly appreciate the observation that BFTS's transparency about the Ideal-vs-Practical distinction is itself a feature, and the suggestion for a greedy ablation, which we believe substantially strengthens the paper.
>
> **Theoretical guarantees on MCMC approximation quality.**
> We agree that the current manuscript does not provide a formal regret guarantee for Practical BFTS; our claim there is empirical rather than theoretical. We note that for Thompson sampling, the more directly relevant approximation target is the induced action distribution rather than the full BART posterior, which may explain why imperfect parameter-space mixing does not translate into large regret in practice. A formal bridge between Practical and Ideal BFTS along these lines is an interesting direction for future work.
>
> **Greedy BART ablation.**
> We thank the reviewer for this suggestion, which we found very informative. We ran a factorial comparison (BART vs XGBoost) × (Greedy vs $\epsilon$-Greedy vs Thompson sampling), keeping the same BART posterior, inference budget, and refresh schedule and changing only the action rule. Table 1 reports results on five representative OpenML datasets (all variants use `n_chains=1`).
>
> First, BFTS's performance comes more from modeling than from exploration. The dominant contrast in Table X is BART vs XGBoost: on Shuttle, XGB-Greedy achieves 803±741 while BART-Greedy achieves 105±18, confirming your observation that BFTS's value may lie mostly in its calibrated Bayesian reward model. Second, $\epsilon$-Greedy (1% random exploration) is consistently worse than both pure Greedy and TS, indicating that blind random perturbation is counterproductive, whereas posterior-guided sampling is principled. Third, TS is the variant covered by our Bayesian regret guarantee and provides robustness insurance. BFTS never catastrophically fails on any dataset, whereas XGB-Greedy does.
>
> **Table 1. Final regret for action-rule ablation (mean ± sd over 16 runs).**
>
> | Dataset        |    BART: Greedy / $\epsilon$-Greedy / TS | XGB: Greedy / $\epsilon$-Greedy / TS |
> | -------------- | ------------------------------: | --------------------------: |
> | Adult          | **1491±38** / 1533±44 / 1553±35 | 1511±34 / 1524±34 / 1548±37 |
> | MagicTelescope | **1393±37** / 1425±37 / 1487±29 | 1513±33 / 1541±22 / 1591±30 |
> | Mushroom       |        59±14 / 93±8 / **55±10** |      79±28 / 132±31 / 80±12 |
> | Shuttle        |     105±18 / 197±30 / **99±14** | 803±741 / 383±330 / 191±172 |
> | PageBlocks     |    299±23 / 327±22 / **278±25** |    369±85 / 339±34 / 301±13 |

---

> > ### Author Rebuttal · Reviewer_asBw · 2026-04-02
> >
> > Thanks for quickly running these extra experiments! I find the extra ablations are quite interesting.
> >
> > One thought for the future: have you thought about looking for settings where Greedy exploration is not so effective, to stress test the posterior sampling component of your solution?
> >
> > I'm going to keep the original rating (5: Accept), which was already high. Thanks for writing a nice paper.

---

> > > ### Author Response · Authors · 2026-04-03
> > >
> > > Thank you for the suggestion and for the encouraging feedback. We agree that this is a natural next step in light of the ablation, which suggests that greedy can already be strong on some tabular benchmarks. It would therefore be very useful to evaluate settings where greedy is less effective, for example, with reduced initial forced exploration, longer horizons, regime shift, or tasks with smaller reward gaps between arms. We will note this in the discussion of future work.

---

### Official Review · Reviewer_FSt3 · 2026-03-16

**Soundness:** 3
**Presentation:** 3
**Significance:** 3
**Originality:** 3
**Overall Recommendation:** 4
**Confidence:** 4

**Summary:**

This paper proposes Bayesian Forest Thompson Sampling (BFTS), a contextual bandit algorithm that uses arm-wise Bayesian Additive Regression Trees (BART) as the reward model for Thompson Sampling (TS). Each arm $a \in \mathcal{A}$ is modeled by an independent BART ensemble $g_{\theta_a}(x) = \sum_{j=1}^{m} g_{T_j^{(a)}, M_j^{(a)}}(x)$, and posterior samples are obtained via batched cold-start MCMC with a logarithmic refresh schedule $r(t) = \lceil 8 \log t \rceil$. The authors derive an information-theoretic Bayesian regret bound

$$\mathbb{E}[\text{Regret}_T] \leq K\sigma\sqrt{2T m \Psi_T}$$

 (where $\Psi_T$ is logarithmic in $p, N_{\max}, T$) for an idealized version with exact posterior sampling, and provide a supplementary frequentist minimax-rate analysis $\tilde{O}(K T^{(\alpha+d)/(2\alpha+d)})$ for a "feel-good" variant in the appendix. Empirically, BFTS achieves the best average rank across eight synthetic DGPs (1.63 vs. 3.00 for NeuralTS), the best average rank across nine OpenML contextual bandit benchmarks (1.89), and the highest estimated policy value on the Drink Less mHealth micro-randomized trial. A comprehensive sensitivity analysis demonstrates robustness to hyperparameter choices.

**Compliance With Llm Reviewing Policy:**

Affirmed.

**Final Justification:**

BFTS fills a genuine gap by equipping BART with information-theoretic regret analysis for contextual bandits. Lemma 2's MI bound for Bayesian forests is novel, and the empirical evaluation is exceptionally thorough.

The rebuttal resolved all five concerns: the $\Delta_t \cdot \mathrm{TV}$ decomposition scopes the theory-practice gap precisely (Q1); the Gaussian vs. logistic comparison and cold/warm-start ablation justify current defaults with direct evidence (Q3, Q4); the $\sqrt{T}$ rate is correctly framed as MI control, not a claim of advantage over linear models (Q2); and the moderate-$K$ scoping is appropriate (Q5). The main residual limitation—no formal guarantee for Practical BFTS—is acknowledged and does not undermine the contribution. I maintain **4 (weak accept)**.

**Key Questions For Authors:**

1. **Theory-practice gap.** Can you provide any formal guarantee connecting Practical BFTS's regret to Ideal BFTS's? For instance, can the Policy $\Delta$-TV diagnostic be related to per-round regret excess? A bound of the form $\text{Regret}_T^{\text{practical}} \leq \text{Regret}_T^{\text{ideal}} + \text{(MCMC error)}$ would substantially strengthen the paper. Recent work such as HyperAgent (Li et al., 2024) achieves this for index-sampling-based posterior approximations with provable guarantees, and Ensemble++ (Li et al., 2025) provides scalable approximate posterior sampling with formal regret control. How does BFTS's MCMC approximation compare, and could similar analytical techniques be applied?

2. **Bound informativeness.** The $\tilde{O}(K\sigma\sqrt{T})$ Bayesian regret rate appears to match what a correctly specified linear model would achieve. Can you clarify what this bound reveals about BART specifically? Is there a way to tighten it to reflect BART's adaptive complexity (e.g., via a data-dependent information complexity term)?

3. **Binary reward handling.** Have you evaluated logistic or probit BART (Murray, 2021) on the OpenML benchmarks? Given the systematic Gaussian misspecification with $\{0,1\}$ rewards, this seems like a natural extension that could improve both calibration and regret.

4. **Cold-start vs. warm-start.** You reinitialize MCMC from the prior at every refresh with $n_{\text{burn}}=500$. Given $\hat{R}$ values up to 1.90, what evidence supports that 500 burn-in steps suffice? Have you compared warm-start (continuing from previous chains) performance?

5. **Scaling with $K$.** BFTS underperforms on Covertype ($K=10$) and MNIST ($K=10$, high-dimensional). At what point does the $K$-fold cost of arm-wise modeling outweigh its theoretical and empirical benefits? Methods like HyperAgent (Li et al., 2024) and Ensemble++ (Li et al., 2025) scale gracefully with $K$ via shared representations. Is there a principled way to share information across arms without the dimension inflation discussed in Remark 1?

**Limitations:**

The authors discuss computational fidelity, scalable inference, information sharing, and non-stationarity in Section 9, which is appreciated. However, the following limitations are insufficiently addressed: (1) the formal gap between Ideal and Practical BFTS—particularly in light of recent methods (Li et al., 2024; Li et al., 2025) that achieve scalable approximate posterior sampling with formal regret guarantees; (2) the implications of Gaussian misspecification for binary rewards; (3) scalability to large $K$ or high-dimensional contexts. The impact statement is appropriate but brief. No negative societal impact concerns are apparent beyond standard considerations for adaptive health interventions.

**Strengths And Weaknesses:**

### Strengths

**S1 (Originality & Motivation).** Overall, the manuscript's broad domain consists of contextual bandits for tabular data, and the paper identifies a genuine gap: tree ensembles dominate tabular supervised learning but lack the Bayesian structure required by TS. Using BART—a fully probabilistic sum-of-trees model—to fill this gap is natural and well-motivated. The arm-wise modeling is cleanly justified both theoretically (enabling posterior factorization in Lemma 2) and empirically (normalized regret $3.15\times$ worse with one-hot encoding; Section 8).

**S2 (Bayesian Regret Analysis).** The mutual information bound for BART ensembles (Lemma 2) is a novel technical contribution. The four-step decomposition—arm-wise factorization $\to$ structure-leaf chain rule $\to$ prior entropy bound on tree structures $\to$ Gaussian capacity bound on leaves—is elegant and clearly presented. The use of the lifted information ratio framework (Neu et al., 2022; Gouverneur et al., 2023) with $\bar{\Gamma}_T \leq 2\sigma^2 K$ is appropriate.

**S3 (Comprehensive Empirics).** The experimental evaluation is thorough: 8 synthetic DGPs, 9 OpenML datasets, and a real mHealth trial. The paper includes uncertainty calibration diagnostics (CI frontiers, ECE evolution in Figure 5), feature-inclusion frontiers (Figure 6), MCMC mixing diagnostics (Table 5 with $\hat{R}$, acceptance rates, Policy $\Delta$-TV), runtime comparisons (Figure 10), and pairwise statistical testing (Table 4 with Bonferroni-corrected Mann–Whitney tests). This level of thoroughness is exemplary.

**S4 (Practical Relevance).** The Drink Less case study is well-executed: OPE uses both SNIPS and DR estimators with cluster bootstrap ($B=30$), importance weight diagnostics (bounded at $\bar{w}_{\max} = 3.33$, ESS $\approx 4944$), and interpretability via posterior split probabilities identifying `days_since_download` (0.192), `AUDIT_score` (0.138), and `age` (0.128) as key features.

**S5 (Robustness to Hyperparameters).** The sensitivity analysis (Section 8, Figures 8–9) sweeps encoding, refresh schedule, chain count, number of trees, burn-in, depth prior $\alpha_{qd}$, leaf shrinkage $\kappa$, Dirichlet prior, and split-grid cap. Aside from the encoding choice, normalized regret varies by only $\sim$1.01–1.17$\times$, supporting minimal tuning.

### Weaknesses

**W1 (Theory-Practice Gap).** The Bayesian regret bound (Theorem 1) applies only to Ideal BFTS with exact posterior sampling, while Practical BFTS uses batched cold-start MCMC with acknowledged mixing issues ($\hat{R} \in [1.19, 1.90]$ at $t=10{,}000$; Table 5). This gap is never formally quantified. The authors argue Policy $\Delta$-TV (0.03–0.08) suffices, but this is a heuristic argument without a formal connection to regret. Existing work on approximate TS (Phan et al., 2019; Mazumdar et al., 2020) is cited but not applied to bound the approximation error. Recent work on scalable posterior sampling—notably HyperAgent (Li, Xu, Han, & Luo, 2024), which bridges theory and practice for posterior sampling via hypermodel-based index sampling with provable regret guarantees, and Ensemble++ (Li, Xu, Wang, & Luo, 2025), which achieves scalable exploration through lightweight ensemble perturbations—demonstrates that it is possible to design practical approximate posterior sampling schemes with formal regret control. The authors should discuss how BFTS relates to these approaches and whether similar techniques could close the Ideal-vs-Practical gap.

**W2 (Limited Informativeness of Bayesian Bound).** The $\tilde{O}(K\sigma\sqrt{T})$ rate does not reflect BART's nonparametric expressiveness. Under correct Bayesian specification, a linear model with $K$ arms also achieves $O(K\sqrt{T \log T})$ Bayesian regret. The bound shows BART is "not worse" but does not demonstrate a theoretical advantage for nonlinear settings. The more informative frequentist bound $\tilde{O}(KT^{(\alpha+d)/(2\alpha+d)})$ applies to the feel-good variant, which the authors themselves recommend against using in practice (Section E.8: normalized regret up to $8.5\times$ at aggressive $\eta$).

**W3 (MCMC Mixing).** The reported $\hat{R}$ values (1.19–1.90) substantially exceed the modern convergence standard ($\hat{R} < 1.01$). With $m=100$ trees per arm and only $n_{\text{burn}} = 500$ burn-in iterations after cold-start reinitialization, it is unclear whether chains adequately traverse the multimodal BART posterior. The authors' own citations (Ronen et al., 2022; Tan et al., 2024) document BART's mixing difficulties.

**W4 (Gaussian Misspecification on Binary Rewards).** The authors attempt to discuss the key challenge of model misspecification briefly (Section 5.3), noting that binary OpenML rewards violate the Gaussian likelihood. However, they do not evaluate its impact on posterior calibration or consider logistic/probit BART alternatives (e.g., Murray, 2021, which they cite for a different purpose).

**W5 (Experimental Scale).** All experiments use $T = 10{,}000$ with only $R = 12$ replications. The small replication count yields wide standard deviations for some baselines (e.g., NeuralTS on Adult: $3979.7 \pm 1684.3$), weakening statistical comparisons. On the Drink Less trial, the final-horizon SNIPS improvement over the best baseline (LinTS tuned) is marginal at 1.5%.

**W6 (Scalability to Many Arms).** Arm-wise modeling requires $K$ independent BART ensembles. On Covertype ($K=10$), BFTS is outperformed by LinTS. The statistical cost is also relevant: each arm's model trains on only $\sim T/K$ observations. The paper does not discuss this scaling limitation beyond a brief remark (Remark 1). Scalable exploration methods such as HyperAgent (Li et al., 2024) and Ensemble++ (Li et al., 2025) avoid the $K$-fold modeling cost by maintaining a single shared model with efficient perturbation-based exploration, and comparing against such approaches would clarify the regime where arm-wise BART modeling remains competitive.

- Li, Y., Xu, J., Han, L., & Luo, Z.-Q. (2024). Q-Star Meets Scalable Posterior Sampling: Bridging Theory and Practice via HyperAgent. *ICML 2024*, PMLR 235, pp. 29022–29062.
- Li, Y., Xu, J., Wang, B., & Luo, Z.-Q. (2025). Scalable Exploration via Ensemble++. *NeurIPS 2025*.

---

> ### Author Rebuttal · Authors · 2026-03-31
>
> We thank you for the detailed and well-organized review.
>
> ### Q1: Theory-practice gap (Ideal vs. Practical BFTS)
> We agree that the paper does not provide a formal regret guarantee for Practical BFTS. Policy $\Delta$-TV is a decision-stability diagnostic, not a formal bridge; we will revise the text to make this clearer. A formal bridge would decompose per-round excess regret as
>
> $$\text{ExcessReg}_t(x) \le \Delta_t(x) \cdot \mathrm{TV}(\tilde\pi_t(\cdot \mid x), \pi_t^{\mathrm{ideal}}(\cdot \mid x)),$$
>
> where $\Delta_t(x) = \max_a f_0(x,a) - \min_a f_0(x,a)$ is the instantaneous gap and $\tilde\pi_t$, $\pi_t^{\mathrm{ideal}}$ are the practical and exact-posterior action distributions. The missing step is nontrivial but well-scoped future work.
>
> HyperAgent and Ensemble++ target tabular RL and linear contextual bandits respectively, whereas our approximation gap is structurally different, arising from batched MCMC for a nonparametric BART posterior with stale-posterior error between logarithmic refreshes. Their analytical frameworks rely on closed-form perturbations of parametric models; applying similar techniques to BFTS would require replacing the MCMC sampler with a perturbation-based approximation, thereby losing the fully Bayesian interpretation central to our approach. We will cite and discuss these works in the revision.
>
> ### Q2: Limited informativeness of the Bayesian bound
> The $\tilde{O}(K\sigma\sqrt{T})$ rate matching linear TS is precisely the point: a far richer nonparametric prior (sum-of-trees with discrete structures and continuous leaves) still achieves $\sqrt{T}$-type regret, with complexity entering $\Psi_T$ only logarithmically in $p$, $N_{\max}$, $T$. The novelty is Lemma 2's mutual information control for Bayesian forests, not a claim of faster rates. Whether this can be tightened to a data-dependent term is an interesting open question.
>
> ### Q3: Gaussian misspecification on binary rewards
> We acknowledge that binary OpenML rewards violate the Gaussian likelihood. This aligns with the classical protocols of Riquelme et al. (2018), Zhang et al. (2021), and Nilsson et al. (2024). We implemented a logistic-BART TS variant (Murray, 2021) and compared it with Gaussian BFTS on four binary-reward OpenML datasets.
>
> **Table 1. Gaussian vs. logistic (final regret, mean ± sd, 12 runs).**
>
> | Dataset | BFTS-Gaussian | BFTS-Logistic |
> |---|---:|---:|
> | Adult | 1551 ± 40 | 1529 ± 29 |
> | MagicTelescope | **1487 ± 31** | 1533 ± 27 |
> | Mushroom | **53 ± 6** | 94 ± 6 |
> | EEGEyeState | 2244 ± 43 | **2183 ± 25** |
>
> Neither dominates (Gaussian 2 wins, Logistic 1 win, 1 tie), while logistic BART is ~1.6× slower due to latent-variable augmentation MCMC. The Gaussian working likelihood is a robust practical default, consistent with its widespread use.
>
> ### Q4: Cold-start vs. warm-start
> We compared four variants on four core OpenML datasets:
>
> **Table 2. Cold-start vs. warm-start MCMC (final regret, mean ± sd, 8 runs).**
>
> | Variant | Adult | MagicTelescope | Mushroom | Shuttle |
> |---|---:|---:|---:|---:|
> | Cold-250 | 1559 ± 38 | 1485 ± 30 | 61 ± 13 | 120 ± 13 |
> | Cold-500 | 1561 ± 40 | 1472 ± 26 | 58 ± 10 | 128 ± 24 |
> | Warm-250 | 1556 ± 31 | 1468 ± 38 | 63 ± 38 | 116 ± 18 |
> | Warm-500 | 1560 ± 34 | 1483 ± 28 | 63 ± 24 | 106 ± 16 |
>
> All four variants produce statistically indistinguishable regret. Combined with the burn-in sweep in our sensitivity analysis (Figure 8(e)), this is direct evidence that 500 burn-in suffices: halving it to 250 does not degrade performance, and switching to warm-start does not help either. The current cold-start with 500 burn-in is a conservative but robust default.
>
> ### Q5: Scalability to many arms
> BFTS targets moderate-$K$ tabular settings; our motivating application is mHealth where Drink Less has $K=3$ and DIAMANTE has $K=9$. On Covertype and MNIST ($K=10$), BFTS does not outperform LinTS or NeuralTS, consistent with the limitation in Section 9. The leading $K$ in our regret bound is inherent to the arm-wise posterior factorization (Lemma 2, Remark 1); shared-model extensions require different theory and are important future work (see our response to Reviewer bbAW Q1&Q2 for more detail).
>
> ### W5: Small replication count and wide standard deviations
> Thanks for raising this point. In the revision we will perform more replications to reduce the standard deviation.
>
> ### References
>
> Garnett, C., Crane, D., West, R., Brown, J., & Michie, S. (2019). The development of Drink Less: an alcohol reduction smartphone app for excessive drinkers. *Translational Behavioral Medicine*, 9(2), 296–307.
>
> Aguilera, A., Arevalo Avalos, M., Xu, J., Chakraborty, B., et al. (2024). Effectiveness of a digital health intervention leveraging reinforcement learning: Results from the DIAMANTE randomized clinical trial. *Journal of Medical Internet Research*, 26(1), e60834.

---

> > ### Author Rebuttal · Reviewer_FSt3 · 2026-04-03
> >
> > **Q1 (theory-practice gap):** The $\Delta_t(x) \cdot \mathrm{TV}(\tilde\pi_t, \pi_t^{\mathrm{ideal}})$ decomposition cleanly scopes the missing bridge. The explanation that HyperAgent/Ensemble++ rely on closed-form perturbations of parametric models—structurally different from batched MCMC over a nonparametric posterior—is reasonable.
> >
> > **Q2 (bound informativeness):** I accept that Lemma 2's contribution is establishing MI control for Bayesian forests at $\sqrt{T}$ rate, not claiming faster rates over linear models.
> >
> > **Q3 (binary rewards):** The Gaussian vs. logistic BART comparison (Table 1) shows neither dominates while logistic BART is ~1.6× slower, supporting the Gaussian default.
> >
> > **Q4 (cold-start vs. warm-start):** Table 2 directly shows all four variants yield indistinguishable regret, addressing both the burn-in sufficiency and warm-start questions.
> >
> > **Q5 (scaling with $K$):** The limitation is clearly acknowledged and appropriately scoped to moderate-$K$ mHealth settings.
> >
> > The commitment to additional replications (W5) is appreciated. I maintain my score of 4 (weak accept).

---

> > > ### Author Response · Authors · 2026-04-04
> > >
> > > Thank you for the update. We are glad the clarification and targeted follow-up experiments addressed your concerns, and we will incorporate these points in the revision.

---

### Decision · Program_Chairs · 2026-04-30

**Decision:**

Accept (spotlight)

**Comment:**

This paper proposes to use Thompson Sampling together with BART (Bayesian Additive Regression Trees), which in principle admits exact posterior inference through its fully Bayesian specification. Theoretically, this paper derives an information-theoretic Bayesian regret bound where the information complexity term depends logarithmically on the tree structure prior and the leaf parameter capacity. Empirically, this paper compares BFTS against linear, neural, and heuristic tree-ensemble baselines across eight synthetic scenarios, nine OpenML contextual bandit benchmarks, and a real mHealth micro-randomized trial (Drink Less). BFTS achieves the best average rank on both synthetic and OpenML benchmarks, with particularly strong gains in nonlinear settings.

To the best of my knowledge, this paper fills a gap in the literature by combining the tree-based methods with Thompson sampling. Its empirical evaluation is thorough and convincing, and its theoretical analyses are also sensible. All reviewers recommend to accept this paper. After reading the paper, the reviews, and the rebuttals, I also recommend accepting this paper.